# How Many Pretraining Tasks Are Needed for In-Context Learning of Linear Regression?

**Jingfeng Wu**
UC Berkeley
`uuujf@berkeley.edu`

**Difan Zou**
The University of Hong Kong
`dzou@cs.hku.hk`

**Zixiang Chen**
UCLA
`chenzx19@cs.ucla.edu`

**Vladimir Braverman**
Rice University
`vb21@rice.edu`

**Quanquan Gu**
UCLA
`qgu@cs.ucla.edu`

**Peter L. Bartlett**
Google DeepMind & UC Berkeley
`peter@berkeley.edu`

## Abstract

Transformers pretrained on diverse tasks exhibit remarkable *in-context learning* (ICL) capabilities, enabling them to solve unseen tasks solely based on input contexts without adjusting model parameters. In this paper, we study ICL in one of its simplest setups: pretraining a linearly parameterized single-layer linear attention model for linear regression with a Gaussian prior. We establish a statistical task complexity bound for the attention model pretraining, showing that effective pretraining only requires a small number of independent tasks. Furthermore, we prove that the pretrained model closely matches the Bayes optimal algorithm, i.e., optimally tuned ridge regression, by achieving nearly Bayes optimal risk on unseen tasks under a fixed context length. These theoretical findings complement prior experimental research and shed light on the statistical foundations of ICL.

## 1 Introduction

Transformer-based large language models (Vaswani et al., 2017) pretrained with diverse tasks have demonstrated strong ability for *in-context learning* (ICL), that is, the pretrained models can answer new queries based on a few in-context demonstrations (see Brown et al., 2020, and references thereafter). ICL is one of the key abilities contributing to the success of large language models, which allows pretrained models to solve multiple downstream tasks without updating their model parameters. However, the statistical foundation of ICL is still in its infancy.

A recent line of research aims to quantify ICL by studying transformers pretrained on the linear regression task with a Gaussian prior (Garg et al., 2022; Akyürek et al., 2022; Li et al., 2023b; Raventos et al., 2023). Specifically, Garg et al. (2022); Akyürek et al. (2022); Li et al. (2023b) study the setting where transformers are pretrained in an online manner using independent linear regression tasks with the same Gaussian prior. They find that such a pretrained transformer can perform ICL on fresh linear regression tasks. More surprisingly, the average regression error achieved by ICL is nearly *Bayes optimal*, and closely matches the average regression error achieved by an optimally tuned ridge regression given the same amount of context data. Later, Raventos et al. (2023) show that the nearly optimal ICL is achievable even if the transformer is pretrained with multiple passes of a *limited* number of independent linear regression tasks.

On the other hand, a connection has been drawn between the *forward pass* of (multi-layer) Transformers and (multi-step) *gradient descent* (GD) algorithms (Akyürek et al., 2022; Von Oswald et al., 2023; Bai et al., 2023; Ahn et al., 2023; Zhang et al., 2023a), offering a potential ICL mechanism by simulating GD (which serves as a meta-algorithm that can realize many machine learning algorithms such as empirical risk minimization). Specifically, Von Oswald et al. (2023); Akyürek et al. (2022); Bai et al. (2023) show, by construction, that multi-layer transformers are sufficiently expressive to implement multi-step GD algorithms. In addition, Ahn et al. (2023); Zhang et al. (2023a) prove that for the ICL of linear regression by *single-layer linear attention* models, a global minimizer of the (population) pretraining loss can be equivalently viewed as *one-step GD with a matrix stepsize*.

**Our contributions.** Motivated by the above two lines of research, in this paper, we consider ICL in the arguably simplest setting: pretraining a (restricted) single-layer linear attention model for linear regression with a Gaussian prior. Our first contribution is **a statistical task complexity bound for pretraining** the attention model (see Theorem 4.1). Despite that the attention model contains $d^2$ free parameters, where $d$ is the dimension of the linear regression task and is assumed to be large, our bound suggests that the attention model can be effectively pretrained with a *dimension-independent* number of linear regression tasks. Our theory is consistent with the empirical observations made by Raventos et al. (2023).

Our second contribution is **a thorough theoretical analysis of the ICL performance** of the pretrained model (see Theorem 5.3). We compute the average linear regression error achieved by an optimally pretrained single-layer linear attention model and compare it with that achieved by an optimally tuned ridge regression. When the context length in inference is close to that in pretraining, the pretrained attention model is a *Bayes optimal* predictor, whose error matches that of an optimally tuned ridge regression. However, when the context length in inference significantly differs from that in pretraining, the pretrained single-layer linear attention model might be suboptimal.

Besides, this paper contributes **novel techniques for analyzing high-order tensors**. Our major tool is an extension of the operator method developed for analyzing 4-th order tensors (i.e., linear operators on matrices) in linear regression (Bach & Moulines, 2013; Dieuleveut et al., 2017; Jain et al., 2018; 2017; Zou et al., 2021; Wu et al., 2022) and ReLU regression (Wu et al., 2023) to 8-th order tensors (which correspond to linear maps on operators). We introduce two powerful new tools, namely *diagonalization* and *operator polynomials*, to this end (see Section 6 for more discussion). We believe our techniques are of independent interest in analyzing similar problems.

## 2 RELATED WORK

**Empirical results for ICL for linear regression.** As mentioned earlier, our paper is motivated by a set of empirical results on ICL for linear regression (Garg et al., 2022; Akyürek et al., 2022; Li et al., 2023b; Raventos et al., 2023; Bai et al., 2023). Along this line, the initial work by Garg et al. (2022) considers ICL for noiseless linear regression, where they find the ICL performance of pretrained transformers is close to ordinary least squares. Later, Akyürek et al. (2022); Li et al. (2023b) extend their results by considering ICL for linear regression with additive noise. In this case, pretrained transformers perform ICL in a Bayes optimal way, matching the performance of optimally tuned ridge regression. Recently, Bai et al. (2023) consider ICL for linear regression with mixed noise levels and demonstrate that pretrained transformers can perform algorithm selection. In all these works, transformers are pretrained by an online algorithm, seeing an independent linear regression task at each optimization step. In contrast, Raventos et al. (2023) pretrain transformers using a multi-pass algorithm over a limited number of linear regression tasks. Quite surprisingly, such pretrained transformers are still able to do ICL nearly Bayes optimally. Our results can be viewed as theoretical justifications for the empirical findings of Garg et al. (2022); Akyürek et al. (2022); Li et al. (2023b); Raventos et al. (2023).

**Attention models simulating GD.** Recent works explain the ICL of transformers by their capability to simulate GD. This idea is formalized by Akyürek et al. (2022); Von Oswald et al. (2023); Dai et al. (2023), where they show, *by construction*, that an attention layer is expressive enough to compute one GD step. Based on the above observations, Giannou et al. (2023); Bai et al. (2023) show transformers can *approximate* programmable computers as well as general machine learning algorithms. In addition, Li et al. (2023a) show the closeness between single-layer self-attention and GD on softmax regression under some conditions. Focusing on ICL for linear regression by single-layer linear attention models, Ahn et al. (2023); Zhang et al. (2023a) prove that one global minimizer of the population ICL loss can be *equivalently* viewed as one-step GD with a matrix stepsize. A similar result specialized to ICL for isotropic linear regression has also appeared in (Mahankali et al., 2024). Notably, Zhang et al. (2023a) also consider the optimization of the attention model, but their results require infinite pretraining tasks; and Bai et al. (2023) also establish task complexity bounds for pretraining, but their bounds are based on *uniform convergence* and are therefore crude (see discussions after Theorem 4.1). In contrast, we conduct a fine-grained analysis of the task complexity bounds for pretraining a single-layer linear attention model with a simplified linear parameterization and obtain much sharper bounds.

**Additional ICL theory.** In addition to the above works, there are other explanations for ICL. For an incomplete list, Li et al. (2023b) use algorithm stability to show a generalization bound for ICL, Xie et al. (2021); Wang et al. (2023) explain ICL via Bayes inference, Li et al. (2023c) show transformers can learn topic structure, Zhang et al. (2023b) explain ICL as Bayes model averaging, and Han et al. (2023) connect ICL to kernel regression. These results are not directly comparable to ours, as we focus on studying the ICL of a single-layer linear attention model for linear regression.

## 3 PRELIMINARIES

**Linear regression with a Gaussian prior.** We will use $\mathbf{x} \in \mathbb{R}^d$ and $y \in \mathbb{R}$ to denote the covariate and response for the regression problem. We state our results in the finite-dimensional setting but most of our results are dimension-free and they can be extended to the case when $\mathbf{x}$ belongs to a possibly infinite dimensional Hilbert space.

**Assumption 1** (A fixed size dataset). *For a fixed number of contexts $n \geq 0$, a dataset[1] of size $n + 1$, denoted by $(\mathbf{X}, \mathbf{y}, \mathbf{x}, y) \in \mathbb{R}^{n \times d} \times \mathbb{R}^n \times \mathbb{R}^d \times \mathbb{R}$, is generated as follows:*

- *A task parameter is generated from a Gaussian prior, $\boldsymbol{\beta} \sim \mathcal{N}(0, \psi^2 \mathbf{I}_d)$.*

- *Conditioned on $\boldsymbol{\beta}$, $(\mathbf{x}, y)$ is generated by $\mathbf{x} \sim \mathcal{N}(0, \mathbf{H})$ and $y \sim \mathcal{N}(\boldsymbol{\beta}^\top \mathbf{x}, \sigma^2)$.*

- *Conditioned on $\boldsymbol{\beta}$, each row of $(\mathbf{X}, \mathbf{y}) \in \mathbb{R}^{n \times (d+1)}$ is an independent copy of $(\mathbf{x}^\top, y) \in \mathbb{R}^{d+1}$.*

*Here, $\psi^2 > 0$, $\sigma^2 \geq 0$, and $\mathbf{H} \succeq 0$ are fixed but unknown quantities that govern the population data distribution. Without loss of generality, we assume $\mathbf{H}$ is strictly positive definite. We will refer to $(\mathbf{X}, \mathbf{y})$, $\mathbf{x}$, and $y$ as* contexts, *covariate, and* response, *respectively.*

**A restricted single-layer linear attention model.** We use $f$ to denote a model for ICL, which takes a sequence of contexts (of an unspecified length) and a covariate as inputs and outputs a prediction of the response, i.e.,

$$f : (\mathbb{R}^d \times \mathbb{R})^* \times \mathbb{R}^d \to \mathbb{R}$$
$$(\mathbf{X}, \mathbf{y}, \mathbf{x}) \mapsto \hat{y} := f(\mathbf{X}, \mathbf{y}, \mathbf{x}).$$

We will consider a (restricted version of a) *single-layer linear attention* model, which is closely related to one-step *gradient descent* (GD) with *matrix stepsizes* as model parameters. Specifically, based on the results of Ahn et al. (2023); Zhang et al. (2023a), one can see that the function class of single-layer linear attention models (when some parameters are fixed to be zero) is *equivalent* to the function class of one-step GD with matrix stepsizes as model parameters (see Appendix B for a proof). Therefore, we will take the latter form for simplicity and consider an ICL model parameterized as a one-step GD with matrix stepsize, that is,

$$f(\mathbf{X}, \mathbf{y}, \mathbf{x}; \boldsymbol{\Gamma}) := \left\langle \frac{\boldsymbol{\Gamma} \mathbf{X}^\top \mathbf{y}}{\dim(\mathbf{y})}, \mathbf{x} \right\rangle, \quad \boldsymbol{\Gamma} \in \mathbb{R}^{d \times d}, \tag{1}$$

where $\boldsymbol{\Gamma}$ is a $d^2$-dimensional matrix parameter to be optimized, and $\dim(\mathbf{y})$ is the dimension of $\mathbf{y}$. That is, we consider two simplifications of the usual soft-max self-attention model: we remove the nonlinearity and we replace the usual parametrization with a simpler linear one (see Appendix B).

**ICL risk.** For model (1) with a fixed parameter $\boldsymbol{\Gamma}$, we measure its ICL risk by its *average regression risk on an independent dataset*. Specifically, for $n \geq 0$, the ICL risk evaluated on a dataset of size $n$ is defined by

$$\mathcal{R}_n(\boldsymbol{\Gamma}) := \mathbb{E}\big(f(\mathbf{X}, \mathbf{y}, \mathbf{x}; \boldsymbol{\Gamma}) - y\big)^2, \tag{2}$$

where the expectation is taken with respect to the dataset $(\mathbf{X}, \mathbf{y}, \mathbf{x}, y)$ generated according to Assumption 1 with $n$ contexts.

---

[1] We will set $n = N$ to generate datasets for pretraining and $n = M$ to generate datasets for inference, where $M$ is allowed to be different from $N$.

We have the following theorem characterizing useful facts about the ICL risk (2). Special cases of Theorem 3.1 when $\sigma^2 = 0$ have appeared in (Ahn et al., 2023; Zhang et al., 2023a). The proof is deferred to Appendix C. For two matrices $\mathbf{A}$ and $\mathbf{B}$ of the same shape, we define $\langle \mathbf{A}, \mathbf{B} \rangle := \mathtt{tr}(\mathbf{A}^\top \mathbf{B})$.

**Theorem 3.1** (ICL risk). *Fix $N \geq 0$ as the number of contexts for generating a dataset according to Assumption 1. The following holds for the ICL risk $\mathcal{R}_N(\cdot)$ defined in (2):*

*1. The minimizer of $\mathcal{R}_N(\cdot)$ is unique and given by*

$$\boldsymbol{\Gamma}_N^* := \left( \frac{\mathtt{tr}(\mathbf{H}) + \sigma^2/\psi^2}{N} \mathbf{I} + \frac{N+1}{N} \mathbf{H} \right)^{-1}. \tag{3}$$

*2. The minimum ICL risk is given by*

$$\min_{\boldsymbol{\Gamma}} \mathcal{R}_N(\boldsymbol{\Gamma}) = \mathcal{R}_N(\boldsymbol{\Gamma}_N^*) = \sigma^2 + \psi^2 \mathtt{tr}\left( \boldsymbol{\Gamma}_N^* \mathbf{H} \left( \frac{\mathtt{tr}(\mathbf{H}) + \sigma^2/\psi^2}{N} \mathbf{I} + \frac{1}{N} \mathbf{H} \right) \right).$$

*3. The excess ICL risk, denoted by $\Delta_N(\cdot)$, is given by*

$$\Delta_N(\boldsymbol{\Gamma}) := \mathcal{R}_N(\boldsymbol{\Gamma}) - \min_{\boldsymbol{\Gamma}} \mathcal{R}_N(\boldsymbol{\Gamma}) = \left\langle \mathbf{H}, \ (\boldsymbol{\Gamma} - \boldsymbol{\Gamma}_N^*) \tilde{\mathbf{H}}_N (\boldsymbol{\Gamma} - \boldsymbol{\Gamma}_N^*)^\top \right\rangle,$$

*where*

$$\tilde{\mathbf{H}}_N := \mathbb{E}\left( \frac{1}{N} \mathbf{X}^\top \mathbf{y} \right) \left( \frac{1}{N} \mathbf{X}^\top \mathbf{y} \right)^\top = \psi^2 \mathbf{H} \left( \frac{\mathtt{tr}(\mathbf{H}) + \sigma^2/\psi^2}{N} \mathbf{I} + \frac{N+1}{N} \mathbf{H} \right). \tag{4}$$

*For simplicity, we may drop the subscript $N$ in $\boldsymbol{\Gamma}_N^*$ and $\tilde{\mathbf{H}}_N$ without causing ambiguity.*

When the size of the dataset $N \to \infty$, we have $\boldsymbol{\Gamma}_N^* \to \mathbf{H}^{-1}$ according to Theorem 3.1. Then for a fresh regression problem with task parameter $\boldsymbol{\beta}$, the attention model (1), after seeing prompt $(\mathbf{X}, \mathbf{y}, \mathbf{x})$ of infinite length, will perform a Newton step on the context $(\mathbf{X}, \mathbf{y})$ and then use the result to make a linear prediction for covariate $\mathbf{x}$. Since the context length is infinite, the output of a Newton step precisely recovers the task parameter $\boldsymbol{\beta}$, which minimizes the prediction error. Thus the attention model (1), with a fixed parameter $\boldsymbol{\Gamma}_\infty^*$, achieves *consistent* in-context learning (Zhang et al., 2023a). When $N$ is finite, (3) is a regularized Hessian inverse, so (1) performs a regularized Newton step in-context — the regression risk of this algorithm will be discussed in depth later in Section 5.

Theorem 3.1 suggests that the ICL risk parameterized by $\boldsymbol{\Gamma}$ is convex and the optimal parameter is unique. However, since the population distribution of the dataset is unknown (because $\psi^2$, $\sigma^2$, and $\mathbf{H}$ are unknown) and the parameter (a $d \times d$ matrix) is high-dimensional, it is not immediately clear how many independent tasks are needed to learn the optimal parameter. We will address this issue in the next section.

## 4 THE TASK COMPLEXITY OF PRETRAINING AN ATTENTION MODEL

**Pretraining dataset.** During the pretraining stage, we are provided with a pretraining dataset that consists of $N + 1$ independent data from each of the $T$ independent regression tasks. Specifically, the pretraining dataset is given by

$$\mathbf{X}_t \in \mathbb{R}^{N \times d}, \quad \mathbf{y}_t \in \mathbb{R}^N, \quad \mathbf{x}_t \in \mathbb{R}^d, \quad y_t \in \mathbb{R}, \qquad t = 1, \dots, T, \tag{5}$$

where each tuple $(\mathbf{X}_t, \mathbf{y}_t, \mathbf{x}_t, y_t)$ is independently generated according to Assumption 1 with $N$ being the number of contexts. We assume $N$ is fixed during pretraining to simplify the analysis.

**Pretraining rule.** Based on the pretraining dataset (5), we pretrain the matrix parameter $\boldsymbol{\Gamma}_T$ by *stochastic gradient descent*. That is, from an initialization $\boldsymbol{\Gamma}_0$, e.g., $\boldsymbol{\Gamma}_0 = \mathbf{0}$, we iteratively generate $(\boldsymbol{\Gamma}_t)_{t=1}^T$ by

$$\boldsymbol{\Gamma}_t = \boldsymbol{\Gamma}_{t-1} - \frac{\gamma_t}{2} \nabla \left( f(\mathbf{X}_t, \mathbf{y}_t, \mathbf{x}_t; \boldsymbol{\Gamma}_{t-1}) - y_t \right)^2, \quad t = 1, \dots, T, \tag{6}$$

where $(\mathbf{X}_t, \mathbf{y}_t, \mathbf{x}_t, y_t)_{t=1}^T$ is the pretraining dataset (5), $f$ is the attention model (1), and $(\gamma_t)_{t=1}^T$ is a geometrically decaying stepsize schedule (Ge et al., 2019; Wu et al., 2022), i.e.,

$$\gamma_t = \frac{\gamma_0}{2^\ell}, \quad \ell = \lfloor t/\log(T) \rfloor, \quad t = 1, \dots, T. \tag{7}$$

Here, $\gamma_0 > 0$ is an initial stepsize that is a hyperparameter. The output of SGD is the last iterate, i.e., $\mathbf{\Gamma}_T$.

Our main result in this section is the following ICL risk bound achieved by pretraining with $T$ independent tasks. The proof is deferred to Appendix D.

**Theorem 4.1** (Task complexity for pretraining). *Fix $N \geq 0$. Let $\mathbf{\Gamma}_T$ be generated by (6) with pretraining dataset (5) and stepsize schedule (7). Suppose that the initialization $\mathbf{\Gamma}_0$ commutes with $\mathbf{H}$ and $\gamma_0 \leq 1/\big(c\mathtt{tr}(\mathbf{H})\mathtt{tr}(\tilde{\mathbf{H}}_N)\big)$, where $c > 1$ is an absolute constant and $\tilde{\mathbf{H}}_N$ is defined in (4) in Theorem 3.1. Then we have*

$$\mathbb{E}\Delta_N(\mathbf{\Gamma}_T) \lesssim \left\langle \mathbf{H}\tilde{\mathbf{H}}_N, \left( \prod_{t=1}^T \big(\mathbf{I} - \gamma_t \mathbf{H}\tilde{\mathbf{H}}_N\big)(\mathbf{\Gamma}_0 - \mathbf{\Gamma}_N^*) \right)^2 \right\rangle$$
$$+ \left( \psi^2 \mathtt{tr}(\mathbf{H}) + \sigma^2 + \left\langle \mathbf{H}\tilde{\mathbf{H}}_N, \ (\mathbf{\Gamma}_0 - \mathbf{\Gamma}_N^*)^2 \right\rangle \right) \frac{D_{\mathtt{eff}}}{T_{\mathtt{eff}}},$$

*where the effective number of tasks and effective dimension are given by*

$$T_{\mathtt{eff}} := \frac{T}{\log(T)}, \quad D_{\mathtt{eff}} := \sum_i \sum_j \min\big\{1, \ T_{\mathtt{eff}}^2 \gamma_0^2 \lambda_i^2 \tilde{\lambda}_j^2\big\}, \tag{8}$$

*respectively, and $\big(\lambda_i\big)_{i \geq 1}$ and $\big(\tilde{\lambda}_j\big)_{j \geq 1}$ are the eigenvalues of $\mathbf{H}$ and $\tilde{\mathbf{H}}_N$ that satisfy*

$$\tilde{\lambda}_j := \psi^2 \lambda_j \left( \frac{\mathtt{tr}(\mathbf{H}) + \sigma^2/\psi^2}{N} + \frac{N+1}{N}\lambda_j \right), \quad j \geq 1.$$

*In particular, when $\mathbf{\Gamma}_0 = 0$, we have*

$$\mathbb{E}\Delta_N(\mathbf{\Gamma}_T) \lesssim \left\langle \mathbf{H}\tilde{\mathbf{H}}_N, \left( \prod_{t=1}^T \big(\mathbf{I} - \gamma_t \mathbf{H}\tilde{\mathbf{H}}_N\big)\mathbf{\Gamma}_N^* \right)^2 \right\rangle + \big( \psi^2 \mathtt{tr}(\mathbf{H}) + \sigma^2 \big) \frac{D_{\mathtt{eff}}}{T_{\mathtt{eff}}}. \tag{9}$$

Theorem 4.1 provides a statistical ICL risk bound for pretraining with $T$ tasks, which suggests that the optimal matrix parameter $\mathbf{\Gamma}_N^*$ (see (3)) can be recovered by SGD pretraining if $T$ is large enough. Focusing on (9) in Theorem 4.1, the first term is the error of directly running gradient descent on the population ICL risk (see Theorem 3.1), which decreases at an exponential rate. However, seeing only finite pretraining tasks, the population ICL risk is directly minimizable by the pretraining rule, and the second term in (9) accounts for the variance caused by pretraining with data from $T$ independent tasks rather than an infinite number. The second term is small when the effective dimension is small compared to the effective number of tasks (see their definitions in (8)). We remark that the initial stepsize $\gamma_0$ induces a trade-off between the two terms, where a larger initial stepsize reduces the first term but increases the second term and vice versa.

We highlight that the bounds in Theorem 4.1 do not explicitly depend on the ambient dimensionality $d^2$, allowing efficient pretraining even with a large number of model parameters. Specifically, our bounds (e.g., (9)) are functions of the effective dimension (8). In the worst case, for example, when $\mathbf{H} = \mathbf{I}$ and $T$ is larger, we have $D_{\mathtt{eff}} = d^2$ so that the excess risk bound is $\tilde{\mathcal{O}}(d^2/T)$. However, the effective dimension $D_{\mathtt{eff}}$ is always no larger, and can even be much smaller, than $d^2$ depending on the spectrum of the data covariance. In contrast, the pretraining bound in (Bai et al., 2023) is based on uniform convergence analysis (see their Theorem 21) and explicitly depends on the number of model parameters, hence is worse than ours.

To further demonstrate the power of our pretraining bounds, we present three examples in the following corollary, which illustrate how pretraining with limited tasks minimizes ICL risk. The proof is deferred to Appendix D.9.

**Corollary 4.2** (Large stepsize). *Under the setup of Theorem 4.1, additionally assume that $\mathbf{\Gamma}_0 = 0$, $\sigma^2 \asymp 1$, $\psi^2 \asymp 1$, $\mathtt{tr}(\mathbf{H}) \asymp 1$, and choose stepsize $\gamma_0 \asymp 1/\big(\mathtt{tr}(\mathbf{H})\mathtt{tr}(\tilde{\mathbf{H}})\big) \asymp 1/\mathtt{tr}(\tilde{\mathbf{H}})$.*

1. **The uniform spectrum.** If $\lambda_i = 1/s$ for $1 \le i \le s$ and $\lambda_i = 0$ for $i > s$, where $s$ and $N$ satisfy $N \le s \le d$, then

$$
\mathbb{E}\Delta_N(\mathbf{\Gamma}_T) \lesssim
\begin{cases}
\dfrac{N}{s} + \dfrac{T_{\mathtt{eff}}}{s^2} & T_{\mathtt{eff}} \le s^2, \\[2mm]
\dfrac{s^2}{T_{\mathtt{eff}}} & T_{\mathtt{eff}} > s^2.
\end{cases}
$$

2. **The polynomial spectrum.** If $\lambda_i = i^{-a}$ for $a > 1$ and $N^3 = o(T_{\mathtt{eff}})$, then

$$
\mathbb{E}\Delta(\mathbf{\Gamma}_T) \lesssim T_{\mathtt{eff}}^{\frac{1}{a}-1}\Big(1 + N^{-\frac{1}{a}}\log(T_{\mathtt{eff}}) + T_{\mathtt{eff}}^{-\frac{1}{2a}}N^{2-\frac{1}{2a}}\Big).
$$

3. **The exponential spectrum.** If $\lambda_i = 2^{-i}$ and $N^3 \le o(T_{\mathtt{eff}})$, then

$$
\mathbb{E}\Delta(\mathbf{\Gamma}_T) \lesssim \frac{N^2 + \log^2(T_{\mathtt{eff}})}{T_{\mathtt{eff}}}.
$$

To summarize this section, we show that the single-layer linear attention model can be effectively pretrained with a small number of independent tasks. We note that our statistical task complexity results are under the one-step GD parameterization, where we have a convex (but high-dimensional) learning problem. Under the orginal attention parameterization (see Appendix B), the learning problem is non-convex, which adds an extra layer of complexity from non-convex optimization. We leave for future work extending our statistical task complexity results to the original attention parameterization. Finally, we also empirically verify our theory both numerically and with a three-layer transformer in Appendix A. Nevertheless, it is still unclear whether or not the pretrained model achieves good ICL performance. This will be our focus in the next section.

## 5 THE IN-CONTEXT LEARNING OF THE PRETRAINED ATTENTION MODEL

In this section, we examine the ICL performance of a pretrained single-layer linear attention model. We have already shown the model can be efficiently pretrained. So in this part, we will focus on the model (1) equipped with the optimal parameter ($\mathbf{\Gamma}_N^*$ in (3)), to simplify our discussions. Our results in this section can be extended to imperfectly pretrained parameters ($\mathbf{\Gamma}_T$) by applying an additional triangle inequality. All proofs for results in this section can be found in Appendix E.

**The attention estimator.** According to (1) and (3), the optimally pretrained attention model corresponds to the following linear estimator:

$$
f(\mathbf{X}, \mathbf{y}, \mathbf{x}) := \left\langle \left( \frac{N+1}{N}\mathbf{H} + \frac{\mathtt{tr}(\mathbf{H}) + \sigma^2/\psi^2}{N}\mathbf{I} \right)^{-1} \frac{\mathbf{X}^\top \mathbf{y}}{\dim(\mathbf{y})}, \ \mathbf{x} \right\rangle. \tag{10}
$$

**Average regression risk.** Given a task-specific dataset $(\mathbf{X}, \mathbf{y}, \mathbf{x}, y)$ generated by Assumption 1, let $g(\mathbf{X}, \mathbf{y}, \mathbf{x})$ be an estimator of $y$. We measure the *average linear regression risk* of $g$ by

$$
\mathcal{L}(g; \mathbf{X}) := \mathbb{E}\big[\big(g(\mathbf{X}, \mathbf{y}, \mathbf{x}) - y\big)^2 \mid \mathbf{X}\big], \tag{11}
$$

where the expectation is taken with respect to $\mathbf{y}$, $\mathbf{x}$, and $y$, and is conditioned on $\mathbf{X}$.

**The Bayes optimal estimator.** It is well known that the optimal estimator for linear regression with a Gaussian prior is an optimally tuned ridge regression estimator (see for example Bishop & Nasrabadi, 2006, Section 3.3). This is formally justified by the following proposition.

**Proposition 5.1** (Optimally tuned ridge regression)**.** *Given a task-specific dataset $(\mathbf{X}, \mathbf{y}, \mathbf{x}, y)$ generated by Assumption 1, the following estimator minimizes the average risk* (11)*:*

$$
\begin{aligned}
h(\mathbf{X}, \mathbf{y}, \mathbf{x}) &:= \left\langle \left(\mathbf{X}^\top \mathbf{X} + \sigma^2/\psi^2 \mathbf{I}\right)^{-1}\mathbf{X}^\top \mathbf{y}, \ \mathbf{x} \right\rangle \\
&= \left\langle \left( \frac{1}{\dim(\mathbf{y})}\mathbf{X}^\top \mathbf{X} + \frac{\sigma^2/\psi^2}{\dim(\mathbf{y})}\mathbf{I} \right)^{-1} \frac{\mathbf{X}^\top \mathbf{y}}{\dim(\mathbf{y})}, \ \mathbf{x} \right\rangle.
\end{aligned} \tag{12}
$$

It is clear that the optimal estimator (12) corresponds to a ridge regression estimator with regularization parameter $\sigma^2/\psi^2/\dim(\mathbf{y})$.

Based on the analysis of ridge regression in (Tsigler & Bartlett, 2023), we can obtain the following bound on the average regression risk induced by the optimally tuned ridge regression.

**Corollary 5.2** (Average risk of ridge regression, corollary of (Tsigler & Bartlett, 2023)). *Consider the average risk defined in* (11). *Assume that the signal-to-noise ratio is upper bounded, i.e.,* $\psi^2 \mathtt{tr}(\mathbf{H}) \lesssim \sigma^2$. *Then for the optimally tuned ridge regression estimator* (12), *with probability at least* $1 - e^{-\Omega(M)}$ *over the randomness in* $\mathbf{X}$, *it holds that*

$$\mathcal{L}(h; \mathbf{X}) - \sigma^2 \asymp \psi^2 \sum_i \min\{\mu_M, \lambda_i\}, \quad \text{where } \mu_M \asymp \frac{\sigma^2/\psi^2}{M},$$

*where* $M = \dim(\mathbf{y})$ *refers to the number of independent data in* $(\mathbf{X}, \mathbf{y})$.

We remark that the attention estimator (10) is *not* the Bayes optimal estimator (12). However, we will show that the average risk induced by the attention estimator (10) can be close to that of the Bayes optimal estimator (12) in suitable regimes. In this way, we can view the attention estimator (10) as a good "statistical shortcut" to the Bayes optimal estimator (12), thus achiving good ICL performance.

Based on Theorem 3.1, we have the following bounds on the average risk for the attention model.

**Theorem 5.3** (Average risk of the pretrained attention model). *Consider the average risk defined in* (11). *Assume that the signal-to-noise ratio is upper bounded, i.e.,* $\psi^2 \mathtt{tr}(\mathbf{H}) \lesssim \sigma^2$. *Then for the attention estimator* (10), *we have*

$$\mathbb{E}\mathcal{L}(f; \mathbf{X}) - \sigma^2 \asymp \psi^2 \sum_i \min\{\mu_M, \lambda_i\} + \psi^2(\mu_M - \mu_N)^2 \sum_i \min\left\{\frac{\lambda_i}{\mu_N^2}, \frac{1}{\lambda_i}\right\} \min\left\{\frac{\lambda_i}{\mu_M}, 1\right\},$$

*where* $\mu_M \asymp \sigma^2/(\psi^2 M)$, *and* $\mu_N \asymp \sigma^2/(\psi^2 N)$.

Theorem 5.3 provides an average risk bound for the optimally pretrained attention model. The first term in the bound in Theorem 5.3 matches the bound in Corollary 5.2. When the context length in pretraining and inference is close, i.e., when $M \asymp N$, the second term in the bound is higher-order, so the average risk bound of the attention model matches that of the optimally tuned ridge regression. In this case, the pretrained attention model achieves optimal ICL.

When $M$ and $N$ are not close, the attention model induces a larger average risk compared to ridge regression. We provide the following three examples to illustrate the gap in their performance.

**Corollary 5.4** (Examples). *Under the setups of Corollary 5.2 and Theorem 5.3, additionally assume that* $\sigma^2 \asymp 1$, $\psi^2 \asymp 1$, $\mathtt{tr}(\mathbf{H}) \asymp 1$, *and* $M < N/c$ *for some constant* $c > 1$.

1. ***The uniform spectrum.*** *When* $\lambda_i = 1/s$ *for* $i \leq s$ *and* $\lambda_i = 0$ *for* $i > s$, *we have*

$$\mathcal{L}(h; \mathbf{X}) - \sigma^2 \asymp \min\left\{1, \frac{s}{M}\right\}, \qquad \text{with probability at least } 1 - e^{-\Omega(M)};$$

$$\mathbb{E}\mathcal{L}(f; \mathbf{X}) - \sigma^2 \asymp \min\left\{1, \frac{s}{M}\right\}, \qquad \text{if } s < M \text{ or } s > N^2/M.$$

2. ***The polynomial spectrum.*** *When* $\lambda_i = i^{-a}$ *for* $a > 1$, *we have*

$$\mathcal{L}(h; \mathbf{X}) - \sigma^2 \asymp M^{\frac{1}{a}-1}, \qquad \text{with probability at least } 1 - e^{-\Omega(M)};$$

$$\mathbb{E}\mathcal{L}(f; \mathbf{X}) - \sigma^2 \asymp N^{\frac{1}{a}} M^{-1}.$$

3. ***The exponential spectrum.*** *When* $\lambda_i = 2^{-i}$, *we have*

$$\mathcal{L}(h; \mathbf{X}) - \sigma^2 \asymp \frac{\log M}{M}, \qquad \text{with probability at least } 1 - e^{-\Omega(M)};$$

$$\mathbb{E}\mathcal{L}(f; \mathbf{X}) - \sigma^2 \asymp \frac{\log N}{M}.$$

To conclude this section, we show that the pretrained model attains Bayes optimal ICL when the inference context length is close to the pretraining context length. However, when the context length is very different in pretraining and in inference, the ICL of the pretrained single-layer linear attention might be suboptimal.

## 6 TECHNIQUE OVERVIEW

In this section, we explain the proof of Theorem 4.1. Our techniques are motivated by the operator method developed for analyzing 4-th order tensors (i.e., linear operators on matrices) arising in linear regression (Bach & Moulines, 2013; Dieuleveut et al., 2017; Jain et al., 2018; 2017; Zou et al., 2021; Wu et al., 2022) and ReLU regression (Wu et al., 2023). However, we need to deal with 8-th order tensors that require two new tools, namely, *diagonalization* and *operator polynomials*, which will be discussed later in this section. For simplicity, we write $\mathbf{\Gamma}_N^*$ and $\tilde{\mathbf{H}}_N$ as $\mathbf{\Gamma}^*$ and $\tilde{\mathbf{H}}$, respectively.

We start with evaluating (6) and get

$$\mathbf{\Gamma}_t = \mathbf{\Gamma}_{t-1} - \gamma_t \mathbf{x}_t \mathbf{x}_t^\top \left(\mathbf{\Gamma}_{t-1} - \mathbf{\Gamma}^*\right)\left(\frac{1}{N}\mathbf{X}_t^\top \mathbf{y}_t\right)\left(\frac{1}{N}\mathbf{X}_t^\top \mathbf{y}_t\right)^\top - \gamma_t \mathbf{\Xi}_t, \quad t = 1, \dots, T,$$

where $\mathbf{\Xi}_t$ is a *zero mean* random matrix given by

$$\mathbf{\Xi}_t := \mathbf{x}_t \mathbf{x}_t^\top \mathbf{\Gamma}^* \left(\frac{1}{N}\mathbf{X}_t^\top \mathbf{y}_t\right)\left(\frac{1}{N}\mathbf{X}_t^\top \mathbf{y}_t\right)^\top - y_t \mathbf{x}_t \left(\frac{1}{N}\mathbf{X}_t^\top \mathbf{y}_t\right)^\top.$$

Define a sequence of (random) linear maps on matrices,

$$\forall \mathbf{A} \in \mathbb{R}^{d \times d}, \quad \mathscr{P}_t \circ \mathbf{A} := \mathbf{A} - \gamma_t \mathbf{x}_t \mathbf{x}_t^\top \mathbf{A}\left(\frac{1}{N}\mathbf{X}_t^\top \mathbf{y}_t\right)\left(\frac{1}{N}\mathbf{X}_t^\top \mathbf{y}_t\right)^\top, \quad 1 \le t \le T.$$

Then we can re-write the recursion as

$$\mathbf{\Gamma}_t - \mathbf{\Gamma}^* = \mathscr{P}_t \circ (\mathbf{\Gamma}_{t-1} - \mathbf{\Gamma}^*) - \gamma_t \mathbf{\Xi}_t, \quad t = 1, \dots, T.$$

The (random) linear recursion allows us to track $\mathbf{\Gamma}_T$, which serves as the basis of the operator method. From now on, we will heavily use tensor notations. We refer the readers to Appendix D.1 for a brief overview of tensors (especially PSD operators).

**Bias-variance decomposition.** Solving the recursion of $\mathbf{\Gamma}_t$ yields

$$\mathbf{\Gamma}_T - \mathbf{\Gamma}^* = \prod_{t=1}^T \mathscr{P}_t \circ (\mathbf{\Gamma}_0 - \mathbf{\Gamma}^*) - \sum_{t=1}^T \gamma_t \prod_{k=t+1}^T \mathscr{P}_k \circ \mathbf{\Xi}_t.$$

Taking outer product and expectation, we have

$$\mathcal{A}_T := \mathbb{E}(\mathbf{\Gamma}_T - \mathbf{\Gamma}^*)^{\otimes 2} = \mathbb{E}\left(\prod_{t=1}^T \mathscr{P}_t \circ (\mathbf{\Gamma}_0 - \mathbf{\Gamma}^*) - \sum_{t=1}^T \gamma_t \prod_{k=t+1}^T \mathscr{P}_k \circ \mathbf{\Xi}_t\right)^{\otimes 2}$$

$$\preceq 2\,\mathbb{E}\underbrace{\left(\prod_{t=1}^T \mathscr{P}_t \circ (\mathbf{\Gamma}_T - \mathbf{\Gamma}^*)\right)^{\otimes 2}}_{=: \mathcal{B}_T} + 2\,\mathbb{E}\underbrace{\left(\sum_{t=1}^T \gamma_t \prod_{k=t+1}^T \mathscr{P}_k \circ \mathbf{\Xi}_t\right)^{\otimes 2}}_{=: \mathcal{C}_T},$$

where $\mathcal{A}_T$, $\mathcal{B}_T$, and $\mathcal{C}_T$ are all PSD operators on matrices (i.e., 4-th order tensors). Then we can decompose the ICL risk (see Theorem 3.1) into a bias error and a variance error:

$$\mathbb{E}\Delta_N(\mathbf{\Gamma}_T) := \left\langle \mathbf{H},\, \mathcal{A}_T \circ \tilde{\mathbf{H}}\right\rangle \le 2\left\langle \mathbf{H},\, \mathcal{B}_T \circ \tilde{\mathbf{H}}\right\rangle + 2\left\langle \mathbf{H},\, \mathcal{C}_T \circ \tilde{\mathbf{H}}\right\rangle.$$

In what follows, we focus on explaining the analysis of the variance error $\left\langle \mathbf{H},\, \mathcal{C}_T \circ \tilde{\mathbf{H}}\right\rangle$.

**Operator recursion.** The variance operator $\mathcal{C}_T$ can be equivalently defined through the following operator recursion (see Appendix D.2 for more details):

$$\mathcal{C}_0 = \mathbf{0} \otimes \mathbf{0}, \quad \mathcal{C}_t = \mathscr{S}_t \circ \mathcal{C}_{t-1} + \gamma_t^2 \mathcal{N}, \quad t = 1, \dots, T, \tag{13}$$

where $\mathcal{N} := \mathbb{E}[\mathbf{\Xi}^{\otimes 2}]$ and $\mathscr{S}_t$ is a linear map on operators (i.e., an 8-th order tensor) given by: for any $\mathcal{O} \in (\mathbb{R}^{d \times d})^{\otimes 2}$,

$$\mathscr{S}_t \circ \mathcal{O} := \mathcal{O} - \gamma_t\left((\mathbf{H} \otimes \mathbf{I}) \circ \mathcal{O} \circ (\tilde{\mathbf{H}} \otimes \mathbf{I}) + (\mathbf{I} \otimes \mathbf{H}) \circ \mathcal{O} \circ (\mathbf{I} \otimes \tilde{\mathbf{H}})\right) + \gamma_t^2 \mathcal{M} \circ \mathcal{O} \circ \mathcal{L},$$

with $\mathcal{M}$, $\mathcal{L}$ being given by

$$\mathcal{M} := \mathbb{E}\big(\mathbf{x}\mathbf{x}^\top\big)^{\otimes 2}, \quad \mathcal{L} := \mathbb{E}\Bigg(\bigg(\frac{1}{N}\mathbf{X}^\top\mathbf{y}\bigg)\bigg(\frac{1}{N}\mathbf{X}^\top\mathbf{y}\bigg)^\top\Bigg)^{\otimes 2}.$$

Appendix D.3 includes several bounds about these operators; among them the following is crucial:

for any PSD operator $\mathcal{O}$, $\mathcal{M} \circ \mathcal{O} \circ \mathcal{L} \preceq c\langle\mathbf{H}, \mathcal{O} \circ \tilde{\mathbf{H}}\rangle\mathcal{S}^{(1)}$, where $\mathcal{S}^{(1)} := \langle\tilde{\mathbf{H}}, \cdot\rangle\mathbf{H}$,

where $c > 1$ is an absolute constant.

**Key idea 1: diagonalization.** The operator recursion (13) involves 8-th order tensors $\mathscr{S}_t$ that are hard to compute. A critical observation is that the variance bound only depends on the results of $\mathcal{C}_T$ applied on *diagonal matrices* (assuming that $\mathbf{H}$ is diagonal, which can be made without loss of generality). More importantly, when restricting the relevant operators to diagonal matrices (instead of all matrices), the 8-th order tensors $\mathscr{S}_t$ can be bounded by simpler 8-th order tensors $\mathscr{G}_t$ plus diagonal operators. Specifically, based on (13), we can show that (see Appendix D.4)

$$\mathring{\mathcal{C}}_t \preceq \mathscr{G}_t \circ \mathring{\mathcal{C}}_{t-1} + c\gamma_t^2\langle\mathbf{H}, \mathring{\mathcal{C}}_{t-1} \circ \tilde{\mathbf{H}}\rangle \cdot \mathcal{S}^{(1)} + c\gamma_t^2(\psi^2\mathtt{tr}(\mathbf{H}) + \sigma^2)\mathcal{S}^{(1)}, \tag{14}$$

where $\mathring{\mathcal{C}}_t$ refers to $\mathcal{C}_t$ restricted to diagonal matrices and $\mathscr{G}_t$ is a linear map on operators given by:

$$\mathscr{G}_t \circ \mathcal{O} := \mathcal{O} - \gamma_t\Big((\mathbf{H} \otimes \mathbf{I}) \circ \mathcal{O} \circ (\tilde{\mathbf{H}} \otimes \mathbf{I}) + (\mathbf{I} \otimes \mathbf{H}) \circ \mathcal{O} \circ (\mathbf{I} \otimes \tilde{\mathbf{H}})\Big) + \gamma_t^2\mathbf{H}^{\otimes 2} \circ \mathcal{O} \circ \tilde{\mathbf{H}}^{\otimes 2}.$$

We remark that Wu et al. (2023) has used the diagonalization idea with matrices for dealing with non-commutable matrices. In comparison, here we use the diagonalization idea with operators for dealing with high-order tensors.

**Key idea 2: operator polynomials.** To solve the operator recursion in (14), we need to know how the 8-th order tensor $\mathscr{G}_t$ interacts with operator $\mathcal{S}^{(1)}$. To this end, we introduce a powerful tool called *operator polynomials*. Specifically, we define operator monomials and their "multiplication" as follows:

$$\mathcal{S}^{(i)} := \langle\tilde{\mathbf{H}}^i, \cdot\rangle\mathbf{H}^i, \quad \mathcal{S}^{(i)} \bullet \mathcal{S}^{(j)} := \mathcal{S}^{(i+j)}, \quad i, j \in \mathbb{N}.$$

One can verify that the multiplication "$\bullet$" distributes with the usual addition "+", therefore we can define polynomials of operators. We prove the following key equations that connect operator polynomials with how the 8-th order tensor $\mathscr{G}_t$ interacts with operator $\mathcal{S}^{(1)}$ (see Appendix D.5):

$$\mathscr{G}_t \circ \mathcal{S}^{(1)} = \big(\mathcal{S}^{(0)} - \gamma_t\mathcal{S}^{(1)}\big)^{\bullet 2} \bullet \mathcal{S}^{(1)}, \quad \Bigg(\prod_{k=1}^t \mathscr{G}_k\Bigg) \circ \mathcal{S}^{(1)} = \prod_{k=1}^t \big(\mathcal{S}^{(0)} - \gamma_k\mathcal{S}^{(1)}\big)^{\bullet 2} \bullet \mathcal{S}^{(1)}.$$

In addition, we note that the operator polynomials are all diagonal operators that contain only $d^2$ degrees of freedom (unlike general operators that contain $d^4$ degrees of freedom), thus we can compute them via relatively simple algebraic rules (see Appendix D.5).

**Variance and bias error.** Up to now, we have introduced *diagonalization* to simplify the operator recursion and *operator polynomials* to compute the simplified operator recursion. The remaining efforts are to analyze the variance error following the methods introduced by Zou et al. (2021); Wu et al. (2022) (see Appendix D.6). The analysis of the bias error is more involved; it is presented in Appendix D.7.

## 7 CONCLUSION

This paper studies the in-context learning of a single-layer linear attention model for linear regression with a Gaussian prior. We prove a statistical task complexity bound for the pretraining of the attention model, where we develop new tools for operator methods. In addition, we compare the average linear regression risk obtained by a pretrained attention model with that obtained by an optimally tuned ridge regression, which clarifies the effectiveness of in-context learning. Our theories complement experimental results in prior works.

ACKNOWLEDGEMENT

We thank the anonymous reviewers and area chairs for their helpful comments. DZ acknowledges the support from NSFC 62306252. ZC and QG are supported in part by the NSF grants IIS-1906169, IIS-2008981, and the Sloan Research Fellowship. VB is partially supported by National Science Foundation Awards 2244899 and 2333887, the ONR award N000142312737, the Ministry of Trade, Industry and Energy (MOTIE) and Korea Institute for Advancement of Technology (KIAT) through the International Cooperative R&D program. JW and PB are supported in part by NSF grants DMS-2023505 and DMS-2031883 and Simons Foundation award 814639. The views and conclusions contained in this paper are those of the authors and should not be interpreted as representing any funding agencies.

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

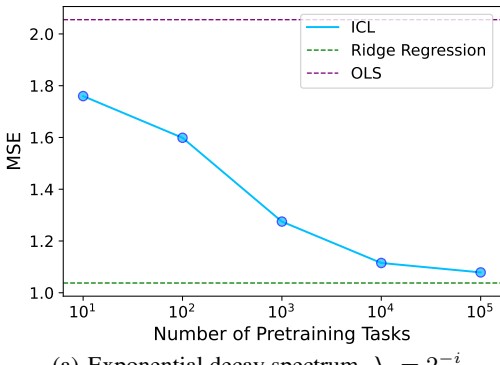
(a) Exponential decay spectrum, $\lambda_i = 2^{-i}$

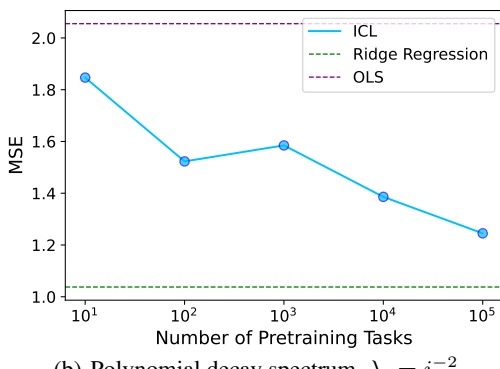
(b) Polynomial decay spectrum, $\lambda_i = i^{-2}$

Figure 1: Task complexity of ICL (of the one-step GD model), ridge regression, and OLS. The context length is $M = N = 200$. The ambient dimension is $d = 100$. We observe that as the number of pretraining tasks increases, one-step GD achieves smaller MSE and becomes closer to the Bayes algorithm, ridge regression. This is consistent with our theory.

# A  EXPRIMENTS

In this section, we conduct experiments on the one-step GD model (1) and a three-layer transformer.

## A.1  THE ONE-STEP GD MODEL

**Data generation.** We follow the generation process outlined in Assumption 1. Specifically, we sample $(\mathbf{x}_n, y_n)_{n=1}^{N+1}$ as independent copies of $(\mathbf{x}, y)$, where

$$\mathbf{x} \sim \mathcal{N}(0, \mathbf{H}), \quad y \sim \mathcal{N}(\boldsymbol{\beta}^\top \mathbf{x}, \sigma^2), \quad \boldsymbol{\beta} \sim \mathcal{N}(0, \psi^2 \mathbf{I}_d).$$

We treat the first $N$ data points $(\mathbf{x}_n, y_n)_{n=1}^{N}$ as the context examples, $\mathbf{x}_{N+1}$ as the covariate, and $y_{N+1}$ as the response.

**Base experiment setup.** We configure the base experiment with the following parameters:

$$d = 100, \ N = 2d, \ \sigma = 1, \ \psi = 1, \ \mathbf{H} = \mathtt{diag}(2^{-1}, 2^{-2}, \dots, 2^{-d}).$$

We sample a fresh sequence $(\mathbf{x}_n, y_n)_{n=1}^{N+1}$ for each task. We train the ICL model using online SGD (see (6)) with a geometrically decaying stepsize schedule defined in (7). We run online SGD for $10^5$ steps. The default initial learning rate is set as $0.1$. For evaluation, we consider in-context sample size $M = N = 200$ and compare against benchmark algorithms such as optimally tuned ridge regression (Theorem 5.1) and Ordinary Least Square (OLS). We conduct a series of experiments by varying parts of this base experiment setup, that is, the experimental setups are identical to this base experiment setup unless noted otherwise.

**The effect of the number of pretraining tasks.** To examine the pretraining task complexity, we vary the number of pretraining tasks in the base setup in the range $[10^1, 10^2, 10^3, 10^4, 10^5]$. In addition to the exponentially decaying spectrum considered in the base setup, we consider a polynomially decaying spectrum with $\lambda_i = i^{-2}$. For different spectrums $\lambda_i = i^{-2}$ and $\lambda_i = 2^{-i}$, the initial learning rates were optimally tuned from the set $\{0.005, 0.01, 0.05, 0.1, 0.5\}$, resulting in an optimal rate of $0.1$ for both. Results are presented in Figure 1. We observe that the ICL error decreases as the number of pretraining tasks increases.

**The effect of the ambient dimension.** To examine the pretraining task complexity, we vary the ambient dimension in the base setup in the range of $d \in \{10, 20, 50, 100\}$. We also consider a polynomial decay spectrum with $\lambda_i = i^{-2}$. Results are presented in Figure 2. We observe that the ICL performance is relatively unaffected by the ambient dimension $d$.

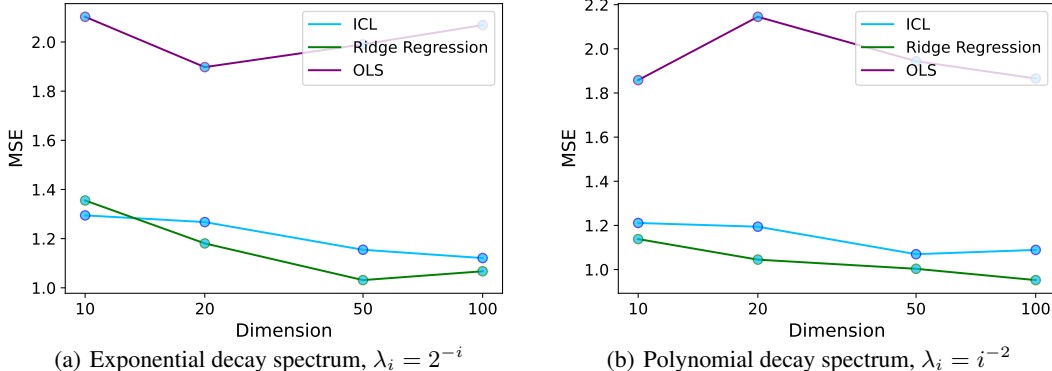

(a) Exponential decay spectrum, $\lambda_i = 2^{-i}$    (b) Polynomial decay spectrum, $\lambda_i = i^{-2}$

Figure 2: The effect of the ambient dimension for ICL (of one-step GD), ridge regression, and OLS. The context length is $M = N = 200$. The number of pretraining tasks is $10^5$ for ICL. We observe that when the spectrum of the data covariance $\mathbf{H}$ decays relatively fast, for example, $\lambda_i \sim 2^{-i}$ and $\lambda_i \sim i^{-2}$, the performances of the three considered algorithms are not sensitive to the ambient dimension. This is consistent with our theory.

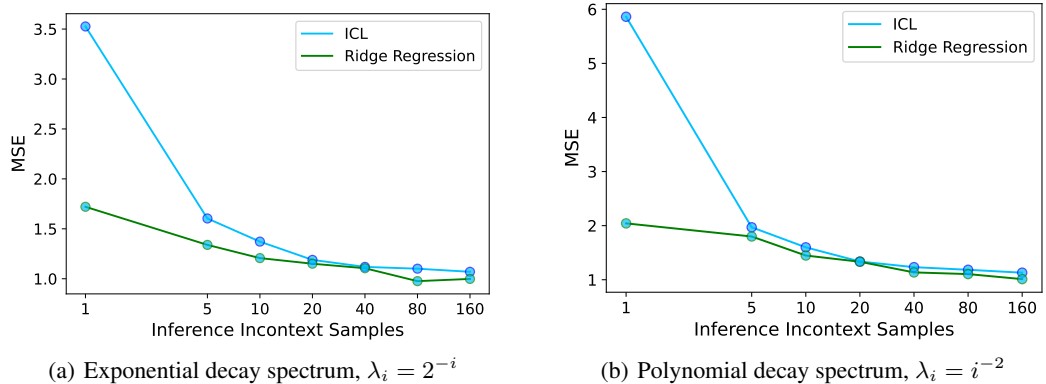

(a) Exponential decay spectrum, $\lambda_i = 2^{-i}$    (b) Polynomial decay spectrum, $\lambda_i = i^{-2}$

Figure 3: The effect of the number of context examples during inference for ICL (of one-step GD) and ridge regression. The number of context examples during pretraining is $N = 40$. The ambient dimension is $d = 20$. The MSE of OLS is significantly worse than ICL and ridge regression when $M \leq N = 20$, so we ignore OLS in this plot for a better visualization. We observe that the ICL achieves a similar MSE to ridge regression when $M$ is close to $N$. However, the gap becomes larger when $M$ is much smaller than $N$. This is consistent with our theory.

**The effect of the number of context examples during inference.** We modify the base experiment setup with $d = 20, N = 40$. We then examine the effect of the number of context examples during inference by varying $M$. Similarly, we also consider a polynomial decay spectrum with $\lambda_i = i^{-2}$. Results are presented in Figure 3. We observe that when $M$ is close to $N$, the number of context examples during pretraining, the ICL risk of one-step GD is close to that of optimally tuned ridge regression. However, the gap becomes larger when $M$ is much smaller than $N$. This is consistent with our theory.

**The effect of model misspecification.** The base experiment setup assumes well-specified data. We now investigate three misspecification scenarios:

1. Replacing the label generation process from $y \sim \mathcal{N}(\boldsymbol{\beta}^\top \mathbf{x}, \sigma^2)$ to $y \sim \boldsymbol{\beta}^\top \mathbf{x} + \text{uniform}[-c, c]$, where we set $c = \sqrt{3}$ to maintain the noise variance.

2. Replacing the label generation process from $y \sim \mathcal{N}(\boldsymbol{\beta}^\top \mathbf{x}, \sigma^2)$ to $y \sim \mathcal{N}(\text{sigmoid}(\boldsymbol{\beta}^\top \mathbf{x}), \sigma^2)$.

3. Replacing the label generation process from $y \sim \mathcal{N}(\boldsymbol{\beta}^\top \mathbf{x}, \sigma^2)$ to $y \sim \mathcal{N}((\boldsymbol{\beta}^\top \mathbf{x})^2, \sigma^2)$.

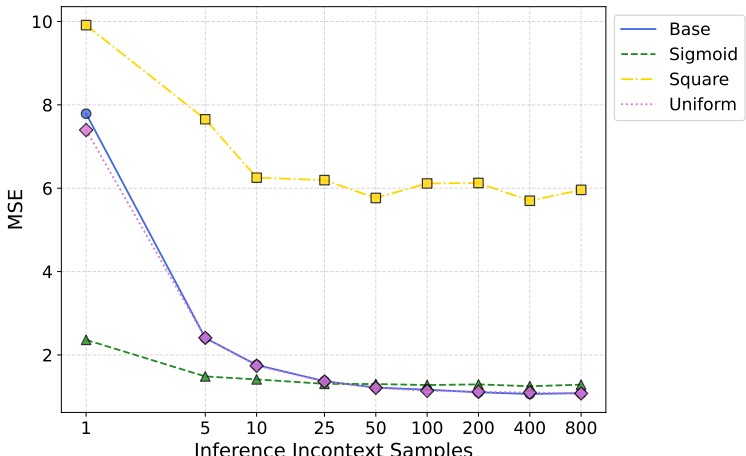

Figure 4: The effect of data misspecification for the ICL of one-step GD. The base setup, $y \sim \mathcal{N}(\boldsymbol{\beta}^\top \mathbf{x}, \sigma^2)$ with $\sigma^2 = 1$, is well-specified. We then consider three misspecification scenarios. **Uniform**: $y \sim \boldsymbol{\beta}^\top \mathbf{x} + \mathrm{uniform}[-\sqrt{3}, \sqrt{3}]$. **Sigmoid**: $y \sim \mathcal{N}(\mathrm{sigmoid}(\boldsymbol{\beta}^\top \mathbf{x}), \sigma^2)$. **Square**: $y \sim \mathcal{N}((\boldsymbol{\beta}^\top \mathbf{x})^2, \sigma^2)$. We observe that the type of misspecification affects the ICL performance. In particular, the ICL performance declines less when the ground-truth model is closer to a linear model.

Results are shown in Figure 4. We observe that the ICL of one-step GD in the uniform noise case is close to the Gaussian noise case in the base setup. However, when the mean of $y$ is not linearly related to $\boldsymbol{x}$ as in the latter two cases, the ICL of one-step GD is significantly worse than the base setup. The performance deterioration depends on the type of misspecification, with $y \sim \mathcal{N}((\boldsymbol{\beta}^\top \boldsymbol{x})^2, \sigma^2)$ showing the most significant decline.

## A.2 A THREE-LAYER TRANSFORMER

We conduct experiments on the task complexity for training a transformer. We adopt the code by Bai et al. (2023)[2]. We consider a three-layer transformer (GPT model) with 2 heads. We follow the generation process outlined in Assumption 1. Specifically, we sample $(\mathbf{x}_n, y_n)_{n=1}^{N+1}$ as independent copies of $(\mathbf{x}, y)$, where

$$\mathbf{x} \sim \mathcal{N}(0, \mathbf{H}), \quad y \sim \mathcal{N}(\boldsymbol{\beta}^\top \mathbf{x}, \sigma^2), \quad \boldsymbol{\beta} \sim \mathcal{N}(0, \psi^2 \mathbf{I}_d).$$

We treat the first $N$ data points $(\mathbf{x}_n, y_n)_{n=1}^{N}$ as the context examples, $\mathbf{x}_{N+1}$ as the covariate, and $y_{N+1}$ as the response. We configure the experiments with

$$d = 20, \ N = 2d, \ \sigma = 0.5, \ \psi = 1, \ \mathbf{H} = \mathtt{diag}(1, 2^{-4}, \ldots, d^{-4}).$$

For each task, we will sample 64 i.i.d. sequences of $(\mathbf{x}_n, y_n)_{n=1}^{N+1}$. The model is trained with Adam with a learning rate of 0.0001. We set the number of context examples during inference to be $M = N$. The results are presented in Figure 5. Similarly to the one-step GD model, we also observe that the ICL error decreases as the number of pretraining tasks increases, approaching the performance of the Bayes optimal algorithm, ridge regression.

## B SINGLE-LAYER LINEAR ATTENTION AND ONE-STEP GD

Results in this part largely follow from (Ahn et al., 2023; Zhang et al., 2023a). We include them here for completeness.

Denote the prompt by

$$\mathbf{Z} := \begin{pmatrix} \mathbf{X}^\top & \mathbf{x} \\ \mathbf{y}^\top & 0 \end{pmatrix} \in \mathbb{R}^{(d+1)\times(n+1)}.$$

---

[2]https://github.com/allenbai01/transformers-as-statisticians/tree/main

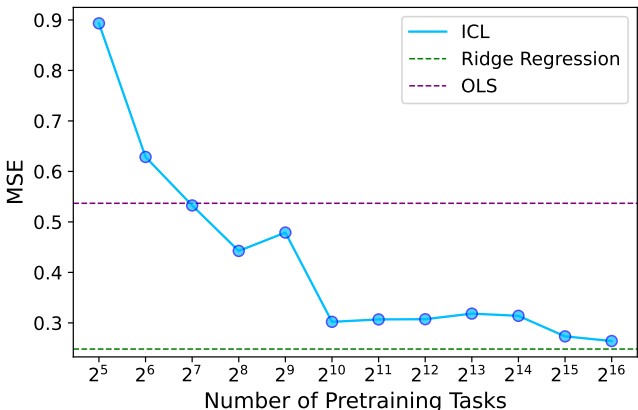

Figure 5:   ICL of a three-layer transformer. The linear regression tasks are generated according to Assumption 1, with $d = 20$, $N = 2d$, standard deviation $\sigma = 0.5$, scaling factor $\psi = 1$, and a polynomial decay spectrum $\lambda_i = i^{-4}$. We fix the number of context examples during inference to $M = N$. We observe that the ICL error decreases as the number of pretraining tasks increases, approaching the performance of the Bayes optimal algorithm, ridge regression.

Denote the query, key, and value parameters by

$$\mathbf{Q}, \mathbf{K}, \mathbf{V} \in \mathbb{R}^{(d+1)\times(d+1)}.$$

Then the single-layer attention with residue connection outputs

$$\mathbf{Z} + (\mathbf{V}\mathbf{Z})\frac{(\mathbf{Q}\mathbf{Z})^\top(\mathbf{K}\mathbf{Z})}{n} \in \mathbb{R}^{(d+1)\times(n+1)}.$$

The prediction is the bottom right entry of the above matrix, that is

$$
\begin{aligned}
\hat{y} &= \left[\mathbf{Z} + \frac{1}{n}(\mathbf{V}\mathbf{Z})(\mathbf{Q}\mathbf{Z})^\top(\mathbf{K}\mathbf{Z})\right]_{d+1,n+1} \\
&= \mathbf{e}_{d+1}^\top\left(\mathbf{Z} + \frac{1}{n}\mathbf{V}\mathbf{Z}\mathbf{Z}^\top\mathbf{Q}^\top\mathbf{K}\mathbf{Z}\right)\mathbf{e}_{n+1} \\
&= 0 + \frac{1}{n}\mathbf{e}_{d+1}^\top\mathbf{V}\mathbf{Z}\mathbf{Z}^\top\mathbf{Q}^\top\mathbf{K}\mathbf{Z}\mathbf{e}_{n+1} \\
&= \frac{1}{n}(\mathbf{e}_{d+1}^\top\mathbf{V})\begin{pmatrix}\mathbf{X}^\top\mathbf{X}+\mathbf{x}\mathbf{x}^\top & \mathbf{X}^\top\mathbf{y} \\ \mathbf{y}^\top\mathbf{X} & \mathbf{y}^\top\mathbf{y}\end{pmatrix}\mathbf{Q}^\top\mathbf{K}\begin{pmatrix}\mathbf{x} \\ 0\end{pmatrix},
\end{aligned}
$$

Our key assumption is that the bottom left $1 \times d$ block in $\mathbf{V}$ is fixed to be zero and the bottom left $1 \times d$ block in $\mathbf{Q}^\top\mathbf{K}$ is fixed to be zero, that is, we assume that

$$\mathbf{V} = \begin{pmatrix} * & * \\ 0 & v \end{pmatrix}, \quad \mathbf{Q}\mathbf{K}^\top = \begin{pmatrix} \mathbf{W} & * \\ 0 & * \end{pmatrix},$$

where $v \in \mathbb{R}$ and $\mathbf{W} \in \mathbb{R}^{d\times d}$ are relevant free parameters. Then we have

$$
\begin{aligned}
\hat{y} &= \frac{1}{n}(0, v)\begin{pmatrix}\mathbf{X}^\top\mathbf{X}+\mathbf{x}\mathbf{x}^\top & \mathbf{X}^\top\mathbf{y} \\ \mathbf{y}^\top\mathbf{X} & \mathbf{y}^\top\mathbf{y}\end{pmatrix}\begin{pmatrix}\mathbf{W}\mathbf{x} \\ 0\end{pmatrix} \\
&= \frac{v}{n}\mathbf{y}^\top\mathbf{X}\mathbf{W}\mathbf{x} \\
&= \left\langle (v\mathbf{W}^\top)\frac{\mathbf{X}^\top\mathbf{y}}{n}, \, \mathbf{x} \right\rangle,
\end{aligned}
$$

which recovers one-step GD when we replace $v\mathbf{W}^\top$ by $\boldsymbol{\Gamma}$, i.e., the update formula in (1).

## C  POPULATION ICL RISK

**Lemma C.1.** *Suppose that the rows in $\mathbf{X} \in \mathbb{R}^{N \times d}$ are generated independently*

$$\mathbf{X}[i] \sim \mathcal{N}(0, \mathbf{H}), \quad i = 1, \dots, N.$$

*Then for every PSD matrix $\mathbf{A}$, it holds that*

$$\mathbb{E}[\mathbf{X}^\top \mathbf{X} \mathbf{A} \mathbf{X}^\top \mathbf{X}] = N\mathtt{tr}(\mathbf{H}\mathbf{A})\mathbf{H} + N(N+1)\mathbf{H}\mathbf{A}\mathbf{H}.$$

*Proof of Lemma C.1.* This is by direct computing.

$$\begin{aligned}
\mathbb{E}[\mathbf{X}^\top \mathbf{X} \mathbf{A} \mathbf{X}^\top \mathbf{X}] &= \mathbb{E}\Big( \sum_i \mathbf{x}_i \mathbf{x}_i^\top \mathbf{A} \sum_j \mathbf{x}_j \mathbf{x}_j^\top \Big) \\
&= N \mathbb{E} \mathbf{x} \mathbf{x}^\top \mathbf{A} \mathbf{x} \mathbf{x}^\top + N(N-1)\mathbf{H}\mathbf{A}\mathbf{H} \\
&= N\big( \mathtt{tr}(\mathbf{H}\mathbf{A})\mathbf{H} + 2\mathbf{H}\mathbf{A}\mathbf{H} \big) + N(N-1)\mathbf{H}\mathbf{A}\mathbf{H} \\
&= N\mathtt{tr}(\mathbf{H}\mathbf{A})\mathbf{H} + N(N+1)\mathbf{H}\mathbf{A}\mathbf{H}.
\end{aligned}$$

This completes the proof. $\qquad\square$

We are ready to present the proof of Theorem 3.1.

*Proof of Theorem 3.1.* Let $\tilde{\boldsymbol{\beta}}$ be the task parameter and let

$$\epsilon := y - \mathbf{x}^\top \tilde{\boldsymbol{\beta}}, \quad \boldsymbol{\epsilon} := \mathbf{y} - \mathbf{X}\tilde{\boldsymbol{\beta}}.$$

Then from Assumption 1, we have

$$\mathbf{x} \sim \mathcal{N}(0, \mathbf{H}), \quad \mathbf{X}[i] \sim \mathcal{N}(0, \mathbf{H}), \quad \tilde{\boldsymbol{\beta}} \sim \mathcal{N}(0, \psi^2 \mathbf{I}), \quad \epsilon \sim \mathcal{N}(0, \sigma^2), \quad \boldsymbol{\epsilon} \sim \mathcal{N}(0, \sigma^2 \mathbf{I}_N).$$

Bringing this into (2), we have

$$\begin{aligned}
\mathcal{R}_N(\boldsymbol{\Gamma}) &= \mathbb{E}\Big( \Big\langle \frac{1}{N} \boldsymbol{\Gamma} \mathbf{X}^\top \mathbf{y}, \ \mathbf{x} \Big\rangle - y \Big)^2 \qquad \text{by (1) and (2)} \\
&= \mathbb{E}\Big( \mathbf{x}^\top \boldsymbol{\Gamma} \frac{1}{N} \mathbf{X}^\top \mathbf{X} \tilde{\boldsymbol{\beta}} + \mathbf{x}^\top \boldsymbol{\Gamma} \frac{1}{N} \mathbf{X}^\top \boldsymbol{\epsilon} - \mathbf{x}^\top \tilde{\boldsymbol{\beta}} - \epsilon \Big)^2 \\
&= \mathbb{E}\Big( \mathbf{x}^\top \Big( \mathbf{I} - \boldsymbol{\Gamma} \frac{1}{N} \mathbf{X}^\top \mathbf{X} \Big) \tilde{\boldsymbol{\beta}} \Big)^2 + \frac{1}{N^2} \mathbb{E}(\mathbf{x}^\top \boldsymbol{\Gamma} \mathbf{X}^\top \boldsymbol{\epsilon})^2 + \sigma^2 \\
&= \Big\langle \mathbb{E}[\mathbf{x}^{\otimes 2}], \ \mathbb{E}\Big( \mathbf{I} - \boldsymbol{\Gamma} \frac{1}{N} \mathbf{X}^\top \mathbf{X} \Big)^{\otimes 2} \circ \mathbb{E}[\tilde{\boldsymbol{\beta}}^{\otimes 2}] \Big\rangle + \frac{1}{N^2} \Big\langle \mathbb{E}[\mathbf{x}^{\otimes 2}], \ \boldsymbol{\Gamma} \mathbb{E}[\mathbf{X}^\top \boldsymbol{\epsilon} \boldsymbol{\epsilon}^\top \mathbf{X}] \boldsymbol{\Gamma}^\top \Big\rangle + \sigma^2 \\
&= \Big\langle \mathbf{H}, \ \mathbb{E}\Big( \mathbf{I} - \boldsymbol{\Gamma} \frac{1}{N} \mathbf{X}^\top \mathbf{X} \Big)^{\otimes 2} \circ (\psi^2 \mathbf{I}) \Big\rangle + \frac{1}{N^2} \Big\langle \mathbf{H}, \ N\sigma^2 \boldsymbol{\Gamma} \mathbf{H} \boldsymbol{\Gamma}^\top \Big\rangle + \sigma^2 \\
&= \Big\langle \mathbf{H}, \ \psi^2 \mathbb{E}\Big( \mathbf{I} - \boldsymbol{\Gamma} \frac{1}{N} \mathbf{X}^\top \mathbf{X} \Big)\Big( \mathbf{I} - \boldsymbol{\Gamma} \frac{1}{N} \mathbf{X}^\top \mathbf{X} \Big)^\top + \frac{\sigma^2}{N} \boldsymbol{\Gamma} \mathbf{H} \boldsymbol{\Gamma}^\top \Big\rangle + \sigma^2. \qquad (15)
\end{aligned}$$

Next, we compute the matrix in (15) that involves $\boldsymbol{\Gamma}$, that is

$$\begin{aligned}
& \psi^2 \mathbb{E}\Big( \mathbf{I} - \boldsymbol{\Gamma} \frac{1}{N} \mathbf{X}^\top \mathbf{X} \Big)\Big( \mathbf{I} - \boldsymbol{\Gamma} \frac{1}{N} \mathbf{X}^\top \mathbf{X} \Big)^\top + \frac{\sigma^2}{N} \boldsymbol{\Gamma} \mathbf{H} \boldsymbol{\Gamma}^\top \\
&= \psi^2 \mathbf{I} - \psi^2 (\boldsymbol{\Gamma} \mathbf{H} + \mathbf{H} \boldsymbol{\Gamma}^\top) + \boldsymbol{\Gamma}\Big( \frac{\psi^2}{N^2} \mathbb{E}[\mathbf{X}^\top \mathbf{X} \mathbf{X}^\top \mathbf{X}] + \frac{\sigma^2}{N} \mathbf{H} \Big) \boldsymbol{\Gamma}^\top \\
&= \psi^2 \mathbf{I} - \psi^2 (\boldsymbol{\Gamma} \mathbf{H} + \mathbf{H} \boldsymbol{\Gamma}^\top) + \boldsymbol{\Gamma}\Big( \frac{\psi^2}{N^2} \big( N\mathtt{tr}(\mathbf{H})\mathbf{H} + N(N+1)\mathbf{H}^2 \big) + \frac{\sigma^2}{N} \mathbf{H} \Big) \boldsymbol{\Gamma}^\top \qquad \text{by Lemma C.1} \\
&= \psi^2 \mathbf{I} - \psi^2 (\boldsymbol{\Gamma} \mathbf{H} + \mathbf{H} \boldsymbol{\Gamma}^\top) + \boldsymbol{\Gamma} \tilde{\mathbf{H}}_N \boldsymbol{\Gamma}^\top \qquad \text{by (4)}
\end{aligned}$$

$$= \big(\boldsymbol{\Gamma} - \boldsymbol{\Gamma}_N^*\big)\tilde{\mathbf{H}}_N\big(\boldsymbol{\Gamma} - \boldsymbol{\Gamma}_N^*\big)^\top + \psi^2\mathbf{I} - \boldsymbol{\Gamma}_N^*\tilde{\mathbf{H}}_N\big(\boldsymbol{\Gamma}_N^*\big)^\top,$$

where the last equality is because $\boldsymbol{\Gamma}_N^* := \psi^2\mathbf{H}\tilde{\mathbf{H}}_N^{-1}$ by (3). Here, we define

$$\tilde{\mathbf{H}}_N := \mathbb{E}\Big(\frac{1}{N}\mathbf{X}^\top\mathbf{y}\Big)\Big(\frac{1}{N}\mathbf{X}^\top\mathbf{y}\Big)^\top = \psi^2\mathbf{H}\Big(\frac{\mathtt{tr}(\mathbf{H}) + \sigma^2/\psi^2}{N}\mathbf{I} + \frac{N+1}{N}\mathbf{H}\Big),$$

$$\boldsymbol{\Gamma}_N^* := \psi^2\mathbf{H}\tilde{\mathbf{H}}_N^{-1}.$$

Bringing this back to (15), we have

$$\mathcal{R}_N(\boldsymbol{\Gamma}) = \Big\langle \mathbf{H}, \ \big(\boldsymbol{\Gamma} - \boldsymbol{\Gamma}_N^*\big)\tilde{\mathbf{H}}_N\big(\boldsymbol{\Gamma} - \boldsymbol{\Gamma}_N^*\big)^\top \Big\rangle + \Big\langle \mathbf{H}, \ \psi^2\mathbf{I} - \boldsymbol{\Gamma}_N^*\tilde{\mathbf{H}}_N\big(\boldsymbol{\Gamma}_N^*\big)^\top \Big\rangle + \sigma^2.$$

It is clear that

$$\min \mathcal{R}_N(\cdot) = \Big\langle \mathbf{H}, \ \psi^2\mathbf{I} - \boldsymbol{\Gamma}_N^*\tilde{\mathbf{H}}_N\big(\boldsymbol{\Gamma}_N^*\big)^\top \Big\rangle + \sigma^2,$$

and

$$\mathcal{R}_N(\boldsymbol{\Gamma}) - \min \mathcal{R}_N(\cdot) = \Big\langle \mathbf{H}, \ \big(\boldsymbol{\Gamma} - \boldsymbol{\Gamma}_N^*\big)\tilde{\mathbf{H}}_N\big(\boldsymbol{\Gamma} - \boldsymbol{\Gamma}_N^*\big)^\top \Big\rangle.$$

We now compute $\min \mathcal{R}_N(\cdot)$ as follows:

$$\min \mathcal{R}_N(\cdot)$$
$$= \Big\langle \mathbf{H}, \ \psi^2\mathbf{I} - \boldsymbol{\Gamma}_N^*\tilde{\mathbf{H}}_N\big(\boldsymbol{\Gamma}_N^*\big)^\top \Big\rangle + \sigma^2$$
$$= \Big\langle \mathbf{H}, \ \psi^2\mathbf{I} - \psi^4\mathbf{H}^2\tilde{\mathbf{H}}_N^{-1} \Big\rangle + \sigma^2$$
$$= \Big\langle \psi^2\mathbf{H}\tilde{\mathbf{H}}_N^{-1}, \ \tilde{\mathbf{H}}_N - \psi^2\mathbf{H}^2 \Big\rangle + \sigma^2$$
$$= \Big\langle \Big(\frac{\mathtt{tr}(\mathbf{H}) + \sigma^2/\psi^2}{N}\mathbf{I} + \frac{N+1}{N}\mathbf{H}\Big)^{-1}, \ \psi^2\mathbf{H}\Big(\frac{\mathtt{tr}(\mathbf{H}) + \sigma^2/\psi^2}{N}\mathbf{I} + \frac{1}{N}\mathbf{H}\Big) \Big\rangle + \sigma^2$$
$$= \psi^2\mathtt{tr}\Big(\big(\big(\mathtt{tr}(\mathbf{H}) + \sigma^2/\psi^2\big)\mathbf{I} + (N+1)\mathbf{H}\big)^{-1}\big(\big(\mathtt{tr}(\mathbf{H}) + \sigma^2/\psi^2\big)\mathbf{H} + \mathbf{H}^2\big)\Big) + \sigma^2,$$

which completes the proof. □

## D   THE TASK COMPLEXITY FOR PRETRAINING AN ATTENTION MODEL

### D.1   PRELIMINARIES OF OPERATOR METHODS

**Tensor product.**   We use $\otimes$ to denote the tensor product or Kronecker product. For convenience, we follow the tensor product convention used by Bach & Moulines (2013); Dieuleveut et al. (2017); Jain et al. (2018; 2017); Zou et al. (2021); Wu et al. (2022) for analyzing SGD.

**Definition 1** (Tensor product).   For matrices $\mathbf{A}$ and $\mathbf{B}$ of any shape, $\mathbf{B}^\top \otimes \mathbf{A}$ is an operator on matrices of an appropriate shape. Specifically, for matrix $\mathbf{X}$ of an appropriate shape, define

$$(\mathbf{B}^\top \otimes \mathbf{A}) \circ \mathbf{X} := \mathbf{A}\mathbf{X}\mathbf{B}.$$

It is clear that $\mathbf{B}^\top \otimes \mathbf{A}$ is a linear operator. For simplicity, we also write

$$\mathbf{A}^{\otimes 2} := \mathbf{A} \otimes \mathbf{A}.$$

We introduce a few facts about linear operators on matrices.

**Fact D.1.**   *For matrices $\mathbf{A}$, $\mathbf{B}$, $\mathbf{C}$, and $\mathbf{D}$ of an appropriate shape, it holds that*

$$(\mathbf{D}^\top \otimes \mathbf{C}) \circ (\mathbf{B}^\top \otimes \mathbf{A}) = \big(\mathbf{D}^\top\mathbf{B}^\top\big) \otimes \big(\mathbf{C}\mathbf{A}\big).$$

*Proof.*   For matrix $\mathbf{X}$ of an appropriate shape, we have

$$(\mathbf{D}^\top \otimes \mathbf{C}) \circ (\mathbf{B}^\top \otimes \mathbf{A}) \circ \mathbf{X} = (\mathbf{D}^\top \otimes \mathbf{C}) \circ \big(\mathbf{A}\mathbf{X}\mathbf{B}\big)$$
$$= \mathbf{C}\mathbf{A}\mathbf{X}\mathbf{B}\mathbf{D}$$
$$= \big(\mathbf{D}^\top\mathbf{B}^\top\big) \otimes \big(\mathbf{C}\mathbf{A}\big) \circ \mathbf{X},$$

which verifies the claim. □

**PSD operators.**  A key notion in our analysis is that of *PSD operators*, which map a PSD matrix to another PSD matrix.

**Definition 2** (PSD operator).  For a linear operator on matrices

$$\mathcal{O} : \mathbb{R}^{d \times d} \to \mathbb{R}^{d \times d},$$

we say $\mathcal{O}$ is a PSD operator, if

$$\mathcal{O} \circ \mathbf{A} \succeq 0, \quad \text{for every } \mathbf{A} \succeq 0.$$

**Definition 3** (Operator order).  For two linear operators on matrices

$$\mathcal{O}_1, \mathcal{O}_2 : \mathbb{R}^{d \times d} \to \mathbb{R}^{d \times d},$$

we say

$$\mathcal{O}_1 \preceq \mathcal{O}_2,$$

if $\mathcal{O}_2 - \mathcal{O}_1$ is a PSD operator.

### D.2  Bias-Variance Decomposition

**SGD iterates.**  Fix the current iterate index as $t \geq 1$. Recall that

$$\frac{\partial}{\partial \mathbf{\Gamma}} \mathcal{R}(\mathbf{\Gamma}; \mathbf{X}_t, \mathbf{y}_t, \mathbf{x}_t, y_t) = \mathbf{x}_t \mathbf{x}_t^\top (\mathbf{\Gamma} - \mathbf{\Gamma}^*) \left( \frac{1}{N} \mathbf{X}_t^\top \mathbf{y}_t \right) \left( \frac{1}{N} \mathbf{X}_t^\top \mathbf{y}_t \right)^\top + \mathbf{\Xi}_t,$$

where $\mathbf{\Gamma}^*$ is defined in (3) and

$$\mathbf{\Xi}_t := \mathbf{x}_t \mathbf{x}_t^\top \mathbf{\Gamma}^* \left( \frac{1}{N} \mathbf{X}_t^\top \mathbf{y}_t \right) \left( \frac{1}{N} \mathbf{X}_t^\top \mathbf{y}_t \right)^\top - y_t \mathbf{x}_t \left( \frac{1}{N} \mathbf{X}_t^\top \mathbf{y}_t \right)^\top. \tag{16}$$

The next lemma shows that $\mathbf{\Xi}$ has zero mean and hence behaves like a "noise".

**Lemma D.2.**  *For random matrix $\mathbf{\Xi}_t$ defined in (16), it holds that $\mathbb{E}[\mathbf{\Xi}_t] = 0$.*

*Proof.*  This is because

$$
\begin{aligned}
\mathbb{E}\mathbf{\Xi}_t &= \mathbb{E}\left[ \mathbf{x}\mathbf{x}^\top \mathbf{\Gamma}^* \left( \frac{1}{N} \mathbf{X}^\top \mathbf{y} \right) \left( \frac{1}{N} \mathbf{X}^\top \mathbf{y} \right)^\top - \mathbf{x} y \left( \frac{1}{N} \mathbf{X}^\top \mathbf{y} \right)^\top \right] \\
&= \mathbf{H}\mathbf{\Gamma}^* \mathbb{E}\left( \frac{1}{N} \mathbf{X}^\top \mathbf{y} \right) \left( \frac{1}{N} \mathbf{X}^\top \mathbf{y} \right)^\top - \mathbb{E}\mathbf{x}\mathbf{x}^\top \tilde{\boldsymbol{\beta}} \left( \frac{1}{N} \mathbf{X}^\top \mathbf{X} \tilde{\boldsymbol{\beta}} \right)^\top \\
&= \mathbf{H}\mathbf{\Gamma}^* \tilde{\mathbf{H}} - \psi^2 \mathbf{H}^2 \\
&= 0,
\end{aligned}
$$

where $\tilde{\mathbf{H}}$ is defined in (4) and it holds that

$$\tilde{\mathbf{H}} = (\mathbf{\Gamma}^*)^{-1} \psi^2 \mathbf{H}.$$

We complete the proof.  $\square$

We can now write the SGD update as

$$
\begin{aligned}
\mathbf{\Gamma}_t &= \mathbf{\Gamma}_{t-1} - \gamma_t \frac{\partial}{\partial \mathbf{\Gamma}} \mathcal{R}(\mathbf{\Gamma}_{t-1}; \mathbf{X}_t, \mathbf{y}_t, \mathbf{x}_t, y_t) \\
&= \mathbf{\Gamma}_{t-1} - \gamma_t \mathbf{x}_t \mathbf{x}_t^\top (\mathbf{\Gamma}_{t-1} - \mathbf{\Gamma}^*) \left( \frac{1}{N} \mathbf{X}_t^\top \mathbf{y}_t \right) \left( \frac{1}{N} \mathbf{X}_t^\top \mathbf{y}_t \right)^\top - \gamma_t \mathbf{\Xi}_t, \quad t = 1, \dots, T,
\end{aligned}
$$

where $(\gamma_t)_{t=1}^T$ is a stepsize schedule defined by (7).

Define

$$\mathbf{\Lambda}_t := \mathbf{\Gamma}_t - \mathbf{\Gamma}^*,$$

then we have

$$\mathbf{\Lambda}_t = \mathbf{\Lambda}_{t-1} - \gamma_t \mathbf{x}_t \mathbf{x}_t^\top \mathbf{\Lambda}_{t-1} \left( \frac{1}{N} \mathbf{X}_t^\top \mathbf{y}_t \right) \left( \frac{1}{N} \mathbf{X}_t^\top \mathbf{y}_t \right)^\top - \gamma_t \mathbf{\Xi}_t.$$

**Bias-variance decomposition.** Define

$$\mathscr{P}_t : \mathbb{R}^{d \times d} \to \mathbb{R}^{d \times d}$$

$$\mathbf{A} \mapsto \mathbf{A} - \gamma_t \mathbf{x}_t \mathbf{x}_t^\top \mathbf{A} \left( \frac{1}{N} \mathbf{X}_t^\top \mathbf{y}_t \right) \left( \frac{1}{N} \mathbf{X}_t^\top \mathbf{y}_t \right)^\top.$$

It is clear that $\mathscr{P}_t$ is a linear map on matrices. Then we have

$$\mathbf{\Lambda}_t = \mathscr{P}_t \circ \mathbf{\Lambda}_{t-1} - \gamma_t \mathbf{\Xi}_t, \quad t \geq 1.$$

Solving the recursion, we have

$$\mathbf{\Lambda}_T = \prod_{t=1}^T \mathscr{P}_t \circ \mathbf{\Lambda}_0 - \sum_{t=1}^T \gamma_t \prod_{k=t+1}^T \mathscr{P}_k \circ \mathbf{\Xi}_t.$$

Taking outer product and expectation, we have

$$\mathcal{A}_T := \mathbb{E} \mathbf{\Lambda}_T^{\otimes 2}$$

$$= \mathbb{E} \left( \prod_{t=1}^T \mathscr{P}_t \circ \mathbf{\Lambda}_0 - \sum_{t=1}^T \gamma_t \prod_{k=t+1}^T \mathscr{P}_k \circ \mathbf{\Xi}_t \right)^{\otimes 2}$$

$$\preceq 2 \mathbb{E} \left( \prod_{t=1}^T \mathscr{P}_t \circ \mathbf{\Lambda}_0 \right)^{\otimes 2} + 2 \mathbb{E} \left( \sum_{t=1}^T \gamma_t \prod_{k=t+1}^T \mathscr{P}_k \circ \mathbf{\Xi}_t \right)^{\otimes 2}$$

$$=: 2 \mathcal{B}_T + 2 \mathcal{C}_T,$$

where we define

$$\mathcal{B}_T := \mathbb{E} \left( \prod_{t=1}^T \mathscr{P}_t \circ \mathbf{\Lambda}_0 \right)^{\otimes 2}, \tag{17}$$

$$\mathcal{C}_T := \mathbb{E} \left( \sum_{t=1}^T \gamma_t \prod_{k=t+1}^T \mathscr{P}_k \circ \mathbf{\Xi}_t \right)^{\otimes 2}. \tag{18}$$

Therefore, we can bound the average risk by

$$\mathbb{E} \mathcal{R}_N(\mathbf{\Gamma}_T) - \min \mathcal{R}_N(\cdot) = \mathbb{E} \langle \mathbf{H}, \ (\mathbf{\Gamma}_T - \mathbf{\Gamma}^*) \tilde{\mathbf{H}} (\mathbf{\Gamma}_T - \mathbf{\Gamma}^*)^\top \rangle \qquad \text{by Theorem 3.1}$$

$$= \langle \mathbf{H}, \ \mathcal{A}_T \circ \tilde{\mathbf{H}} \rangle$$

$$\leq 2 \langle \mathbf{H}, \ \mathcal{B}_T \circ \tilde{\mathbf{H}} \rangle + 2 \langle \mathbf{H}, \ \mathcal{C}_T \circ \tilde{\mathbf{H}} \rangle.$$

The above gives the bias-variance decomposition of the risk.

**Operators and operator maps.** Define the following three linear operators on symmetric matrices:

$$\mathcal{M} := \mathbb{E} (\mathbf{x} \mathbf{x}^\top)^{\otimes 2}, \tag{19}$$

$$\mathcal{L} := \mathbb{E} \left( \left( \frac{1}{N} \mathbf{X}^\top \mathbf{y} \right) \left( \frac{1}{N} \mathbf{X}^\top \mathbf{y} \right)^\top \right)^{\otimes 2}, \tag{20}$$

$$\mathcal{N} := \mathbb{E} \left[ \mathbf{\Xi}^{\otimes 2} \right]. \tag{21}$$

It is easy to verify that all three operators are PSD operators, that is, a PSD matrix is mapped to another PSD matrix.

Define the following *SGD* map on linear operators:

$$\mathscr{S}_t : \left( \mathbb{R}^{d \times d} \right)^{\otimes 2} \to \left( \mathbb{R}^{d \times d} \right)^{\otimes 2}$$

$$\mathcal{O} \mapsto \mathcal{O} - \gamma_t \Big( (\mathbf{H} \otimes \mathbf{I}) \circ \mathcal{O} \circ (\tilde{\mathbf{H}} \otimes \mathbf{I}) + (\mathbf{I} \otimes \mathbf{H}) \circ \mathcal{O} \circ (\mathbf{I} \otimes \tilde{\mathbf{H}}) \Big) \tag{22}$$

$$+ \gamma_t^2 \mathcal{M} \circ \mathcal{O} \circ \mathcal{L}.$$

Similarly, define a *GD* map on linear operators:

$$
\begin{aligned}
\mathscr{G}_t : \left(\mathbb{R}^{d\times d}\right)^{\otimes 2} &\to \left(\mathbb{R}^{d\times d}\right)^{\otimes 2} \\
\mathcal{O} \mapsto \ &\mathcal{O} - \gamma_t\Big((\mathbf{H}\otimes\mathbf{I})\circ\mathcal{O}\circ(\tilde{\mathbf{H}}\otimes\mathbf{I}) + (\mathbf{I}\otimes\mathbf{H})\circ\mathcal{O}\circ(\mathbf{I}\otimes\tilde{\mathbf{H}})\Big) \\
&+ \gamma_t^2\mathbf{H}^{\otimes 2}\circ\mathcal{O}\circ\tilde{\mathbf{H}}^{\otimes 2}.
\end{aligned}
\tag{23}
$$

When the context is clear, we also use $\mathscr{G}$ and $\mathscr{S}$ and ignore the subscript in stepsize $\gamma_t$. When the context is clear, we also write

$$
\mathscr{G}(\mathcal{O}) = \mathscr{G}\circ\mathcal{O}, \quad \mathscr{S}(\mathcal{O}) = \mathscr{S}\circ\mathcal{O}.
$$

The following lemma explains the reason we call these two maps SGD and GD maps, respectively.

**Lemma D.3** (GD and SGD maps). *We have the following properties of the GD and SGD maps defined in* (23) *and* (22)*, respectively.*

1. *$\mathscr{G}$ and $\mathscr{S}$ are both linear maps over the space of matrix operators, i.e., for every pair of matrix operators $\mathcal{O}_1$ and $\mathcal{O}_2$ and every scalar $a\in\mathbb{R}$,*

$$
\mathscr{G}(\mathcal{O}_1 + a\mathcal{O}_2) = \mathscr{G}(\mathcal{O}_1) + a\mathscr{G}(\mathcal{O}_2), \quad \mathscr{S}(\mathcal{O}_1 + a\mathcal{O}_2) = \mathscr{S}(\mathcal{O}_1) + a\mathscr{S}(\mathcal{O}_2).
$$

2. *For every matrix $\mathbf{P}$ of an appropriate shape, it holds that*

$$
\mathscr{G}(\mathbf{P}^{\otimes 2}) = (\mathbf{P} - \gamma\mathbf{H}\mathbf{P}\tilde{\mathbf{H}})^{\otimes 2}
$$

*and that*

$$
\mathscr{S}(\mathbf{P}^{\otimes 2}) = \mathbb{E}(\mathscr{P}\circ\mathbf{P})^{\otimes 2} = \mathbb{E}\left(\mathbf{P} - \gamma\mathbf{x}\mathbf{x}^\top\mathbf{P}\left(\frac{1}{N}\mathbf{X}^\top\mathbf{y}\right)\left(\frac{1}{N}\mathbf{X}^\top\mathbf{y}\right)^\top\right)^{\otimes 2},
$$

*which corresponds to a single (population) GD and SGD steps on matrix $\mathbf{P}$, respectively.*

3. *As a consequence of the first two conclusions, it holds that $\mathscr{G}(\mathcal{O})$ and $\mathscr{S}(\mathcal{O})$ are both PSD operators if $\mathcal{O}$ is given by*

$$
\mathcal{O} := \mathbb{E}\big[\mathbf{P}\otimes\mathbf{P}\big], \text{ where } \mathbf{P} \text{ is of an appropriate shape and is possibly random.}
$$

4. *It holds that*

$$
\mathscr{G}(\mathbf{0}\otimes\mathbf{0}) = \mathscr{S}(\mathbf{0}\otimes\mathbf{0}) = \mathbf{0}\otimes\mathbf{0}.
$$

*Proof.* The first conclusion is clear by the definitions of (23) and (22).

The second conclusion also follows from the definitions of (23) and (22). For example, we can check that

$$
\begin{aligned}
\mathscr{G}\big(\mathbf{P}^{\otimes 2}\big) &= \mathbf{P}^{\otimes 2} - \gamma\Big((\mathbf{H}\otimes\mathbf{I})\circ\mathbf{P}^{\otimes 2}\circ(\tilde{\mathbf{H}}\otimes\mathbf{I}) + (\mathbf{I}\otimes\mathbf{H})\circ\mathbf{P}^{\otimes 2}\circ(\mathbf{I}\otimes\tilde{\mathbf{H}})\Big) \\
&\quad + \gamma^2\mathbf{H}^{\otimes 2}\circ\mathbf{P}^{\otimes 2}\circ\tilde{\mathbf{H}}^{\otimes 2} \\
&= \mathbf{P}^{\otimes 2} - \gamma\big((\mathbf{H}\mathbf{P}\tilde{\mathbf{H}})\otimes\mathbf{P} + \mathbf{P}\otimes(\mathbf{H}\mathbf{P}\tilde{\mathbf{H}})\big) + \gamma^2(\mathbf{H}\mathbf{P}\tilde{\mathbf{H}})^{\otimes 2} \\
&= \big(\mathbf{P} - \gamma\mathbf{H}\mathbf{P}\tilde{\mathbf{H}}\big)^{\otimes 2}.
\end{aligned}
$$

The third conclusion follows from the first two conclusions.

The last conclusion is clear by the definitions of (23) and (22). $\qquad\square$

**Bias iterate.** Using the SGD map (22), we can re-write (17) recursively as

$$
\begin{aligned}
\mathcal{B}_0 &= \mathbf{\Lambda}^{\otimes 2} = \big(\mathbf{\Gamma}_0 - \mathbf{\Gamma}^*\big)^{\otimes 2}, \\
\mathcal{B}_t &= \mathscr{S}_t\circ\mathcal{B}_{t-1}, \quad t = 1,\ldots,T.
\end{aligned}
\tag{24}
$$

**Variance iterates.** Let us consider the variance iterate defined in (18). Since $\boldsymbol{\Xi}_t$ has zero mean and is independent of $\mathscr{P}_k$ for $k \geq t+1$, we have

$$\mathcal{C}_T = \mathbb{E}\bigg( \sum_{t=1}^{T} \gamma_t \prod_{k=t+1}^{T} \mathscr{P}_k \circ \boldsymbol{\Xi}_t \bigg)^{\otimes 2}$$

$$= \sum_{t=1}^{T} \gamma_t^2 \mathbb{E}\bigg( \prod_{k=t+1}^{T} \mathscr{P}_k \circ \boldsymbol{\Xi}_t \bigg)^{\otimes 2}.$$

Using the SGD map (22) and the noise operator (21), we can re-write the above recursively as

$$\begin{aligned} \mathcal{C}_0 &= \mathbf{0} \otimes \mathbf{0}, \\ \mathcal{C}_t &= \mathscr{S}_t \circ \mathcal{C}_{t-1} + \gamma_t^2 \mathcal{N}, \quad t = 1, \dots, T. \end{aligned} \tag{25}$$

### D.3 SOME OPERATOR BOUNDS

**Lemma D.4.** *Suppose that $\mathbf{z} \in \mathcal{N}(0, \mathbf{I}_d)$, then*

1. *For every $\mathbf{u}, \mathbf{v} \in \mathbb{R}^d$,*
$$\mathbb{E}\langle \mathbf{z}, \mathbf{u} \rangle^2 \langle \mathbf{z}, \mathbf{v} \rangle^2 \leq 3\|\mathbf{u}\|_2^2 \cdot \|\mathbf{v}\|_2^2.$$

2. *For every $\mathbf{u}, \mathbf{v}, \mathbf{w} \in \mathbb{R}^d$,*
$$\mathbb{E}\langle \mathbf{z}, \mathbf{u} \rangle^2 \langle \mathbf{z}, \mathbf{v} \rangle^2 \langle \mathbf{z}, \mathbf{w} \rangle^2 \leq 15\|\mathbf{u}\|_2^2 \cdot \|\mathbf{v}\|_2^2 \cdot \|\mathbf{w}\|_2^2.$$

3. *For every $\mathbf{u}, \mathbf{v}, \mathbf{w}, \mathbf{x} \in \mathbb{R}^d$,*
$$\mathbb{E}\langle \mathbf{z}, \mathbf{u} \rangle^2 \langle \mathbf{z}, \mathbf{v} \rangle^2 \langle \mathbf{z}, \mathbf{w} \rangle^2 \langle \mathbf{z}, \mathbf{x} \rangle^2 \leq 105\|\mathbf{u}\|_2^2 \cdot \|\mathbf{v}\|_2^2 \cdot \|\mathbf{w}\|_2^2 \cdot \|\mathbf{x}\|_2^2.$$

*Proof.* These inequalities can be proved by using Gaussian moment tensor equations in Section 20.5.2 in (Seber, 2008) and Section 11.6 in (Schott, 2016). Specifically, for the fourth moment, we have

$$\begin{aligned} \mathbb{E}\langle \mathbf{z}, \mathbf{u} \rangle^2 \langle \mathbf{z}, \mathbf{v} \rangle^2 &= \mathbb{E}\mathbf{z}^\top \mathbf{u}\mathbf{u}^\top \mathbf{z} \cdot \mathbf{z}^\top \mathbf{v}\mathbf{v}^\top \mathbf{z} \\ &= \mathrm{tr}(\mathbf{u}\mathbf{u}^\top)\mathrm{tr}(\mathbf{v}\mathbf{v}^\top) + 2\mathrm{tr}(\mathbf{u}\mathbf{u}^\top \mathbf{v}\mathbf{v}^\top) \\ &= \|\mathbf{u}\|_2^2 \cdot \|\mathbf{v}\|_2^2 + 2\langle \mathbf{u}, \mathbf{v} \rangle^2 \\ &\leq 3\|\mathbf{u}\|_2^2 \cdot \|\mathbf{v}\|_2^2. \end{aligned}$$

For the sixth moment, we have

$$\begin{aligned} &\mathbb{E}\langle \mathbf{z}, \mathbf{u} \rangle^2 \langle \mathbf{z}, \mathbf{v} \rangle^2 \langle \mathbf{z}, \mathbf{w} \rangle^2 \\ &= \mathbb{E}\mathbf{z}^\top \mathbf{u}\mathbf{u}^\top \mathbf{z} \cdot \mathbf{z}^\top \mathbf{v}\mathbf{v}^\top \mathbf{z} \cdot \mathbf{z}^\top \mathbf{w}\mathbf{w}^\top \mathbf{z} \\ &= \mathrm{tr}(\mathbf{u}\mathbf{u}^\top)\mathrm{tr}(\mathbf{v}\mathbf{v}^\top)\mathrm{tr}(\mathbf{w}\mathbf{w}^\top) + 2\mathrm{tr}(\mathbf{u}\mathbf{u}^\top)\mathrm{tr}(\mathbf{v}\mathbf{v}^\top \mathbf{w}\mathbf{w}^\top) \\ &\quad + 2\mathrm{tr}(\mathbf{v}\mathbf{v}^\top)\mathrm{tr}(\mathbf{u}\mathbf{u}^\top \mathbf{w}\mathbf{w}^\top) + 2\mathrm{tr}(\mathbf{w}\mathbf{w}^\top)\mathrm{tr}(\mathbf{u}\mathbf{u}^\top \mathbf{v}\mathbf{v}^\top) + 8\mathrm{tr}(\mathbf{u}\mathbf{u}^\top \mathbf{v}\mathbf{v}^\top \mathbf{w}\mathbf{w}^\top) \\ &= \|\mathbf{u}\|_2^2 \cdot \|\mathbf{v}\|_2^2 \cdot \|\mathbf{w}\|_2^2 + 2\|\mathbf{u}\|_2^2\langle \mathbf{v}, \mathbf{w} \rangle^2 + 2\|\mathbf{v}\|_2^2\langle \mathbf{u}, \mathbf{w} \rangle^2 \\ &\quad + 2\|\mathbf{w}\|_2^2\langle \mathbf{u}, \mathbf{v} \rangle^2 + 8\langle \mathbf{u}, \mathbf{v} \rangle\langle \mathbf{v}, \mathbf{w} \rangle\langle \mathbf{u}, \mathbf{w} \rangle \\ &\leq 15\|\mathbf{u}\|_2^2 \cdot \|\mathbf{v}\|_2^2 \cdot \|\mathbf{w}\|_2^2. \end{aligned}$$

For the eighth moment, we have

$$\begin{aligned} &\mathbb{E}\langle \mathbf{z}, \mathbf{u} \rangle^2 \langle \mathbf{z}, \mathbf{v} \rangle^2 \langle \mathbf{z}, \mathbf{w} \rangle^2 \langle \mathbf{z}, \mathbf{x} \rangle^2 \\ &= \mathbb{E}\mathbf{z}^\top \mathbf{u}\mathbf{u}^\top \mathbf{z} \cdot \mathbf{z}^\top \mathbf{v}\mathbf{v}^\top \mathbf{z} \cdot \mathbf{z}^\top \mathbf{w}\mathbf{w}^\top \mathbf{z} \cdot \mathbf{z}^\top \mathbf{x}\mathbf{x}^\top \mathbf{z} \\ &= \mathrm{tr}(\mathbf{u}\mathbf{u}^\top)\mathrm{tr}(\mathbf{v}\mathbf{v}^\top)\mathrm{tr}(\mathbf{w}\mathbf{w}^\top)\mathrm{tr}(\mathbf{x}\mathbf{x}^\top) \\ &\quad + 8\Big(\mathrm{tr}(\mathbf{u}\mathbf{u}^\top)\mathrm{tr}(\mathbf{v}\mathbf{v}^\top \mathbf{w}\mathbf{w}^\top \mathbf{x}\mathbf{x}^\top) + \mathrm{tr}(\mathbf{v}\mathbf{v}^\top)\mathrm{tr}(\mathbf{u}\mathbf{u}^\top \mathbf{w}\mathbf{w}^\top \mathbf{x}\mathbf{x}^\top) \end{aligned}$$

$$
\begin{aligned}
&\qquad\qquad + \operatorname{tr}(\mathbf{w}\mathbf{w}^\top)\operatorname{tr}(\mathbf{u}\mathbf{u}^\top\mathbf{v}\mathbf{v}^\top\mathbf{x}\mathbf{x}^\top) + \operatorname{tr}(\mathbf{x}\mathbf{x}^\top)\operatorname{tr}(\mathbf{u}\mathbf{u}^\top\mathbf{v}\mathbf{v}^\top\mathbf{w}\mathbf{w}^\top)\Big) \\
&\quad + 4\Big( \operatorname{tr}(\mathbf{u}\mathbf{u}^\top\mathbf{v}\mathbf{v}^\top)\operatorname{tr}(\mathbf{w}\mathbf{w}^\top\mathbf{x}\mathbf{x}^\top) + \operatorname{tr}(\mathbf{u}\mathbf{u}^\top\mathbf{w}\mathbf{w}^\top)\operatorname{tr}(\mathbf{v}\mathbf{v}^\top\mathbf{x}\mathbf{x}^\top) \\
&\qquad\qquad + \operatorname{tr}(\mathbf{u}\mathbf{u}^\top\mathbf{x}\mathbf{x}^\top)\operatorname{tr}(\mathbf{v}\mathbf{v}^\top\mathbf{w}\mathbf{w}^\top)\Big) \\
&\quad + 2\Big( \operatorname{tr}(\mathbf{u}\mathbf{u}^\top)\operatorname{tr}(\mathbf{v}\mathbf{v}^\top)\operatorname{tr}(\mathbf{w}\mathbf{w}^\top\mathbf{x}\mathbf{x}^\top) + \operatorname{tr}(uB\mathbf{u}^\top)\operatorname{tr}(\mathbf{w}\mathbf{w}^\top)\operatorname{tr}(\mathbf{v}\mathbf{v}^\top\mathbf{x}\mathbf{x}^\top) \\
&\qquad\qquad + \operatorname{tr}(\mathbf{u}\mathbf{u}^\top)\operatorname{tr}(\mathbf{x}\mathbf{x}^\top)\operatorname{tr}(\mathbf{v}\mathbf{v}^\top\mathbf{w}\mathbf{w}^\top) + \operatorname{tr}(\mathbf{v}\mathbf{v}^\top)\operatorname{tr}(\mathbf{w}\mathbf{w}^\top)\operatorname{tr}(\mathbf{u}\mathbf{u}^\top\mathbf{x}\mathbf{x}^\top) \\
&\qquad\qquad + \operatorname{tr}(\mathbf{v}\mathbf{v}^\top)\operatorname{tr}(\mathbf{x}\mathbf{x}^\top)\operatorname{tr}(\mathbf{u}\mathbf{u}^\top\mathbf{w}\mathbf{w}^\top) + \operatorname{tr}(\mathbf{w}\mathbf{w}^\top)\operatorname{tr}(\mathbf{x}\mathbf{x}^\top)\operatorname{tr}(\mathbf{u}\mathbf{u}^\top\mathbf{v}\mathbf{v}^\top)\Big) \\
&\quad + 16\Big( \operatorname{tr}(\mathbf{u}\mathbf{u}^\top\mathbf{v}\mathbf{v}^\top\mathbf{w}\mathbf{w}^\top\mathbf{x}\mathbf{x}^\top) + \operatorname{tr}(\mathbf{u}\mathbf{u}^\top\mathbf{v}\mathbf{v}^\top\mathbf{x}\mathbf{x}^\top\mathbf{w}\mathbf{w}^\top) + \operatorname{tr}(\mathbf{u}\mathbf{u}^\top\mathbf{w}\mathbf{w}^\top\mathbf{v}\mathbf{v}^\top\mathbf{x}\mathbf{x}^\top)\Big)\Big) \\
&= \|\mathbf{u}\|_2^2 \cdot \|\mathbf{v}\|_2^2 \cdot \|\mathbf{w}\|_2^2 \cdot \|\mathbf{x}\|_2^2 \\
&\quad + 8\big( \|\mathbf{u}\|_2^2 \langle \mathbf{v}, \mathbf{w}\rangle\langle \mathbf{w}, \mathbf{x}\rangle\langle \mathbf{v}, \mathbf{x}\rangle + \|\mathbf{v}\|_2^2 \langle \mathbf{u}, \mathbf{w}\rangle\langle \mathbf{w}, \mathbf{x}\rangle\langle \mathbf{u}, \mathbf{x}\rangle \\
&\qquad\qquad + \|\mathbf{w}\|_2^2 \langle \mathbf{u}, \mathbf{v}\rangle\langle \mathbf{u}, \mathbf{x}\rangle\langle \mathbf{v}, \mathbf{x}\rangle + \|\mathbf{x}\|_2^2 \langle \mathbf{u}, \mathbf{v}\rangle\langle \mathbf{u}, \mathbf{w}\rangle\langle \mathbf{v}, \mathbf{w}\rangle \big) \\
&\quad + 4\big( \langle \mathbf{u}, \mathbf{v}\rangle^2 \langle \mathbf{w}, \mathbf{x}\rangle^2 + \langle \mathbf{u}, \mathbf{w}\rangle^2 \langle \mathbf{v}, \mathbf{x}\rangle^2 + \langle \mathbf{u}, \mathbf{x}\rangle^2 \langle \mathbf{v}, \mathbf{w}\rangle^2 \big) \\
&\quad + 2\big( \|\mathbf{u}\|_2^2 \cdot \|\mathbf{v}\|_2^2 \langle \mathbf{w}, \mathbf{x}\rangle^2 + \|\mathbf{u}\|_2^2 \cdot \|\mathbf{w}\|_2^2 \langle \mathbf{v}, \mathbf{x}\rangle^2 + \|\mathbf{u}\|_2^2 \cdot \|\mathbf{x}\|_2^2 \langle \mathbf{v}, \mathbf{w}\rangle^2 \\
&\qquad\qquad + \|\mathbf{v}\|_2^2 \cdot \|\mathbf{w}\|_2^2 \langle \mathbf{u}, \mathbf{x}\rangle^2 + \|\mathbf{v}\|_2^2 \cdot \|\mathbf{x}\|_2^2 \langle \mathbf{u}, \mathbf{w}\rangle^2 + \|\mathbf{w}\|_2^2 \cdot \|\mathbf{x}\|_2^2 \langle \mathbf{u}, \mathbf{v}\rangle^2 \big) \\
&\quad + 16\big( \langle \mathbf{u}, \mathbf{v}\rangle\langle \mathbf{v}, \mathbf{w}\rangle\langle \mathbf{w}, \mathbf{x}\rangle\langle \mathbf{u}, \mathbf{x}\rangle + \langle \mathbf{u}, \mathbf{v}\rangle\langle \mathbf{v}, \mathbf{x}\rangle\langle \mathbf{w}, \mathbf{x}\rangle\langle \mathbf{u}, \mathbf{w}\rangle \\
&\qquad\qquad + \langle \mathbf{u}, \mathbf{w}\rangle\langle \mathbf{v}, \mathbf{w}\rangle\langle \mathbf{v}, \mathbf{x}\rangle\langle \mathbf{u}, \mathbf{x}\rangle \big) \\
&\leq 105 \|\mathbf{u}\|_2^2 \cdot \|\mathbf{v}\|_2^2 \cdot \|\mathbf{w}\|_2^2 \cdot \|\mathbf{x}\|_2^2.
\end{aligned}
$$

We have completed the proof. $\qquad\square$

**Lemma D.5** (Upper bound on $\mathcal{M}$)**.** *Consider $\mathcal{M}$ defined in* (19)*. For every PSD matrix* $\mathbf{A}$*, we have*

$$
\begin{aligned}
\mathcal{M} \circ \mathbf{A} &= \mathbb{E}\big[ \mathbf{x}\mathbf{x}^\top \mathbf{A} \mathbf{x}\mathbf{x}^\top \big] \\
&\preceq 3\langle \mathbf{H}, \mathbf{A}\rangle \mathbf{H}.
\end{aligned}
$$

*Proof.* This follows from Lemma D.4. $\qquad\square$

**Lemma D.6.** *Consider $\mathcal{L}$ defined in* (20)*. For every PSD matrix* $\mathbf{A}$*, we have*

$$
\begin{aligned}
\langle \mathbf{A}, \ \mathcal{L} \circ \mathbf{A}\rangle &= \mathbb{E}\bigg( \Big( \frac{1}{N}\mathbf{X}^\top \mathbf{y}\Big)^\top \mathbf{A}\Big( \frac{1}{N}\mathbf{X}^\top \mathbf{y}\Big)\bigg)^2 \\
&\leq 8 \cdot 3^6 \langle \tilde{\mathbf{H}}, \ \mathbf{A}\rangle^2.
\end{aligned}
$$

*Proof.* By definition, we have

$$
\begin{aligned}
\langle \mathbf{A}, \ \mathcal{L} \circ \mathbf{A}\rangle &= \mathbb{E}\bigg( \Big( \frac{1}{N}\mathbf{X}^\top \mathbf{y}\Big)^\top \mathbf{A}\Big( \frac{1}{N}\mathbf{X}^\top \mathbf{y}\Big)\bigg)^2 \\
&= \mathbb{E}\bigg\| \frac{1}{N}\mathbf{X}^\top \mathbf{y}\bigg\|_{\mathbf{A}}^4 \\
&= \frac{1}{N^4}\mathbb{E}\big\| \mathbf{X}^\top \mathbf{X}\tilde{\boldsymbol{\beta}} + \mathbf{X}^\top \boldsymbol{\epsilon}\big\|_{\mathbf{A}}^4 \\
&\leq \frac{8}{N^4}\Big( \mathbb{E}\big\| \mathbf{X}^\top \mathbf{X}\tilde{\boldsymbol{\beta}}\big\|_{\mathbf{A}}^4 + \mathbb{E}\big\| \mathbf{X}^\top \boldsymbol{\epsilon}\big\|_{\mathbf{A}}^4 \Big).
\end{aligned} \tag{26}
$$

Next, we bound each of the two terms separately.

**Bound on $\mathbb{E}\big\|\mathbf{X}^\top\mathbf{X}\tilde{\boldsymbol{\beta}}\big\|_{\mathbf{A}}^4$.** We have

$$\begin{aligned}
\mathbb{E}\big\|\mathbf{X}^\top\mathbf{X}\tilde{\boldsymbol{\beta}}\big\|_{\mathbf{A}}^4 &= \mathbb{E}\big(\tilde{\boldsymbol{\beta}}\mathbf{X}^\top\mathbf{X}\mathbf{A}\mathbf{X}^\top\mathbf{X}\tilde{\boldsymbol{\beta}}\big)^2 \\
&\leq 3\mathbb{E}\big\langle\psi^2\mathbf{I},\ \mathbf{X}^\top\mathbf{X}\mathbf{A}\mathbf{X}^\top\mathbf{X}\big\rangle^2 \qquad \text{by Lemma D.4} \\
&= 3\psi^4\mathbb{E}\big\langle\mathbf{A},\ \mathbf{X}^\top\mathbf{X}\mathbf{X}^\top\mathbf{X}\big\rangle^2 \\
&= 3\psi^4\sum_{i,j,k,\ell}\mathbb{E}\big\langle\mathbf{A},\ \mathbf{x}_i\mathbf{x}_i^\top\mathbf{x}_j\mathbf{x}_j^\top\big\rangle\big\langle\mathbf{A},\ \mathbf{x}_k\mathbf{x}_k^\top\mathbf{x}_\ell\mathbf{x}_\ell^\top\big\rangle \\
&= 3\psi^4\bigg(\sum_{\text{4-distinct}} + \sum_{\text{3-distinct}} + \sum_{\text{2-distinct}} + \sum_{\text{1-distinct}}\bigg)f(i,j,k,\ell), \qquad (27)
\end{aligned}$$

where we define

$$\begin{aligned}
f(i,j,k,\ell) &:= \mathbb{E}\big\langle\mathbf{A},\ \mathbf{x}_i\mathbf{x}_i^\top\mathbf{x}_j\mathbf{x}_j^\top\big\rangle\big\langle\mathbf{A},\ \mathbf{x}_k\mathbf{x}_k^\top\mathbf{x}_\ell\mathbf{x}_\ell^\top\big\rangle \\
&= \mathbb{E}\big[\mathbf{x}_i^\top\mathbf{A}\mathbf{x}_j\cdot\mathbf{x}_i^\top\mathbf{x}_j\cdot\mathbf{x}_k^\top\mathbf{A}\mathbf{x}_\ell\cdot\mathbf{x}_k^\top\mathbf{x}_\ell\big].
\end{aligned}$$

In (27), we group $f(i,j,k,\ell)$ by their number of distinct indexes (i.e., the number of distinct random variables). We now bound the sum of the terms in each group separately.

- There are no more than $N^4$ terms that have 4 distinct random variables and each such term can be bounded by
$$f(1,2,3,4) = \langle\mathbf{H}^2,\mathbf{A}\rangle\langle\mathbf{H}^2,\mathbf{A}\rangle.$$
  So we have
$$\sum_{\text{4-distinct}}f(i,j,k,\ell) \leq N^4\langle\mathbf{H}^2,\mathbf{A}\rangle^2.$$

- There are no more than $3^4N^3$ terms that have 3 distinct random variables. Due to the i.i.d.-ness, we may assume $\mathbf{x}_1$ appears twice and $\mathbf{x}_2$ and $\mathbf{x}_3$ appear once in such a 3-distinct term without loss of generality. Due to the symmetry of $f(i,j,k,\ell)$, there are essentially two situations.

  1. If two $\mathbf{x}_1$'s appear in the same inner product, such a 3-distinct term can be bounded by
$$\begin{aligned}
f(1,1,2,3) &= \mathbb{E}\big\langle\mathbf{A},\ \mathbf{x}_1\mathbf{x}_1^\top\mathbf{x}_1\mathbf{x}_1^\top\big\rangle\big\langle\mathbf{A},\ \mathbf{x}_2\mathbf{x}_2^\top\mathbf{x}_3\mathbf{x}_3^\top\big\rangle \\
&= \langle\mathbf{A},\mathbf{H}^2\rangle\mathbb{E}\big[\mathbf{x}_1^\top\mathbf{x}_1\cdot\mathbf{x}_1^\top\mathbf{A}\mathbf{x}_1\big] \\
&\leq \langle\mathbf{A},\mathbf{H}^2\rangle\cdot 3\langle\mathbf{H},\mathbf{A}\rangle\mathtt{tr}(\mathbf{H}) \\
&= 3\mathtt{tr}(\mathbf{H})\langle\mathbf{H},\mathbf{A}\rangle\langle\mathbf{H}^2,\mathbf{A}\rangle.
\end{aligned}$$

  2. If two $\mathbf{x}_1$'s appear in different inner products, such a 3-distinct term can be bounded by
$$\begin{aligned}
f(1,2,1,3) &= \mathbb{E}\big[\mathbf{x}_1^\top\mathbf{A}\mathbf{x}_2\cdot\mathbf{x}_1^\top\mathbf{x}_2\cdot\mathbf{x}_1^\top\mathbf{A}\mathbf{x}_3\cdot\mathbf{x}_1^\top\mathbf{x}_3\big] \\
&= \mathbb{E}\big(\mathbf{x}_1^\top\mathbf{A}\mathbf{H}\mathbf{x}_1\big)^2 \\
&\leq 3\langle\mathbf{H}^2,\mathbf{A}\rangle^2 \\
&\leq 3\mathtt{tr}(\mathbf{H})\langle\mathbf{H},\mathbf{A}\rangle\langle\mathbf{H}^2,\mathbf{A}\rangle.
\end{aligned}$$

  Therefore, we can upper bound the sum of all 3-distinct terms by
$$\sum_{\text{3-distinct}}f(i,j,k,\ell) \leq 3^4N^3\cdot 3\mathtt{tr}(\mathbf{H})\langle\mathbf{H},\mathbf{A}\rangle\langle\mathbf{H}^2,\mathbf{A}\rangle.$$

- There are no more than $2^4\cdot N^2$ terms that have 2 distinct random variables. Due to the i.i.d.-ness, we may assume $\mathbf{x}_1$ appears twice and $\mathbf{x}_2$ appears twice in such a 2-distinct term without loss of generality. Due to the symmetricity of $f(i,j,k,\ell)$, there are essentially two situations.

1. If two $\mathbf{x}_1$'s appear in the same inner product, such a 2-distinct term can be bounded by

$$
\begin{aligned}
f(1,1,2,2) &= \mathbb{E}\langle \mathbf{A},\ \mathbf{x}_1\mathbf{x}_1^\top\mathbf{x}_1\mathbf{x}_1^\top \rangle\langle \mathbf{A},\ \mathbf{x}_2\mathbf{x}_2^\top\mathbf{x}_2\mathbf{x}_2^\top \rangle \\
&\leq \big(3\langle \mathbf{H}, \mathbf{A}\rangle \mathtt{tr}(\mathbf{H})\big)^2 \\
&= 9\mathtt{tr}(\mathbf{H})^2\langle \mathbf{H}, \mathbf{A}\rangle^2.
\end{aligned}
$$

2. If two $\mathbf{x}_1$'s appear in different inner products, such a 2-distinct term can be bounded by

$$
\begin{aligned}
f(1,2,1,2) &= \mathbb{E}\big[\mathbf{x}_1^\top\mathbf{A}\mathbf{x}_2 \cdot \mathbf{x}_1^\top\mathbf{x}_2 \cdot \mathbf{x}_1^\top\mathbf{A}\mathbf{x}_2 \cdot \mathbf{x}_1^\top\mathbf{x}_2\big] \\
&= \mathbb{E}\big[\mathbf{x}_1^\top\mathbf{x}_2\mathbf{x}_2^\top\mathbf{x}_1 \cdot \mathbf{x}_1^\top\mathbf{A}\mathbf{x}_2\mathbf{x}_2^\top\mathbf{A}\mathbf{x}_1\big] \\
&\leq 3\mathbb{E}\langle \mathbf{H}, \mathbf{x}_2\mathbf{x}_2^\top\rangle\langle \mathbf{H}, \mathbf{A}\mathbf{x}_2\mathbf{x}_2^\top\mathbf{A}\rangle \\
&= 3\mathbb{E}\big[\mathbf{x}_2^\top\mathbf{H}\mathbf{x}_2\mathbf{x}_2^\top\mathbf{A}\mathbf{H}\mathbf{A}\mathbf{x}_2\big] \\
&\leq 9\mathtt{tr}(\mathbf{H}^2)\langle \mathbf{H}, \mathbf{A}\mathbf{H}\mathbf{A}\rangle \\
&\leq 9\mathtt{tr}(\mathbf{H})^2\langle \mathbf{H}, \mathbf{A}\rangle^2.
\end{aligned}
$$

Therefore, we can upper bound the sum of all 2-distinct terms by

$$
\sum_{\text{2-distinct}} f(i,j,k,\ell) \leq 2^4 N^2 \cdot 9\mathtt{tr}(\mathbf{H})^2\langle \mathbf{H}, \mathbf{A}\rangle^2.
$$

- There are $N$ terms that have only 1 distinct random variable and each such term can be bounded by

$$
\begin{aligned}
f(1,1,1,1) &= \mathbb{E}\big[\|\mathbf{x}\|_2^4\big(\mathbf{x}^\top\mathbf{A}\mathbf{x}\big)^2\big] \\
&\leq 105\mathtt{tr}(\mathbf{H})^2\langle \mathbf{H}, \mathbf{A}\rangle^2.
\end{aligned}
$$

So we have

$$
\sum_{\text{1-distinct}} f(i,j,k,\ell) \leq 105 N\mathtt{tr}(\mathbf{H})^2\langle \mathbf{H}, \mathbf{A}\rangle^2.
$$

Applying these bounds to (27), we get

$$
\begin{aligned}
\mathbb{E}\big\|\mathbf{X}^\top\mathbf{X}\tilde{\boldsymbol{\beta}}\big\|_\mathbf{A}^4 &\leq 3\psi^4\Big(N^4\langle \mathbf{H}^2, \mathbf{A}\rangle^2 + 3^4 N^3 \cdot 3\mathtt{tr}(\mathbf{H})\langle \mathbf{H}, \mathbf{A}\rangle\langle \mathbf{H}^2, \mathbf{A}\rangle \\
&\qquad + 2^4 N^2 \cdot 9\mathtt{tr}(\mathbf{H})^2\langle \mathbf{H}, \mathbf{A}\rangle^2 + 105 N\mathtt{tr}(\mathbf{H})^2\langle \mathbf{H}, \mathbf{A}\rangle^2\Big) \\
&\leq 3^6 N^4 \psi^4\Big(\langle \mathbf{H}^2, \mathbf{A}\rangle^2 + \frac{\mathtt{tr}(\mathbf{H})}{N}\langle \mathbf{H}, \mathbf{A}\rangle\langle \mathbf{H}^2, \mathbf{A}\rangle + \frac{\mathtt{tr}(\mathbf{H})^2}{N^2}\langle \mathbf{H}, \mathbf{A}\rangle^2\Big) \\
&\leq 3^6 N^4 \psi^4\Big(\langle \mathbf{H}^2, \mathbf{A}\rangle + \frac{\mathtt{tr}(\mathbf{H})}{N}\langle \mathbf{H}, \mathbf{A}\rangle\Big)^2 \\
&= 3^6 N^4\Big\langle \psi^2\mathbf{H}\Big(\frac{\mathtt{tr}(\mathbf{H})}{N}\mathbf{I} + \mathbf{H}\Big),\ \mathbf{A}\Big\rangle^2.
\end{aligned} \tag{28}
$$

**Bound on $\mathbb{E}\big\|\mathbf{X}^\top\boldsymbol{\epsilon}\big\|_\mathbf{A}^4$.** We have

$$
\begin{aligned}
\mathbb{E}\big\|\mathbf{X}^\top\boldsymbol{\epsilon}\big\|_\mathbf{A}^4 &= \mathbb{E}\big(\boldsymbol{\epsilon}^\top\mathbf{X}\mathbf{A}\mathbf{X}^\top\boldsymbol{\epsilon}\big)^2 \\
&\leq 3\mathbb{E}\langle \sigma^2\mathbf{I},\ \mathbf{X}\mathbf{A}\mathbf{X}^\top\rangle^2 \\
&= 3\sigma^4\mathbb{E}\langle \mathbf{A},\ \mathbf{X}^\top\mathbf{X}\rangle^2 \\
&= 3\sigma^4 \sum_{i,j} \mathbb{E}\langle \mathbf{A},\ \mathbf{x}_i\mathbf{x}_i^\top\rangle\langle \mathbf{A},\ \mathbf{x}_j\mathbf{x}_j^\top\rangle
\end{aligned}
$$

$$= 3\sigma^4 \left( N\mathbb{E}\langle \mathbf{A}, \mathbf{xx}^\top \rangle^2 + N(N-1)\mathbb{E}\langle \mathbf{A}, \mathbf{x}_1\mathbf{x}_1^\top \rangle\langle \mathbf{A}, \mathbf{x}_2\mathbf{x}_2^\top \rangle \right)$$

$$\leq 3\sigma^4 \left( 3N\langle \mathbf{A}, \mathbf{H} \rangle^2 + N(N-1)\langle \mathbf{A}, \mathbf{H} \rangle^2 \right)$$

$$\leq 3^3\sigma^4 N^2\langle \mathbf{H}, \mathbf{A} \rangle^2$$

$$= 3^3 N^4 \left\langle \psi^2\mathbf{H}\left( \frac{\sigma^2/\psi^2}{N}\mathbf{I} \right),\ \mathbf{A} \right\rangle^2 \tag{29}$$

**Putting things together.** Bring (28) and (29) to (26), we obtain

$$\langle \mathbf{A},\ \mathcal{L}\circ\mathbf{A} \rangle \leq \frac{8}{N^4}\left( 3^6 N^4\left\langle \psi^2\mathbf{H}\left( \frac{\texttt{tr}(\mathbf{H})}{N}\mathbf{I} + \mathbf{H} \right),\ \mathbf{A} \right\rangle^2 + 3^3 N^4\left\langle \psi^2\mathbf{H}\left( \frac{\sigma^2/\psi^2}{N}\mathbf{I} \right),\ \mathbf{A} \right\rangle^2 \right)$$

$$\leq 8\cdot 3^6\left\langle \psi^2\mathbf{H}\left( \frac{\texttt{tr}(\mathbf{H}) + \sigma^2/\psi^2}{N}\mathbf{I} + \mathbf{H} \right),\ \mathbf{A} \right\rangle^2$$

$$\leq 8\cdot 3^6\langle \tilde{\mathbf{H}},\ \mathbf{A} \rangle^2.$$

We have completed the proof. $\qquad\square$

**Lemma D.7** (Upper bound on $\mathcal{L}$). *Consider $\mathcal{L}$ defined in* (20)*. For every PSD matrix $\mathbf{A}$, we have*

$$\mathcal{L}\circ\mathbf{A} = \mathbb{E}\left( \frac{1}{N}\mathbf{X}^\top\mathbf{y} \right)\left( \frac{1}{N}\mathbf{X}^\top\mathbf{y} \right)^\top \mathbf{A}\left( \frac{1}{N}\mathbf{X}^\top\mathbf{y} \right)\left( \frac{1}{N}\mathbf{X}^\top\mathbf{y} \right)^\top$$

$$\preceq 8\cdot 3^6\langle \tilde{\mathbf{H}}, \mathbf{A} \rangle\tilde{\mathbf{H}}.$$

*Proof.* We only need to show that for every PSD matrices $\mathbf{A}$ and $\mathbf{B}$, it holds that

$$\langle \mathbf{B},\ \mathcal{L}\circ\mathbf{A} \rangle \leq 8\cdot 3^6\langle \tilde{\mathbf{H}}, \mathbf{A} \rangle\langle \tilde{\mathbf{H}}, \mathbf{B} \rangle.$$

This is because:

$$\langle \mathbf{B},\ \mathcal{L}\circ\mathbf{A} \rangle = \mathbb{E}\left\langle \mathbf{B},\ \left( \frac{1}{N}\mathbf{X}^\top\mathbf{y} \right)\left( \frac{1}{N}\mathbf{X}^\top\mathbf{y} \right)^\top \mathbf{A}\left( \frac{1}{N}\mathbf{X}^\top\mathbf{y} \right)\left( \frac{1}{N}\mathbf{X}^\top\mathbf{y} \right)^\top \right\rangle$$

$$= \mathbb{E}\left[ \left( \frac{1}{N}\mathbf{X}^\top\mathbf{y} \right)^\top \mathbf{B}\left( \frac{1}{N}\mathbf{X}^\top\mathbf{y} \right)\cdot\left( \frac{1}{N}\mathbf{X}^\top\mathbf{y} \right)^\top \mathbf{A}\left( \frac{1}{N}\mathbf{X}^\top\mathbf{y} \right) \right]$$

$$\leq \sqrt{\mathbb{E}\left( \left( \frac{1}{N}\mathbf{X}^\top\mathbf{y} \right)^\top \mathbf{B}\left( \frac{1}{N}\mathbf{X}^\top\mathbf{y} \right) \right)^2}\cdot\sqrt{\mathbb{E}\left( \left( \frac{1}{N}\mathbf{X}^\top\mathbf{y} \right)^\top \mathbf{A}\left( \frac{1}{N}\mathbf{X}^\top\mathbf{y} \right) \right)^2}$$

$$= \sqrt{\langle \mathbf{B},\ \mathcal{L}\circ\mathbf{B} \rangle}\cdot\sqrt{\langle \mathbf{A},\ \mathcal{L}\circ\mathbf{A} \rangle}$$

$$\leq 8\cdot 3^6\langle \tilde{\mathbf{H}}, \mathbf{A} \rangle\langle \tilde{\mathbf{H}}, \mathbf{B} \rangle,$$

where the last inequality is by Lemma D.6. $\qquad\square$

**Lemma D.8.** *For every PSD matrix $\mathbf{A}$, we have*

$$\mathbb{E}\left( y\mathbf{x}\left( \frac{1}{N}\mathbf{X}^\top\mathbf{y} \right)^\top \right)^{\otimes 2}\circ\mathbf{A} \preceq 9\langle \tilde{\mathbf{H}}, \mathbf{A} \rangle(\psi^2\texttt{tr}(\mathbf{H}) + \sigma^2)\mathbf{H}.$$

*Proof.* First, notice that

$$\mathbb{E}\left( y\mathbf{x}\left( \frac{1}{N}\mathbf{X}^\top\mathbf{y} \right)^\top \right)^{\otimes 2}\circ\mathbf{A}$$

$$= \mathbb{E}\left( \frac{1}{N}\mathbf{X}^\top\mathbf{y} \right)^\top \mathbf{A}\left( \frac{1}{N}\mathbf{X}^\top\mathbf{y} \right)y^2\mathbf{xx}^\top.$$

For the first factor, we take expectation with respect to $\mathbf{X}$ and $\epsilon$ (i.e., conditional on $\tilde{\boldsymbol{\beta}}$) to get

$$
\mathbb{E}\left(\frac{1}{N}\mathbf{X}^\top\mathbf{y}\right)^\top\mathbf{A}\left(\frac{1}{N}\mathbf{X}^\top\mathbf{y}\right) = \mathbb{E}\left(\frac{1}{N}\mathbf{X}^\top\mathbf{X}\tilde{\boldsymbol{\beta}} + \frac{1}{N}\mathbf{X}^\top\epsilon\right)^\top\mathbf{A}\left(\frac{1}{N}\mathbf{X}^\top\mathbf{X}\tilde{\boldsymbol{\beta}} + \frac{1}{N}\mathbf{X}^\top\epsilon\right)
$$

$$
= \frac{1}{N^2}\tilde{\boldsymbol{\beta}}^\top\mathbb{E}\mathbf{X}^\top\mathbf{X}\mathbf{A}\mathbf{X}^\top\mathbf{X}\tilde{\boldsymbol{\beta}} + \frac{1}{N^2}\mathbb{E}\epsilon^\top\mathbf{X}^\top\mathbf{A}\mathbf{X}\epsilon
$$

$$
= \tilde{\boldsymbol{\beta}}^\top\left(\frac{\langle\mathbf{H},\mathbf{A}\rangle}{N}\mathbf{H} + \frac{N+1}{N}\mathbf{H}\mathbf{A}\mathbf{H}\right)\tilde{\boldsymbol{\beta}} + \sigma^2\frac{\langle\mathbf{H},\mathbf{A}\rangle}{N}.
$$

Similarly, we compute the expectation of the second factor with respect to $\mathbf{x}$ and $\epsilon$ (i.e., conditional on $\tilde{\boldsymbol{\beta}}$) to get

$$
\mathbb{E}y^2\mathbf{x}\mathbf{x}^\top = \mathbb{E}\left(\mathbf{x}^\top\tilde{\boldsymbol{\beta}} + \epsilon\right)^2\mathbf{x}\mathbf{x}^\top
$$

$$
= \mathbb{E}\mathbf{x}\mathbf{x}^\top\tilde{\boldsymbol{\beta}}\tilde{\boldsymbol{\beta}}^\top\mathbf{x}\mathbf{x}^\top + \mathbb{E}\epsilon^2\mathbf{x}\mathbf{x}^\top
$$

$$
= \langle\mathbf{H},\tilde{\boldsymbol{\beta}}\tilde{\boldsymbol{\beta}}^\top\rangle\mathbf{H} + 2\mathbf{H}\tilde{\boldsymbol{\beta}}\tilde{\boldsymbol{\beta}}^\top\mathbf{H} + \sigma^2\mathbf{H}
$$

$$
\preceq \left(3\tilde{\boldsymbol{\beta}}^\top\mathbf{H}\tilde{\boldsymbol{\beta}} + \sigma^2\right)\mathbf{H}.
$$

Therefore, we have

$$
\mathbb{E}\left(y\mathbf{x}\left(\frac{1}{N}\mathbf{X}^\top\mathbf{y}\right)^\top\right)^{\otimes 2}\circ\mathbf{A}
$$

$$
= \mathbb{E}\left(\frac{1}{N}\mathbf{X}^\top\mathbf{y}\right)^\top\mathbf{A}\left(\frac{1}{N}\mathbf{X}^\top\mathbf{y}\right)y^2\mathbf{x}\mathbf{x}^\top
$$

$$
= \mathbb{E}\left(\tilde{\boldsymbol{\beta}}^\top\left(\frac{\langle\mathbf{H},\mathbf{A}\rangle}{N}\mathbf{H} + \frac{N+1}{N}\mathbf{H}\mathbf{A}\mathbf{H}\right)\tilde{\boldsymbol{\beta}} + \sigma^2\frac{\langle\mathbf{H},\mathbf{A}\rangle}{N}\right)\left(3\tilde{\boldsymbol{\beta}}^\top\mathbf{H}\tilde{\boldsymbol{\beta}} + \sigma^2\right)\mathbf{H}
$$

$$
\preceq \left(3\mathbb{E}\tilde{\boldsymbol{\beta}}^\top\left(\frac{\langle\mathbf{H},\mathbf{A}\rangle}{N}\mathbf{H} + \frac{N+1}{N}\mathbf{H}\mathbf{A}\mathbf{H}\right)\tilde{\boldsymbol{\beta}}\tilde{\boldsymbol{\beta}}^\top\mathbf{H}\tilde{\boldsymbol{\beta}} + 3\sigma^2\frac{\langle\mathbf{H},\mathbf{A}\rangle}{N}\mathbb{E}\tilde{\boldsymbol{\beta}}^\top\mathbf{H}\tilde{\boldsymbol{\beta}}\right.
$$

$$
\left. + \mathbb{E}\tilde{\boldsymbol{\beta}}^\top\left(\frac{\langle\mathbf{H},\mathbf{A}\rangle}{N}\mathbf{H} + \frac{N+1}{N}\mathbf{H}\mathbf{A}\mathbf{H}\right)\tilde{\boldsymbol{\beta}}\cdot\sigma^2 + \sigma^2\frac{\langle\mathbf{H},\mathbf{A}\rangle}{N}\cdot\sigma^2\right)\mathbf{H}
$$

$$
\preceq \left(9\psi^4\mathrm{tr}\left(\frac{\langle\mathbf{H},\mathbf{A}\rangle}{N}\mathbf{H} + \frac{N+1}{N}\mathbf{H}\mathbf{A}\mathbf{H}\right)\mathrm{tr}(\mathbf{H}) + 3\sigma^2\frac{\langle\mathbf{H},\mathbf{A}\rangle}{N}\psi^2\mathrm{tr}(\mathbf{H})\right.
$$

$$
\left. + \psi^2\mathrm{tr}\left(\frac{\langle\mathbf{H},\mathbf{A}\rangle}{N}\mathbf{H} + \frac{N+1}{N}\mathbf{H}\mathbf{A}\mathbf{H}\right)\cdot\sigma^2 + \sigma^2\frac{\langle\mathbf{H},\mathbf{A}\rangle}{N}\cdot\sigma^2\right)\mathbf{H}
$$

$$
\preceq \left(9\frac{\psi^4\mathrm{tr}(\mathbf{H})^2}{N}\langle\mathbf{H},\mathbf{A}\rangle + 9\frac{N+1}{N}\psi^4\mathrm{tr}(\mathbf{H})\langle\mathbf{H}^2,\mathbf{A}\rangle + 3\frac{\sigma^2\psi^2\mathrm{tr}(\mathbf{H})}{N}\langle\mathbf{H},\mathbf{A}\rangle\right.
$$

$$
\left. + \frac{\sigma^2\psi^2\mathrm{tr}(\mathbf{H})}{N}\langle\mathbf{H},\mathbf{A}\rangle + \frac{N+1}{N}\sigma^2\psi^2\langle\mathbf{H}^2,\mathbf{A}\rangle + \frac{\sigma^4}{N}\langle\mathbf{H},\mathbf{A}\rangle\right)\mathbf{H}
$$

$$
\preceq 9\left(\frac{(\psi^2\mathrm{tr}(\mathbf{H}) + \sigma^2)^2}{N}\langle\mathbf{H},\mathbf{A}\rangle + \frac{N+1}{N}\psi^2(\psi^2\mathrm{tr}(\mathbf{H}) + \sigma^2)\langle\mathbf{H}^2,\mathbf{A}\rangle\right)\mathbf{H}
$$

$$
= 9\left\langle\frac{\psi^2\mathrm{tr}(\mathbf{H}) + \sigma^2}{N}\mathbf{H} + \frac{N+1}{N}\psi^2\mathbf{H}^2,\ \mathbf{A}\right\rangle(\psi^2\mathrm{tr}(\mathbf{H}) + \sigma^2)\mathbf{H}
$$

$$
= 9\langle\tilde{\mathbf{H}},\mathbf{A}\rangle(\psi^2\mathrm{tr}(\mathbf{H}) + \sigma^2)\mathbf{H}.
$$

This completes the proof. $\qquad\square$

**Lemma D.9.** *For every PSD matrix* $\mathbf{A}$*, we have*

$$
\mathbb{E}\left(\mathbf{x}\mathbf{x}^\top\boldsymbol{\Gamma}^*\left(\frac{1}{N}\mathbf{X}^\top\mathbf{y}\right)\left(\frac{1}{N}\mathbf{X}^\top\mathbf{y}\right)^\top\right)^{\otimes 2}\circ\mathbf{A} \preceq 8\cdot 3^7\langle\tilde{\mathbf{H}},\mathbf{A}\rangle\psi^2\mathrm{tr}(\mathbf{H})\mathbf{H}.
$$

*Proof.* By definition, we have

$$\mathbb{E}\left(\mathbf{x}\mathbf{x}^\top \boldsymbol{\Gamma}^*\left(\frac{1}{N}\mathbf{X}^\top\mathbf{y}\right)\left(\frac{1}{N}\mathbf{X}^\top\mathbf{y}\right)^\top\right)^{\otimes 2}\circ\mathbf{A}$$

$$=\mathbb{E}\mathbf{x}\mathbf{x}^\top\boldsymbol{\Gamma}^*\left(\frac{1}{N}\mathbf{X}^\top\mathbf{y}\right)\left(\frac{1}{N}\mathbf{X}^\top\mathbf{y}\right)^\top\mathbf{A}\left(\frac{1}{N}\mathbf{X}^\top\mathbf{y}\right)\left(\frac{1}{N}\mathbf{X}^\top\mathbf{y}\right)^\top\boldsymbol{\Gamma}^*\mathbf{x}\mathbf{x}^\top$$

$$=\mathbb{E}\mathbf{x}\mathbf{x}^\top\boldsymbol{\Gamma}^*(\mathcal{L}\circ\mathbf{A})\boldsymbol{\Gamma}^*\mathbf{x}\mathbf{x}^\top$$

$$\preceq 3\langle\mathbf{H},\,\boldsymbol{\Gamma}^*(\mathcal{L}\circ\mathbf{A})\boldsymbol{\Gamma}^*\rangle\mathbf{H}$$

$$\preceq 8\cdot 3^7\langle\tilde{\mathbf{H}},\,\mathbf{A}\rangle\langle\mathbf{H},\,\boldsymbol{\Gamma}^*\tilde{\mathbf{H}}\boldsymbol{\Gamma}^*\rangle\mathbf{H},$$

where the last inequality is due to Lemma D.7. Recall from Theorem 3.1 that

$$\tilde{\mathbf{H}}:=(\boldsymbol{\Gamma}^*)^{-1}\cdot\psi^2\mathbf{H}$$

$$\boldsymbol{\Gamma}^*:=\left(\frac{\mathtt{tr}(\mathbf{H})+\sigma^2/\psi^2}{N}\mathbf{I}+\frac{N+1}{N}\mathbf{H}\right)^{-1}\preceq\mathbf{H}^{-1},$$

which implies that

$$\langle\mathbf{H},\,\boldsymbol{\Gamma}^*\tilde{\mathbf{H}}\boldsymbol{\Gamma}^*\rangle=\psi^2\mathtt{tr}(\mathbf{H}^2\boldsymbol{\Gamma}^*)$$

$$\leq\psi^2\mathtt{tr}(\mathbf{H}).$$

Bringing this back, we complete the proof. $\qquad\square$

**Lemma D.10** (Upper bound on $\mathcal{N}$). *Consider $\mathcal{N}$ defined in* (21). *For every PSD matrix $\mathbf{A}$, we have*

$$\mathcal{N}\circ\mathbf{A}=\mathbb{E}\boldsymbol{\Xi}\mathbf{A}\boldsymbol{\Xi}^\top$$

$$\preceq(16\cdot 3^7+18)(\psi^2\mathtt{tr}(\mathbf{H})+\sigma^2)\langle\tilde{\mathbf{H}},\mathbf{A}\rangle\mathbf{H}.$$

*Proof.* Note that

$$(\mathbf{A}+\mathbf{B})\mathbf{X}(\mathbf{A}+\mathbf{B})^\top\preceq 2\big(\mathbf{A}\mathbf{X}\mathbf{A}^\top+\mathbf{B}\mathbf{X}\mathbf{B}^\top\big).$$

So we have

$$\mathcal{N}\circ\mathbf{A}=\mathbb{E}\boldsymbol{\Xi}^{\otimes 2}\circ\mathbf{A}$$

$$=\mathbb{E}\left(\mathbf{x}\mathbf{x}^\top\boldsymbol{\Gamma}^*\left(\frac{1}{M}\mathbf{X}^\top\mathbf{y}\right)\left(\frac{1}{M}\mathbf{X}^\top\mathbf{y}\right)^\top-\mathbf{x}y\left(\frac{1}{M}\mathbf{X}^\top\mathbf{y}\right)^\top\right)^{\otimes 2}\circ\mathbf{A}$$

$$\preceq 2\mathbb{E}\left(\mathbf{x}\mathbf{x}^\top\boldsymbol{\Gamma}^*\left(\frac{1}{N}\mathbf{X}^\top\mathbf{y}\right)\left(\frac{1}{N}\mathbf{X}^\top\mathbf{y}\right)^\top\right)^{\otimes 2}\circ\mathbf{A}+2\mathbb{E}\left(y\mathbf{x}\left(\frac{1}{N}\mathbf{X}^\top\mathbf{y}\right)^\top\right)^{\otimes 2}\circ\mathbf{A}$$

$$\preceq 16\cdot 3^7\langle\tilde{\mathbf{H}},\mathbf{A}\rangle\psi^2\mathtt{tr}(\mathbf{H})\mathbf{H}+18\langle\tilde{\mathbf{H}},\mathbf{A}\rangle(\psi^2\mathtt{tr}(\mathbf{H})+\sigma^2)\mathbf{H}\qquad\text{by Lemmas D.8 and D.9}$$

$$\preceq(16\cdot 3^7+18)(\psi^2\mathtt{tr}(\mathbf{H})+\sigma^2)\langle\tilde{\mathbf{H}},\mathbf{A}\rangle\mathbf{H},$$

which completes the proof. $\qquad\square$

**Lemma D.11** (Lower bounds on $\mathcal{M}$ and $\mathcal{L}$). *For $\mathcal{M}$ defined in* (19) *and $\mathcal{L}$ defined in* (20), *we have*

$$\mathcal{M}\succeq\mathbf{H}\otimes\mathbf{H},$$

*and*

$$\mathcal{L}\succeq\tilde{\mathbf{H}}\otimes\tilde{\mathbf{H}}.$$

*Proof.* For every PSD matrix $\mathbf{A}$, we have

$$(\mathcal{M}-\mathbf{H}\otimes\mathbf{H})\circ\mathbf{A}=\mathbb{E}\mathbf{x}\mathbf{x}^\top\mathbf{H}\mathbf{x}\mathbf{x}^\top-\mathbf{H}\mathbf{A}\mathbf{H}$$

$$=\mathbb{E}\big(\mathbf{x}\mathbf{x}^\top-\mathbf{H}\big)\mathbf{A}\big(\mathbf{x}\mathbf{x}^\top-\mathbf{H}\big)$$

$$\succeq 0,$$

where the second equality is because
$$\mathbb{E}\mathbf{x}\mathbf{x}^\top = \mathbf{H}.$$

Similarly, for every PSD matrix $\mathbf{A}$, we have

$$
\begin{aligned}
\left(\mathcal{L} - \tilde{\mathbf{H}} \otimes \tilde{\mathbf{H}}\right) \circ \mathbf{A} &= \mathbb{E}\left(\frac{1}{N}\mathbf{X}^\top\mathbf{y}\right)\left(\frac{1}{N}\mathbf{X}^\top\mathbf{y}\right)^\top \mathbf{A}\left(\frac{1}{N}\mathbf{X}^\top\mathbf{y}\right)\left(\frac{1}{N}\mathbf{X}^\top\mathbf{y}\right)^\top - \tilde{\mathbf{H}}\mathbf{A}\tilde{\mathbf{H}} \\
&= \mathbb{E}\left(\left(\frac{1}{N}\mathbf{X}^\top\mathbf{y}\right)\left(\frac{1}{N}\mathbf{X}^\top\mathbf{y}\right)^\top - \tilde{\mathbf{H}}\right)\mathbf{A}\left(\left(\frac{1}{N}\mathbf{X}^\top\mathbf{y}\right)\left(\frac{1}{N}\mathbf{X}^\top\mathbf{y}\right)^\top - \tilde{\mathbf{H}}\right) \\
&\succeq 0,
\end{aligned}
$$

where the second equality is because

$$\mathbb{E}\left(\frac{1}{N}\mathbf{X}^\top\mathbf{y}\right)\left(\frac{1}{N}\mathbf{X}^\top\mathbf{y}\right)^\top = \tilde{\mathbf{H}}.$$

$\square$

**Lemma D.12** (Composition of PSD operators). *For every PSD operator $\mathcal{O}$, it holds that*

$$\mathbf{H}^{\otimes 2} \circ \mathcal{O} \circ \tilde{\mathbf{H}}^{\otimes 2} \preceq \mathcal{M} \circ \mathcal{O} \circ \mathcal{L} \preceq 8 \cdot 3^7 \langle \mathbf{H}, \mathcal{O} \circ \tilde{\mathbf{H}}\rangle \mathcal{S}^{(1)},$$

*where $\mathcal{S}^{(1)}$ is a PSD operator defined by*

$$\mathcal{S}^{(1)} := \langle \tilde{\mathbf{H}}, \cdot \rangle \mathbf{H}.$$

*As a direct consequence of the lower bound, we have*

$$\mathscr{S} \circ \mathcal{O} \succeq \mathscr{G} \circ \mathcal{O}.$$

*Proof.* For the upper bound, let us consider an arbitrary PSD matrix $\mathbf{A}$. We have

$$
\begin{aligned}
\mathcal{M} \circ \mathcal{O} \circ \mathcal{L} \circ \mathbf{A} &\preceq 8 \cdot 3^6 \langle \tilde{\mathbf{H}}, \mathbf{A}\rangle \mathcal{M} \circ \mathcal{O} \circ \tilde{\mathbf{H}} && \text{by Lemma D.7} \\
&\preceq 8 \cdot 3^7 \langle \tilde{\mathbf{H}}, \mathbf{A}\rangle \langle \mathbf{H}, \mathcal{O} \circ \tilde{\mathbf{H}}\rangle \mathbf{H} && \text{by Lemma D.5} \\
&= 8 \cdot 3^7 \langle \mathbf{H}, \mathcal{O} \circ \tilde{\mathbf{H}}\rangle \mathcal{S}^{(1)} \circ \mathbf{A}, && \text{by the definition of } \mathcal{S}^{(1)}
\end{aligned}
$$

which verifies the upper bound.

The lower bound is a direct consequence of Lemma D.11. $\square$

### D.4 Diagonalization

Without loss of generality, assume that $\mathbf{H}$ is diagonal. Let $\mathbb{D}$ be the set of PSD diagonal matrices. For a PSD operator $\mathcal{O}$, define its *diagonalization* by

$$
\begin{aligned}
\mathring{\mathcal{O}} : \mathbb{D} &\to \mathbb{D} \\
\mathbf{D} &\mapsto \texttt{diag}\{\mathcal{O} \circ \mathbf{D}\}
\end{aligned}
\tag{30}
$$

When the context is clear, we also write

$$\texttt{diag}\{\mathcal{O}\} := \mathring{\mathcal{O}}.$$

**Lemma D.13** (Diagnoalization of operators). *We have the following properties of diagonalization.*

1. *For every pair of operators $\mathcal{O}_1$ and $\mathcal{O}_2$ and for every scalar $a \in \mathbb{R}$, it holds that*

$$\texttt{diag}\{\mathcal{O}_1 + \mathcal{O}_2\} = \texttt{diag}\{\mathcal{O}_1\} + \texttt{diag}\{\mathcal{O}_2\}, \quad \texttt{diag}\{a\mathcal{O}_1\} = a\texttt{diag}\{\mathcal{O}_1\}.$$

2. *For two operators $\mathcal{O}_1$ and $\mathcal{O}_2$ such that $\mathcal{O}_1 \preceq \mathcal{O}_2$, it holds that*

$$\texttt{diag}\{\mathcal{O}_1\} \preceq \texttt{diag}\{\mathcal{O}_2\}.$$

3. *For every operator $\mathcal{O}$, it holds that*

$$\mathrm{diag}\{\mathscr{G}(\mathcal{O})\} = \mathscr{G}(\mathring{\mathcal{O}}).$$

*Proof.* It should be clear. We only prove the last claim.

Let $\mathbf{K}$ be a PSD diagonal matrix. By (23), we have

$$\mathscr{G}(\mathcal{O}) \circ \mathbf{K} = \mathcal{O} \circ \mathbf{K} - \gamma\big(\mathbf{H}\mathcal{O} \circ (\tilde{\mathbf{H}}\mathbf{K}) + \mathcal{O} \circ (\mathbf{K}\tilde{\mathbf{H}})\mathbf{H}\big) + \gamma^2\mathbf{H}\mathcal{O} \circ (\tilde{\mathbf{H}}\mathbf{K}\tilde{\mathbf{H}})\mathbf{H}.$$

Now taking a diagonal on both sides and using that $\mathbf{K}$ is also diagonal, we obtain that

$$\begin{aligned}
\mathrm{diag}\{\mathscr{G}(\mathcal{O}) \circ \mathbf{K}\} &= \mathrm{diag}\{\mathcal{O} \circ \mathbf{K}\} - \gamma\Big(\mathrm{diag}\{\mathbf{H}\mathcal{O} \circ (\tilde{\mathbf{H}}\mathbf{K})\} + \mathrm{diag}\{\mathcal{O} \circ (\mathbf{K}\tilde{\mathbf{H}})\mathbf{H}\}\Big) \\
&\quad + \gamma^2\mathrm{diag}\{\mathbf{H}\mathcal{O} \circ (\tilde{\mathbf{H}}\mathbf{K}\tilde{\mathbf{H}})\mathbf{H}\} \\
&= \mathring{\mathcal{O}} \circ \mathbf{K} - \gamma\big(\mathbf{H}\mathring{\mathcal{O}} \circ (\tilde{\mathbf{H}}\mathbf{K}) + \mathring{\mathcal{O}} \circ (\mathbf{K}\tilde{\mathbf{H}})\mathbf{H}\big) \\
&\quad + \gamma^2\mathbf{H}\mathring{\mathcal{O}} \circ (\tilde{\mathbf{H}}\mathbf{K}\tilde{\mathbf{H}})\mathbf{H} \\
&= \mathscr{G}(\mathring{\mathcal{O}}) \circ \mathbf{K},
\end{aligned}$$

which implies that

$$\mathrm{diag}\{\mathscr{G}(\mathcal{O})\} = \mathscr{G}(\mathring{\mathcal{O}}).$$

$\square$

**Bias and variance error under operator diagonalization.** Since both $\mathbf{H}$ and $\tilde{\mathbf{H}}$ are diagonal matrices, we have

$$\begin{aligned}
\langle\mathbf{H}, \mathcal{B}_T \circ \tilde{\mathbf{H}}\rangle &= \langle\mathbf{H}, \mathring{\mathcal{B}}_T \circ \tilde{\mathbf{H}}\rangle, \\
\langle\mathbf{H}, \mathcal{C}_T \circ \tilde{\mathbf{H}}\rangle &= \langle\mathbf{H}, \mathring{\mathcal{C}}_T \circ \tilde{\mathbf{H}}\rangle,
\end{aligned}$$

which motivates us to control only the diagonalized bias and variance iterates. We next establish recursions about the diagonalized bias and variance iterates, respectively.

**Diagonalization of the bias iterates.** Consider the bias iterates given by (24). By definition of $\mathscr{S}$ in (22) and $\mathscr{G}$ in (23), we have

$$\begin{aligned}
\mathcal{B}_t &= \mathscr{S}_t \circ \mathcal{B}_{t-1} && \text{by (24)} \\
&= \mathscr{G}_t \circ \mathcal{B}_{t-1} + \gamma_t^2\mathcal{M} \circ \mathcal{B}_{t-1} \circ \mathcal{L} - \gamma_t^2\mathbf{H}^{\otimes 2} \circ \mathcal{B}_{t-1} \circ \tilde{\mathbf{H}}^{\otimes 2} && \text{by (22) and (23)} \\
&\preceq \mathscr{G}_t \circ \mathcal{B}_{t-1} + \gamma_t^2\mathcal{M} \circ \mathcal{B}_{t-1} \circ \mathcal{L} && \text{since } \mathcal{B}_{t-1} \text{ is PSD} \\
&\preceq \mathscr{G}_t \circ \mathcal{B}_{t-1} + \gamma_t^2 8 \cdot 3^7\langle\mathbf{H}, \mathcal{B}_{t-1} \circ \tilde{\mathbf{H}}\rangle\mathcal{S}^{(1)} && \text{by Lemma D.12} \\
&= \mathscr{G}_t \circ \mathcal{B}_{t-1} + \gamma_t^2 8 \cdot 3^7\langle\mathbf{H}, \mathring{\mathcal{B}}_{t-1} \circ \tilde{\mathbf{H}}\rangle\mathcal{S}^{(1)},
\end{aligned}$$

where the last equality is because both $\mathbf{H}$ and $\tilde{\mathbf{H}}$ are diagonal. Next, taking diagonal on both sides and using Lemma D.13, we have

$$\begin{aligned}
\mathring{\mathcal{B}}_t &\preceq \mathrm{diag}\{\mathscr{G}_t \circ \mathcal{B}_{t-1}\} + \gamma_t^2 \cdot 8 \cdot 3^7\langle\mathbf{H}, \mathring{\mathcal{B}}_{t-1} \circ \tilde{\mathbf{H}}\rangle\mathcal{S}^{(1)} && \text{by Lemma D.13} \\
&= \mathscr{G}_t \circ \mathring{\mathcal{B}}_{t-1} + \gamma_t^2 \cdot 8 \cdot 3^7\langle\mathbf{H}, \mathring{\mathcal{B}}_{t-1} \circ \tilde{\mathbf{H}}\rangle\mathcal{S}^{(1)}, && \text{by Lemma D.13} \quad (31)
\end{aligned}$$

where

$$\mathring{\mathcal{B}}_0 = \mathrm{diag}\{(\boldsymbol{\Gamma}_0 - \boldsymbol{\Gamma}^*)^{\otimes 2}\}.$$

We have obtained a recursion about the diagonalized bias iterates.

**Diagonalization of the variance iterates.** Similarly, let us treat the variance iterates given by (25). By repeating the argument for the bias iterate, we have

$$\begin{aligned}
\mathcal{C}_t &= \mathscr{S}_t \circ \mathcal{C}_{t-1} + \gamma_t^2\mathcal{N} \\
&\preceq \mathscr{G}_t \circ \mathcal{C}_{t-1} + \gamma_t^2 \cdot 8 \cdot 3^7\langle\mathbf{H}, \mathring{\mathcal{C}}_{t-1} \circ \tilde{\mathbf{H}}\rangle\mathcal{S}^{(1)} + \gamma_t^2\mathcal{N}.
\end{aligned}$$

Using Lemma D.10, we have

$$\mathcal{N} \preceq (16 \cdot 3^7 + 18)(\psi^2 \mathtt{tr}(\mathbf{H}) + \sigma^2)\mathcal{S}^{(1)}.$$

So we have

$$\mathcal{C}_t \preceq \mathscr{G}_t \circ \mathcal{C}_{t-1} + \gamma_t^2 \cdot 8 \cdot 3^7 \langle \mathbf{H}, \mathring{\mathcal{C}}_{t-1} \circ \tilde{\mathbf{H}} \rangle \mathcal{S}^{(1)} + \gamma_t^2 (16 \cdot 3^7 + 18)(\psi^2 \mathtt{tr}(\mathbf{H}) + \sigma^2)\mathcal{S}^{(1)}.$$

Similar to the treatment to the bias iterate, we take diagonalization on both sides and apply Lemma D.13, then we have

$$\mathring{\mathcal{C}}_t \preceq \mathscr{G}_t \circ \mathring{\mathcal{C}}_{t-1} + \gamma_t^2 \cdot 8 \cdot 3^7 \langle \mathbf{H}, \mathring{\mathcal{C}}_{t-1} \circ \tilde{\mathbf{H}} \rangle \mathcal{S}^{(1)} + \gamma_t^2 (16 \cdot 3^7 + 18)(\psi^2 \mathtt{tr}(\mathbf{H}) + \sigma^2)\mathcal{S}^{(1)}, \quad (32)$$

where

$$\mathring{\mathcal{C}}_0 = \mathbf{0} \otimes \mathbf{0}.$$

We have established the recursion about the diagonalized variance iterates.

**Monotonicity and contractivity of $\mathscr{G}$ on diagonal PSD operators.** Finally, we introduce the following important lemma, which shows that $\mathscr{G}$ is monotone when applied to diagonal operators.

**Lemma D.14** (Diagonalization of $\mathscr{G}$). *We have the following about the $\mathscr{G}$ defined in* (23).

1. *For every diagonal operator $\mathcal{D}$ and every diagonal matrix $\mathbf{K}$, it holds that*

$$\mathscr{G}(\mathcal{D}) \circ \mathbf{K} = \mathcal{D} \circ \mathbf{K} - 2\gamma \mathbf{H}\mathcal{D} \circ (\tilde{\mathbf{H}}\mathbf{K}) + \gamma^2 \mathbf{H}^2 \mathcal{D} \circ (\tilde{\mathbf{H}}^2 \mathbf{K}).$$

2. *Suppose that*

$$0 < \gamma \leq \frac{1}{2\mathtt{tr}(\mathbf{H})\mathtt{tr}(\tilde{\mathbf{H}})},$$

*then $\mathscr{G}$ is an* increasing *map on the diagonal operators. That is, for every pair of diagonal operators such that*

$$\mathcal{D}_1 \preceq \mathcal{D}_2,$$

*we have*

$$\mathscr{G}(\mathcal{D}_1) \preceq \mathscr{G}(\mathcal{D}_2).$$

3. *Suppose that*

$$0 < \gamma \leq \frac{1}{2\mathtt{tr}(\mathbf{H})\mathtt{tr}(\tilde{\mathbf{H}})},$$

*then $\mathscr{G}$ is a* contractive *map on the diagonal operators. That is, for every diagonal PSD operator*

$$\mathcal{D} \succeq 0,$$

*we have*

$$\mathscr{G}(\mathcal{D}) \preceq \mathcal{D}.$$

*Proof.* The first claim is clear from the definitions:

$$\begin{aligned} \mathscr{G}(\mathcal{D}) \circ \mathbf{K} &= \mathcal{D} \circ \mathbf{K} - \gamma \big( \mathbf{H}\mathcal{D} \circ (\tilde{\mathbf{H}}\mathbf{K}) + \mathcal{D} \circ (\mathbf{K}\tilde{\mathbf{H}})\mathbf{H} \big) \\ &\quad + \gamma^2 \mathbf{H}\mathcal{D} \circ (\tilde{\mathbf{H}}\mathbf{K}\tilde{\mathbf{H}})\mathbf{H} \\ &= \mathcal{D} \circ \mathbf{K} - 2\gamma \mathbf{H}\mathcal{D} \circ (\tilde{\mathbf{H}}\mathbf{K}) + \gamma^2 \mathbf{H}^2 \mathcal{D} \circ (\tilde{\mathbf{H}}^2 \mathbf{K}). \end{aligned}$$

For showing the second claim, notice that, by the linearity of $\mathscr{G}$, we only need to verify that for every diagonal PSD operator $\mathcal{D}$, it holds that

$$\mathscr{G}(\mathcal{D}) \succeq 0.$$

By definition, we only need to show that for every diagonal PSD matrix $\mathbf{K}$, it holds that

$$\mathscr{G}(\mathcal{D}) \circ \mathbf{K} \succeq 0.$$

We lower bound the left-hand side using the first conclusion:

$$\mathscr{G}(\mathcal{D}) \circ \mathbf{K} = \mathcal{D} \circ \mathbf{K} - 2\gamma \mathbf{H}\mathcal{D} \circ (\tilde{\mathbf{H}}\mathbf{K}) + \gamma^2 \mathbf{H}^2 \mathcal{D} \circ (\tilde{\mathbf{H}}^2 \mathbf{K})$$

$$\succeq \mathcal{D} \circ \mathbf{K} - 2\gamma \mathbf{H}\mathcal{D} \circ (\tilde{\mathbf{H}}\mathbf{K})$$
$$\succeq \mathcal{D} \circ \mathbf{K} - 2\gamma \mathtt{tr}(\mathbf{H})\mathbf{I}\mathcal{D} \circ (\mathtt{tr}(\tilde{\mathbf{H}})\mathbf{K})$$
$$= \big(1 - 2\gamma \mathtt{tr}(\mathbf{H})\mathtt{tr}(\tilde{\mathbf{H}})\big)\mathcal{D} \circ \mathbf{K}$$
$$\succeq 0.$$

Similarly, we can prove the last claim by showing that

$$\mathscr{G}(\mathcal{D}) \circ \mathbf{K} = \mathcal{D} \circ \mathbf{K} - 2\gamma \mathbf{H}\mathcal{D} \circ (\tilde{\mathbf{H}}\mathbf{K}) + \gamma^2 \mathbf{H}^2 \mathcal{D} \circ (\tilde{\mathbf{H}}^2 \mathbf{K})$$
$$\preceq \mathcal{D} \circ \mathbf{K} - 2\gamma \mathbf{H}\mathcal{D} \circ (\tilde{\mathbf{H}}\mathbf{K}) + \gamma^2 \mathtt{tr}(\mathbf{H})\mathtt{tr}(\tilde{\mathbf{H}})\mathbf{H}\mathcal{D} \circ (\tilde{\mathbf{H}}\mathbf{K})$$
$$\preceq \mathcal{D} \circ \mathbf{K} - \gamma \mathbf{H}\mathcal{D} \circ (\tilde{\mathbf{H}}\mathbf{K})$$
$$\preceq \mathcal{D} \circ \mathbf{K}.$$

We have completed the proof. $\square$

### D.5 OPERATOR POLYNOMIALS

In this section, we develop several useful new tools for computing the diagonal bias and variance iterates, (31) and (32).

**Operator polynomials.** We first introduce *operator polynomials*.

**Definition 4** (Operator monomials). Define a sequence of *operator monomials*:

$$\mathcal{S}^{(t)} := \langle \tilde{\mathbf{H}}^t, \cdot \rangle \mathbf{H}^t, \quad t \in \mathbb{N}.$$

That is, for every $t \in \mathbb{N}$ and for every symmetric matrix $\mathbf{K}$,

$$\mathcal{S}^{(t)} \circ \mathbf{K} := \langle \tilde{\mathbf{H}}^t, \mathbf{K} \rangle \mathbf{H}^t.$$

Denote the set of all operator monomials by

$$\mathbb{S} := \{\mathcal{S}^{(i)} : i \in \mathbb{N}\}.$$

**Definition 5** (Operator polynomials). Let "$\bullet$" be a multiplication operation on $\mathbb{S}$, defined by

$$\mathcal{S}^{(i)} \bullet \mathcal{S}^{(j)} := \mathcal{S}^{(i+j)}, \quad i, j \in \mathbb{N}.$$

Let "$+$" be the canonical operator addition operation. Let "$\bullet$" distribute over "$+$" in the canonical manner, i.e.,

$$\mathcal{S}^{(i)} \bullet (\mathcal{S}^{(j)} + \mathcal{S}^{(k)}) := \mathcal{S}^{(i)} \bullet \mathcal{S}^{(j)} + \mathcal{S}^{(i)} \bullet \mathcal{S}^{(k)} = \mathcal{S}^{(i+j)} + \mathcal{S}^{(i+k)}.$$

It is straightforward to verify that $\mathcal{S}^{(0)}$ is the identity element under "$\bullet$", $0 \in \mathbb{R}^{d^2 \times d^2}$ is the zero element under "$+$". We define a set of operator polynomials by

$$\big(\mathcal{S}^{(0)} - \gamma \mathcal{S}^{(1)}\big)^{\bullet t} := \sum_{k=0}^{t} \binom{t}{k} (-\gamma)^k \mathcal{S}^{(k)}, \quad t \in \mathbb{N}, \ \gamma \in \mathbb{R}_+.$$

When the context is clear, we also use "$\prod$" to refer to a sequence of multiplication operations among the operator polynomials, e.g.,

$$\prod_{k=1}^{t} \big(\mathcal{S}^{(0)} - \gamma_k \mathcal{S}^{(1)}\big)^{\bullet 2} := \big(\mathcal{S}^{(0)} - \gamma_t \mathcal{S}^{(1)}\big)^{\bullet 2} \bullet \big(\mathcal{S}^{(0)} - \gamma_{t-1}\mathcal{S}^{(1)}\big)^{\bullet 2} \bullet \cdots \bullet \big(\mathcal{S}^{(0)} - \gamma_1 \mathcal{S}^{(1)}\big)^{\bullet 2},$$

where $(\gamma_k)_{k=1}^{t}$ refers a sequence of positive stepsize.

The following lemma allows us to represent the composition of $\mathscr{G}$ over operator monomials as operator polynomials.

**Lemma D.15** (Operator polynomials). *We have the following results regarding the composition of operator monomials and other operators.*

1. *For $t \geq 0$,*

$$(\mathbf{H} \otimes \mathbf{I}) \circ \mathcal{S}^{(t)} \circ (\tilde{\mathbf{H}} \otimes \mathbf{I}) = (\mathbf{I} \otimes \mathbf{H}) \circ \mathcal{S}^{(t)} \circ (\mathbf{I} \otimes \tilde{\mathbf{H}}) = \mathcal{S}^{(t+1)}.$$

2. *For $t \geq 0$,*

$$\mathbf{H}^{\otimes 2} \circ \mathcal{S}^{(t)} \circ \tilde{\mathbf{H}}^{\otimes 2} = \mathcal{S}^{(t+2)}.$$

3. *For $t \geq 0$,*

$$\mathscr{G}^t(\mathcal{S}^{(1)}) = \left(\mathcal{S}^{(0)} - \gamma \mathcal{S}^{(1)}\right)^{\bullet 2t} \bullet \mathcal{S}^{(1)}.$$

4. *For $t \geq 0$,*

$$\left(\prod_{k=1}^{t} \mathscr{G}_k\right)(\mathcal{S}^{(1)}) = \prod_{k=1}^{t} \left(\mathcal{S}^{(0)} - \gamma_k \mathcal{S}^{(1)}\right)^{\bullet 2} \bullet \mathcal{S}^{(1)}.$$

*Proof.* We now prove each claim respectively.

1. We consider a symmetric matrix $\mathbf{K}$ and notice that

$$
\begin{aligned}
&(\mathbf{H} \otimes \mathbf{I}) \circ \mathcal{S}^{(t)} \circ (\tilde{\mathbf{H}} \otimes \mathbf{I}) \circ \mathbf{K} \\
&= (\mathbf{H} \otimes \mathbf{I}) \circ \mathcal{S}^{(t)} \circ (\mathbf{K}\tilde{\mathbf{H}}) \\
&= \langle \tilde{\mathbf{H}}^t, \mathbf{K}\tilde{\mathbf{H}} \rangle (\mathbf{H} \otimes \mathbf{I}) \circ \mathbf{H}^t \\
&= \langle \tilde{\mathbf{H}}^t, \mathbf{K}\tilde{\mathbf{H}} \rangle \mathbf{H}^{t+1} \\
&= \langle \tilde{\mathbf{H}}^{t+1}, \mathbf{K} \rangle \mathbf{H}^{t+1} \\
&= \mathcal{S}^{(t+1)} \circ \mathbf{K}.
\end{aligned}
$$

Similarly, we have

$$(\mathbf{I} \otimes \mathbf{H}) \circ \mathcal{S}^{(t)} \circ (\mathbf{I} \otimes \tilde{\mathbf{H}}) \circ \mathbf{K} = \mathcal{S}^{(t+1)} \circ \mathbf{K}.$$

These verify the first claim.

2. We consider a symmetric matrix $\mathbf{K}$ and notice that

$$
\begin{aligned}
&(\mathbf{H} \otimes \mathbf{H}) \circ \mathcal{S}^{(t)} \circ (\tilde{\mathbf{H}} \otimes \tilde{\mathbf{H}}) \circ \mathbf{K} \\
&= (\mathbf{H} \otimes \mathbf{H}) \circ \mathcal{S}^{(t)} \circ (\tilde{\mathbf{H}}\mathbf{K}\tilde{\mathbf{H}}) \\
&= \langle \tilde{\mathbf{H}}^t, \tilde{\mathbf{H}}\mathbf{K}\tilde{\mathbf{H}} \rangle (\mathbf{H} \otimes \mathbf{H}) \circ \mathbf{H}^t \\
&= \langle \tilde{\mathbf{H}}^{t+2}, \mathbf{K} \rangle \mathbf{H}^{t+2} \\
&= \mathcal{S}^{(t+2)} \circ \mathbf{K},
\end{aligned}
$$

which verifies the second claim.

3. Using the firs two claims and (23), we have

$$
\begin{aligned}
\mathscr{G}(\mathcal{S}^{(t)}) &= \mathcal{S}^{(t)} - \gamma \Big( (\mathbf{H} \otimes \mathbf{I}) \circ \mathcal{S}^{(t)} \circ (\tilde{\mathbf{H}} \otimes \mathbf{I}) + (\mathbf{I} \otimes \mathbf{H}) \circ \mathcal{S}^{(t)} \circ (\mathbf{I} \otimes \tilde{\mathbf{H}}) \Big) \\
&\quad + \gamma^2 \mathbf{H}^{\otimes 2} \circ \mathcal{S}^{(t)} \circ \tilde{\mathbf{H}}^{\otimes 2} \\
&= \mathcal{S}^{(t)} - 2\gamma \mathcal{S}^{(t+1)} + \gamma^2 \mathcal{S}^{(t+2)} \\
&=: \left(\mathcal{S}^{(0)} - \gamma \mathcal{S}^{(1)}\right)^{\bullet 2} \bullet \mathcal{S}^{(t)}.
\end{aligned}
$$

Recursively applying the above equation and using the operator polynomials notation (see Definition 5), we get

$$
\begin{aligned}
\mathscr{G}^0(\mathcal{S}^{(1)}) &= \mathcal{S}^{(1)}, \\
\mathscr{G}(\mathcal{S}^{(1)}) &= \left(\mathcal{S}^{(0)} - \gamma \mathcal{S}^{(1)}\right)^{\bullet 2} \bullet \mathcal{S}^{(1)}, \\
\mathscr{G}^2(\mathcal{S}^{(1)}) &= \left(\mathcal{S}^{(0)} - \gamma \mathcal{S}^{(1)}\right)^{\bullet 4} \bullet \mathcal{S}^{(1)},
\end{aligned}
$$

$$\vdots$$
$$\mathscr{G}^t(\mathcal{S}^{(1)}) = \left(\mathcal{S}^{(0)} - \gamma\mathcal{S}^{(1)}\right)^{\bullet 2t} \bullet \mathcal{S}^{(1)}.$$

This verifies the third claim.

4. The fourth claim can be verified similarly to the third claim.

We have completed the proof. $\qquad\square$

**Computing operator polynomials.** We now introduce a method to compute operator polynomials.

Notice that we only need to deal with diagonal PSD operators. Since a diagonal PSD matrix has $d$ degrees of freedom, which can be equivalently represented by a $d$-dimensional (non-negative) vector. Similarly, a diagonal operator has $d \times d$ degrees of freedom and thus can be equivalently represented as a linear map on $d$-dimensional (non-negative) vectors.

Define a *matrixization* operation as

$$\mathtt{mat} : \mathbb{R}^d \to \mathbb{R}^{d\times d}$$

$$\mathbf{k} \mapsto \mathtt{mat}(\mathbf{k}) := \begin{pmatrix} \mathbf{k}_1 & & \\ & \ddots & \\ & & \mathbf{k}_d \end{pmatrix}.$$

Then the operator monomial on diagonal PSD matrices can be equivalently written as

$$\mathcal{S}^{(t)} : \mathbb{D} \to \mathbb{D}$$
$$\mathtt{mat}\{\mathbf{v}\} \mapsto \langle \tilde{\mathbf{H}}^t, \mathtt{mat}\{\mathbf{v}\}\rangle \mathbf{H}^t = \mathtt{mat}\Big\{\mathbf{h}^{\odot t}\big(\tilde{\mathbf{h}}^{\odot t}\big)^\top \mathbf{k}\Big\}, \tag{33}$$

where "$\odot$" refers to Hadamard product (i.e., entry-wise product) and $\mathbf{h}$ and $\tilde{\mathbf{h}}$ are the diagonals of $\mathbf{H}$ and $\tilde{\mathbf{H}}$, respectively, that is,

$$\mathbf{h} := \begin{pmatrix} \mathbf{H}_{11} \\ \vdots \\ \mathbf{H}_{dd} \end{pmatrix}, \qquad \tilde{\mathbf{h}} := \begin{pmatrix} \tilde{\mathbf{H}}_{11} \\ \vdots \\ \tilde{\mathbf{H}}_{dd} \end{pmatrix}. \tag{34}$$

This viewpoint allows us to compute operator polynomials. In particular, we can prove the following results.

**Lemma D.16.** *When restricted as a diagonal operator, we have the following*

1. *For every $t \geq 0$ and every $\mathbf{v} \in \mathbb{R}^d$,*

$$\left(\left(\mathcal{S}^{(0)} - \gamma\mathcal{S}^{(1)}\right)^{\bullet t} \bullet \mathcal{S}^{(1)}\right) \circ \mathtt{mat}\{\mathbf{v}\} = \mathtt{mat}\Big\{\Big(\big(\mathbf{J} - \gamma\mathbf{h}\tilde{\mathbf{h}}^\top\big)^{\odot t} \odot \big(\mathbf{h}\tilde{\mathbf{h}}^\top\big)\Big)\mathbf{v}\Big\},$$

*where $\mathbf{J}$ refers to the "all-one" matrix, that is,*

$$\mathbf{J} = \mathbf{1}\mathbf{1}^\top.$$

2. *For every $t \geq 0$ and every $\mathbf{v} \in \mathbb{R}^d$,*

$$\left(\prod_{k=1}^t \left(\mathcal{S}^{(0)} - \gamma_k\mathcal{S}^{(1)}\right)^{\bullet 2} \bullet \mathcal{S}^{(1)}\right) \circ \mathtt{mat}\{\mathbf{v}\} = \mathtt{mat}\Big\{\Big(\prod_{k=1}^t \big(\mathbf{J} - \gamma_k\mathbf{h}\tilde{\mathbf{h}}^\top\big)^{\odot 2} \odot \big(\mathbf{h}\tilde{\mathbf{h}}^\top\big)\Big)\mathbf{v}\Big\}.$$

*Proof.* By Definition 5, we have

$$\left(\mathcal{S}^{(0)} - \gamma\mathcal{S}^{(1)}\right)^{\bullet t} \bullet \mathcal{S}^{(1)} := \sum_{k=0}^t \binom{t}{k}(-\gamma)^k \mathcal{S}^{k+1}.$$

Now using (33), we have

$$
\left(\left(\mathcal{S}^{(0)} - \gamma \mathcal{S}^{(1)}\right)^{\bullet t} \bullet \mathcal{S}^{(1)}\right) \circ \mathtt{mat}\{\mathbf{v}\} = \sum_{k=0}^{t} \binom{t}{k} (-\gamma)^k \mathcal{S}^{k+1} \circ \mathtt{mat}\{\mathbf{v}\}
$$
$$
= \sum_{k=0}^{t} \binom{t}{k} (-\gamma)^k \mathtt{mat}\left\{ \mathbf{h}^{\odot k+1} \big( \tilde{\mathbf{h}}^{\odot k+1} \big)^{\top} \mathbf{k} \right\}
$$
$$
= \mathtt{mat}\left\{ \left( \sum_{k=0}^{t} \binom{t}{k} (-\gamma)^k \mathbf{h}^{\odot k+1} \big( \tilde{\mathbf{h}}^{\odot k+1} \big)^{\top} \right) \mathbf{k} \right\}
$$
$$
= \mathtt{mat}\left\{ \left( \big( \mathbf{J} - \gamma \mathbf{h} \tilde{\mathbf{h}}^{\top} \big)^{\odot t} \odot \big( \mathbf{h} \tilde{\mathbf{h}}^{\top} \big) \right) \mathbf{v} \right\},
$$

which verifies the first claim. The second claim can be verified in the same way.  □

### D.6  VARIANCE ERROR ANALYSIS

We first show a crude variance upper bound.

**Lemma D.17** (A crude variance bound). *Suppose that*

$$
\gamma_0 \le \frac{1}{16 \cdot 3^7 \mathtt{tr}(\mathbf{H}) \mathtt{tr}(\tilde{\mathbf{H}})}.
$$

*Then for* (32)*, we have*

$$
\mathring{\mathcal{C}}_t \preceq c \gamma_0 \mathcal{S}^{(0)}, \quad t \ge 0,
$$

*where*

$$
c := (32 \cdot 3^7 + 36)\big(\psi^2 \mathtt{tr}(\mathbf{H}) + \sigma^2\big).
$$

*Proof.* We prove the claim by induction. For $t = 0$, the claim holds since

$$
\mathring{\mathcal{C}}_0 = \mathbf{0} \otimes \mathbf{0} \preceq c\gamma_0 \mathcal{S}^{(0)}.
$$

Now suppose that

$$
\mathring{\mathcal{C}}_{t-1} \preceq c\gamma_0 \mathcal{S}^{(0)}.
$$

Let us compute $\mathring{\mathcal{C}}_t$ by (32):

$$
\mathring{\mathcal{C}}_t \preceq \mathscr{G}_t \circ \mathring{\mathcal{C}}_{t-1} + \gamma_t^2 \cdot 8 \cdot 3^7 \langle \mathbf{H}, \mathring{\mathcal{C}}_{t-1} \circ \tilde{\mathbf{H}} \rangle \mathcal{S}^{(1)} + \gamma_t^2 (16 \cdot 3^7 + 18)(\psi^2 \mathtt{tr}(\mathbf{H}) + \sigma^2) \mathcal{S}^{(1)}
$$
$$
= \mathscr{G}_t \circ \mathring{\mathcal{C}}_{t-1} + \gamma_t^2 \cdot 8 \cdot 3^7 \langle \mathbf{H}, \mathring{\mathcal{C}}_{t-1} \circ \tilde{\mathbf{H}} \rangle \mathcal{S}^{(1)} + \gamma_t^2 \frac{c}{2} \mathcal{S}^{(1)} \qquad \text{by the definition of } c
$$
$$
\preceq \mathscr{G}_t (c\gamma_0 \mathcal{S}^{(0)}) + c\gamma_0 \gamma_t^2 \cdot 8 \cdot 3^7 \langle \mathbf{H}, \mathcal{S}^{(0)} \circ \tilde{\mathbf{H}} \rangle \mathcal{S}^{(1)} + \gamma_t^2 \frac{c}{2} \mathcal{S}^{(1)}
$$
$$
\qquad \text{by the induction hypothesis}
$$
$$
\preceq \mathscr{G}_t (c\gamma_0 \mathcal{S}^{(0)}) + \gamma_t^2 \frac{c}{2} \mathcal{S}^{(1)} + \gamma_t^2 \frac{c}{2} \mathcal{S}^{(1)}
$$
$$
\qquad \text{by the definition of } \mathcal{S}^{(0)} \text{ and the choice of } \gamma_0
$$
$$
= c\gamma_0 \mathscr{G}_t (\mathcal{S}^{(0)}) + c\gamma_t^2 \mathcal{S}^{(1)} \qquad\qquad \text{since } \mathscr{G}_t \text{ is linear}
$$
$$
= c\gamma_0 \big( \mathcal{S}^{(0)} - \gamma_t \mathcal{S}^{(1)} \big)^{\bullet 2} + c\gamma_t^2 \mathcal{S}^{(1)} \qquad\qquad \text{by Lemma D.15}
$$
$$
= c\gamma_0 \big( \mathcal{S}^{(0)} - 2\gamma_t \mathcal{S}^{(1)} + \gamma_t^2 \mathcal{S}^{(2)} \big) + c\gamma_t^2 \mathcal{S}^{(1)} \qquad \text{by Definition 5}
$$
$$
\preceq c\gamma_0 \big( \mathcal{S}^{(0)} - \gamma_t \mathcal{S}^{(1)} \big) + c\gamma_t^2 \mathcal{S}^{(1)} \qquad\qquad \text{since } \gamma_0 \mathcal{S}^{(2)} \preceq \mathcal{S}^{(1)}
$$
$$
\preceq c\gamma_0 \mathcal{S}^{(0)}. \qquad\qquad\qquad \text{since } \gamma_t \le \gamma_0
$$

This completes the induction.  □

We next show a sharper variance bound.

**Lemma D.18** (A sharp bound on the variance iterate). *Suppose that*

$$\gamma_0 \le \frac{1}{16 \cdot 3^7 \texttt{tr}(\mathbf{H})\texttt{tr}(\tilde{\mathbf{H}})}.$$

*For every entry-wise non-negative vector $\mathbf{v} \in \mathbb{R}^d$, we have*

$$\mathring{\mathcal{C}}_T \circ \texttt{mat}\{\mathbf{v}\} \preceq \texttt{cmat}\Big\{\Big(f(\gamma_0 \mathbf{h}\tilde{\mathbf{h}}^\top) \odot (\mathbf{h}\tilde{\mathbf{h}}^\top)^{\odot -1}\Big)\mathbf{v}\Big\},$$

*where*

$$f(x) := \sum_{\ell=0}^{L-1} \frac{x}{2^\ell}\left(1 - \left(1 - \frac{x}{2^\ell}\right)^K\right) \prod_{j=\ell+1}^{L-1}\left(1 - \frac{x}{2^j}\right)^K, \quad 0 < x < 1,$$

*and is applied on matrix $\gamma_0 \mathbf{h}\tilde{\mathbf{h}}^\top$ entry-wise.*

*Proof.* We first use Lemma D.17 to simplify the recursion in (32):

$$\mathring{\mathcal{C}}_t \preceq \mathscr{G}_t \circ \mathring{\mathcal{C}}_{t-1} + \gamma_t^2 \cdot 8 \cdot 3^7 \langle \mathbf{H}, \mathring{\mathcal{C}}_{t-1} \circ \tilde{\mathbf{H}}\rangle \mathcal{S}^{(1)} + \gamma_t^2(16 \cdot 3^7 + 18)(\psi^2 \texttt{tr}(\mathbf{H}) + \sigma^2)\mathcal{S}^{(1)}$$

$$\preceq \mathscr{G}_t \circ \mathring{\mathcal{C}}_{t-1} + c\gamma_0\gamma_t^2 \cdot 8 \cdot 3^7 \langle \mathbf{H}, \mathcal{S}^{(0)} \circ \tilde{\mathbf{H}}\rangle \mathcal{S}^{(1)} + \gamma_t^2 \frac{c}{2}\mathcal{S}^{(1)}$$

$$\qquad \text{by Lemma D.17 and the definition of } c$$

$$= \mathscr{G}_t \circ \mathring{\mathcal{C}}_{t-1} + \gamma_t^2 c\mathcal{S}^{(1)}, \quad t \ge 1. \qquad \text{by the definition of } \mathcal{S}^{(0)} \text{ and the choice of } \gamma_0$$

We can unroll the above recursion using the monotonicity of $\mathscr{G}$ on diagonal operators by Lemma D.3. Then we have

$$\mathring{\mathcal{C}}_T \preceq \left(\prod_{t=1}^T \mathscr{G}_t\right) \circ \mathcal{C}_0 + c\sum_{t=1}^T \gamma_t^2\left(\prod_{k=t+1}^T \mathscr{G}_k\right) \circ \mathcal{S}^{(1)}$$

$$= c\sum_{t=1}^T \gamma_t^2\left(\prod_{k=t+1}^T \mathscr{G}_k\right) \circ \mathcal{S}^{(1)} \qquad \text{by Lemma D.3 and } \mathcal{C}_0 = \mathbf{0}^{\otimes 2}$$

$$= c\sum_{t=1}^T \gamma_t^2 \prod_{k=t+1}^T \left(\mathcal{S}^{(0)} - \gamma_k\mathcal{S}^{(1)}\right)^{\bullet 2} \bullet \mathcal{S}^{(1)}. \qquad \text{by Lemma D.15}$$

Consider an arbitrary non-negative vector

$$\mathbf{v} \in \mathbb{R}^d, \quad \mathbf{v} \succeq \mathbf{0},$$

and use Lemma D.16, then we have

$$\mathring{\mathcal{C}}_T \circ \texttt{mat}\{\mathbf{v}\} \preceq c\sum_{t=1}^T \gamma_t^2\left(\prod_{k=t+1}^T \left(\mathcal{S}^{(0)} - \gamma_t\mathcal{S}^{(1)}\right)^{\bullet 2} \bullet \mathcal{S}^{(1)}\right) \circ \texttt{mat}\{\tilde{\mathbf{h}}\}$$

$$= \texttt{cmat}\Big\{\sum_{t=1}^T \gamma_t^2\Big(\prod_{k=t+1}^T \left(\mathbf{J} - \gamma_k\mathbf{h}\tilde{\mathbf{h}}^\top\right)^{\odot 2} \odot \left(\mathbf{h}\tilde{\mathbf{h}}^\top\right)\Big)\mathbf{v}\Big\}$$

$$\preceq \texttt{cmat}\Big\{\sum_{t=1}^T \gamma_t^2\Big(\prod_{k=t+1}^T \left(\mathbf{J} - \gamma_k\mathbf{h}\tilde{\mathbf{h}}^\top\right) \odot \left(\mathbf{h}\tilde{\mathbf{h}}^\top\right)\Big)\mathbf{v}\Big\},$$

where the last inequality is because, by our choice of $\gamma_0$, the following holds in entry-wise:

$$0 \le \mathbf{J} - \gamma_k\mathbf{h}\tilde{\mathbf{h}}^\top \le \mathbf{J}.$$

Let

$$K := T/\log(T), \quad L = \log(T),$$

and recall the stepsize schedule (7), then for the non-negative vector $\mathbf{v}$, we have

$$\mathring{\mathcal{C}}_T \circ \texttt{mat}\{\mathbf{v}\}$$

$$\preceq c\,\texttt{mat}\bigg\{\sum_{t=1}^{T}\gamma_t^2\bigg(\prod_{k=t+1}^{T}\big(\mathbf{J}-\gamma_k\mathbf{h}\tilde{\mathbf{h}}^\top\big)\odot\big(\mathbf{h}\tilde{\mathbf{h}}^\top\big)\bigg)\mathbf{v}\bigg\} \tag{35}$$

$$= c\,\texttt{mat}\bigg\{\sum_{\ell=0}^{L-1}\Big(\frac{\gamma_0}{2^\ell}\Big)^2\bigg(\sum_{i=1}^{K}\Big(\mathbf{J}-\frac{\gamma_0}{2^\ell}\mathbf{h}\tilde{\mathbf{h}}^\top\Big)^{\odot(K-i)}\odot$$

$$\prod_{j=\ell+1}^{L-1}\Big(\mathbf{J}-\frac{\gamma_0}{2^j}\mathbf{h}\tilde{\mathbf{h}}^\top\Big)^{\odot K}\odot\big(\mathbf{h}\tilde{\mathbf{h}}^\top\big)\bigg)\mathbf{v}\bigg\}$$

$$= c\,\texttt{mat}\bigg\{\sum_{\ell=0}^{L-1}\frac{\gamma_0}{2^\ell}\bigg(\Big(\mathbf{J}-\big(\mathbf{J}-\frac{\gamma_0}{2^\ell}\mathbf{h}\tilde{\mathbf{h}}^\top\big)^{\odot K}\Big)\odot\prod_{j=\ell+1}^{L-1}\Big(\mathbf{J}-\frac{\gamma_0}{2^j}\mathbf{h}\tilde{\mathbf{h}}^\top\Big)^{\odot K}\bigg)\mathbf{v}\bigg\}$$

$$= c\,\texttt{mat}\bigg\{\Big(f\big(\gamma_0\mathbf{h}\tilde{\mathbf{h}}^\top\big)\odot\big(\mathbf{h}\tilde{\mathbf{h}}^\top\big)^{\odot-1}\Big)\mathbf{v}\bigg\},$$

where

$$f(x):=\sum_{\ell=0}^{L-1}\frac{x}{2^\ell}\bigg(1-\Big(1-\frac{x}{2^\ell}\Big)^K\bigg)\prod_{j=\ell+1}^{L-1}\Big(1-\frac{x}{2^j}\Big)^K,\quad 0<x<1,$$

and is applied on matrix $\gamma_0\mathbf{h}\tilde{\mathbf{h}}^\top$ entry-wise. $\qquad\square$

The following lemma is an adaptation of Lemma C.3 in Wu et al. (2022).

**Lemma D.19.** *Consider a scalar function*

$$f(x):=\sum_{\ell=0}^{L-1}\frac{x}{2^\ell}\bigg(1-\Big(1-\frac{x}{2^\ell}\Big)^K\bigg)\prod_{j=\ell+1}^{L-1}\Big(1-\frac{x}{2^j}\Big)^K,\quad 0<x<1.$$

*Then*

$$0<f(x)\le\min\Big\{\frac{8}{K},\,2Kx^2\Big\},\quad 0<x<1.$$

We are ready to show our final variance error upper bound.

**Theorem D.20** (Variance error bound)**.** *Suppose that*

$$\gamma_0\le\frac{1}{16\cdot3^7\texttt{tr}(\mathbf{H})\texttt{tr}(\tilde{\mathbf{H}})}.$$

*Then we have*

$$\big\langle\mathbf{H},\,\mathcal{C}_T\circ\tilde{\mathbf{H}}\big\rangle\le\frac{8c}{K}\sum_{i,j}\min\big\{1,\,K^2\gamma_0^2\lambda_i^2\tilde{\lambda}_j^2\big\},$$

*where*

$$c:=\big(32\cdot3^7+36\big)\big(\psi^2\texttt{tr}(\mathbf{H})+\sigma^2\big),\quad K:=T/\log(T),$$

$\big(\lambda_i\big)_{i\ge1}$ *are the eigenvalues of* $\mathbf{H}$*, and* $\big(\tilde{\lambda}_i\big)_{i\ge1}$ *are the eigenvalues of* $\tilde{\mathbf{H}}$*, that is*

$$\tilde{\lambda}_j=\psi^2\lambda_j\Big(\frac{\texttt{tr}(\mathbf{H})+\sigma^2/\psi^2}{N}+\frac{N+1}{N}\lambda_j\Big),\quad j\ge1.$$

*Proof.* Let us compute a variance error bound using Lemma D.18:

$$\big\langle\mathbf{H},\,\mathcal{C}_T\circ\tilde{\mathbf{H}}\big\rangle=\big\langle\mathbf{H},\,\mathring{\mathcal{C}}_T\circ\tilde{\mathbf{H}}\big\rangle$$

$$=\big\langle\texttt{mat}\{\mathbf{h}\},\,\mathring{\mathcal{C}}_T\circ\texttt{mat}\{\tilde{\mathbf{h}}\}\big\rangle$$

$$\le c\big\langle\texttt{mat}\{\mathbf{h}\},\,\texttt{mat}\big\{\big(f\big(\gamma_0\mathbf{h}\tilde{\mathbf{h}}^\top\big)\odot\big(\mathbf{h}\tilde{\mathbf{h}}^\top\big)^{\odot-1}\big)\tilde{\mathbf{h}}\big\}\big\rangle\qquad\text{by Lemma D.18}$$

$$=c\,\mathbf{h}^\top\big(f\big(\gamma_0\mathbf{h}\tilde{\mathbf{h}}^\top\big)\odot\big(\mathbf{h}\tilde{\mathbf{h}}^\top\big)^{\odot-1}\big)\tilde{\mathbf{h}}.$$

By Lemma D.19, we have

$$0 \leq f\big(\gamma_0 \mathbf{h}\tilde{\mathbf{h}}^\top\big) \odot \big(\mathbf{h}\tilde{\mathbf{h}}^\top\big)^{\odot -1} \leq \min\left\{\frac{8}{K}\mathbf{J},\, 2K\big(\gamma_0 \mathbf{h}\tilde{\mathbf{h}}^\top\big)^{\odot 2}\right\} \odot \big(\mathbf{h}\tilde{\mathbf{h}}^\top\big)^{\odot -1}$$

$$\leq \frac{8}{K}\min\left\{\big(\mathbf{h}\tilde{\mathbf{h}}^\top\big)^{\odot -1},\, K^2\gamma_0^2 \mathbf{h}\tilde{\mathbf{h}}^\top\right\},$$

where "min" and "$\leq$" are taken entrywise. So the variance error can be bounded by

$$\big\langle \mathbf{H},\, \mathcal{C}_T \circ \tilde{\mathbf{H}} \big\rangle \leq c\mathbf{h}^\top \Big(f\big(\gamma_0 \mathbf{h}\tilde{\mathbf{h}}^\top\big) \odot \big(\mathbf{h}\tilde{\mathbf{h}}^\top\big)^{\odot -1}\Big)\tilde{\mathbf{h}}$$

$$\leq \frac{8}{K}\mathbf{h}^\top \min\left\{\big(\mathbf{h}\tilde{\mathbf{h}}^\top\big)^{\odot -1},\, K^2\gamma_0^2 \mathbf{h}\tilde{\mathbf{h}}^\top\right\}\tilde{\mathbf{h}}$$

$$= \frac{8c}{K} \begin{pmatrix}\lambda_1 & \cdots & \lambda_d\end{pmatrix} \begin{pmatrix} \ddots & & \vdots & & \ddots \\ \cdots & & \min\left\{\frac{1}{\lambda_i \tilde{\lambda}_j},\, K^2\gamma_0^2\lambda_i\tilde{\lambda}_j\right\} & & \cdots \\ \ddots & & \vdots & & \ddots \end{pmatrix}\begin{pmatrix}\tilde{\lambda}_1 \\ \vdots \\ \tilde{\lambda}_d\end{pmatrix}$$

$$= \frac{8c}{K} \begin{pmatrix}\lambda_1 & \cdots & \lambda_d\end{pmatrix}\begin{pmatrix}\vdots \\ \sum_j \min\left\{\frac{1}{\lambda_i \tilde{\lambda}_j},\, K^2\gamma_0^2\lambda_i\tilde{\lambda}_j\right\}\tilde{\lambda}_j \\ \vdots\end{pmatrix}$$

$$= \frac{8c}{K}\sum_i \sum_j \min\left\{\frac{1}{\lambda_i \tilde{\lambda}_j},\, K^2\gamma_0^2\lambda_i\tilde{\lambda}_j\right\}\lambda_i\tilde{\lambda}_j$$

$$= \frac{8c}{K}\sum_{i,j} \min\left\{1,\, K^2\gamma_0^2\lambda_i^2\tilde{\lambda}_j^2\right\},$$

where

$$\tilde{\lambda}_j = \psi^2 \lambda_j \left(\frac{\mathtt{tr}(\mathbf{H}) + \sigma^2/\psi^2}{N} + \frac{N+1}{N}\lambda_j\right).$$

We have completed the proof. $\qquad\square$

### D.7  BIAS ERROR ANALYSIS

Throughout this section, we denote the bias error at the $t$-th iterate by

$$b_t := \big\langle \mathbf{H},\, \mathcal{B}_t \circ \tilde{\mathbf{H}} \big\rangle = \big\langle \mathbf{H},\, \mathring{\mathcal{B}}_t \circ \tilde{\mathbf{H}} \big\rangle, \tag{36}$$

where $\mathbf{H}$ (hence also $\tilde{\mathbf{H}}$) is assumed to be diagonal and $\mathring{\mathcal{B}}_t$ admits the recursion in (31).

#### D.7.1  CONSTANT-STEPSIZE CASE

Since the stepsize schedule (7) is epoch-wise constant, we begin our bias error analysis by considering constant-stepsize cases, where the stepsize is denoted by $\gamma > 0$. In this case, (31) reduces to

$$\mathring{\mathcal{B}}_t \preceq \mathscr{G} \circ \mathring{\mathcal{B}}_{t-1} + \gamma^2 c_1 b_{t-1}\mathcal{S}^{(1)}, \quad t \geq 1, \quad \text{where } c_1 := 8 \cdot 3^7. \tag{37}$$

Unrolling (37), we have

$$\mathring{\mathcal{B}}_n \preceq \mathscr{G}^n \circ \mathring{\mathcal{B}}_0 + \gamma^2 c_1 \sum_{t=0}^{n-1} b_t \mathscr{G}^{n-1-t} \circ \mathcal{S}^{(1)}$$

$$= \mathscr{G}^n \circ \mathcal{B}_0 + \gamma^2 c_1 \sum_{t=0}^{n-1} b_t\big(\mathcal{S}^{(0)} - \gamma\mathcal{S}^{(1)}\big)^{\bullet 2(n-1-t)} \bullet \mathcal{S}^{(1)}, \quad n \geq 1. \quad \text{by Lemma D.15} \tag{38}$$

**Lemma D.21** (Controlled blow-up of bias error). *Consider* (37). *If*

$$\gamma \le \frac{1}{2c_1 \mathtt{tr}(\mathbf{H})\mathtt{tr}(\tilde{\mathbf{H}})},$$

*then for every $n \ge 0$, it holds that*

$$b_n \le \big(1 + 2c_1\gamma\mathtt{tr}(\mathbf{H})\mathtt{tr}(\tilde{\mathbf{H}})\big)b_0.$$

*Proof.* We prove the claim by induction. The claim clearly holds when $n = 0$. Now suppose that

$$b_t \le \big(1 + 2c_1\gamma\mathtt{tr}(\mathbf{H})\mathtt{tr}(\tilde{\mathbf{H}})\big)b_0, \quad t = 0, \dots, n-1.$$

For $n$, we have

$$\mathring{\mathcal{B}}_n \preceq \mathscr{G}^n \circ \mathring{\mathcal{B}}_0 + \gamma^2 c_1 \sum_{t=0}^{n-1} b_t \big(\mathcal{S}^{(0)} - \gamma\mathcal{S}^{(1)}\big)^{\bullet 2(n-1-t)} \bullet \mathcal{S}^{(1)} \qquad \text{by (37)}$$

$$\preceq \mathring{\mathcal{B}}_0 + \gamma^2 c_1 \sum_{t=0}^{n-1} b_t \big(\mathcal{S}^{(0)} - \gamma\mathcal{S}^{(1)}\big)^{\bullet 2(n-1-t)} \bullet \mathcal{S}^{(1)} \qquad \text{by Lemma D.14}$$

$$\preceq \mathring{\mathcal{B}}_0 + \gamma^2 c_1 2b_0 \sum_{t=0}^{n-1} \big(\mathcal{S}^{(0)} - \gamma\mathcal{S}^{(1)}\big)^{\bullet 2(n-1-t)} \bullet \mathcal{S}^{(1)},$$

where the last inequality is by the induction hypothesis and $\gamma \le 1/2c_1\mathtt{tr}(\mathbf{H})\mathtt{tr}(\tilde{\mathbf{H}})$. Next, consider an arbitrary non-negative vector $\mathbf{v} \in \mathbb{R}^d$, by Lemma D.16, we have

$$\mathring{\mathcal{B}}_n \circ \mathtt{mat}\{\mathbf{v}\}$$

$$\preceq \mathring{\mathcal{B}}_0 \circ \mathtt{mat}\{\mathbf{v}\} + \gamma^2 c_1 2b_0 \bigg(\sum_{t=0}^{n-1} \big(\mathcal{S}^{(0)} - \gamma\mathcal{S}^{(1)}\big)^{\bullet 2(n-1-t)} \bullet \mathcal{S}^{(1)}\bigg) \circ \mathtt{mat}\{\mathbf{v}\}$$

$$= \mathring{\mathcal{B}}_0 \circ \mathtt{mat}\{\mathbf{v}\} + \gamma^2 c_1 2b_0 \mathtt{mat}\bigg\{\bigg(\sum_{t=0}^{n-1} \big(\mathbf{J} - \gamma\mathbf{h}\tilde{\mathbf{h}}^\top\big)^{\odot 2(n-1-t)} \odot \mathbf{h}\tilde{\mathbf{h}}^\top\bigg)\mathbf{v}\bigg\}$$

$$\qquad \text{by Lemma D.16}$$

$$\preceq \mathring{\mathcal{B}}_0 \circ \mathtt{mat}\{\mathbf{v}\} + \gamma^2 c_1 2b_0 \mathtt{mat}\bigg\{\bigg(\sum_{t=0}^{n-1} \big(\mathbf{J} - \gamma\mathbf{h}\tilde{\mathbf{h}}^\top\big)^{\odot (n-1-t)} \odot \mathbf{h}\tilde{\mathbf{h}}^\top\bigg)\mathbf{v}\bigg\}$$

$$\qquad \text{since } 0 \le \mathbf{J} - \gamma\mathbf{h}\tilde{\mathbf{h}}^\top \le \mathbf{J}, \text{ entrywise}$$

$$= \mathring{\mathcal{B}}_0 \circ \mathtt{mat}\{\mathbf{v}\} + \gamma c_1 2b_0 \mathtt{mat}\bigg\{\bigg(\mathbf{J} - \big(\mathbf{J} - \gamma\mathbf{h}\tilde{\mathbf{h}}^\top\big)^{\odot n}\bigg)\mathbf{v}\bigg\}$$

$$\preceq \mathring{\mathcal{B}}_0 \circ \mathtt{mat}\{\mathbf{v}\} + \gamma c_1 2b_0 \mathtt{mat}\{\mathbf{v}\}. \quad \text{since } 0 \le \mathbf{J} - \big(\mathbf{J} - \gamma\mathbf{h}\tilde{\mathbf{h}}^\top\big)^{\odot n} \le \mathbf{J}, \text{ entrywise}$$

Then we have

$$\begin{aligned}
b_n &= \big\langle \mathbf{H}, \ \mathring{\mathcal{B}}_n \circ \tilde{\mathbf{H}} \big\rangle \\
&= \big\langle \mathbf{H}, \ \mathring{\mathcal{B}}_n \circ \mathtt{mat}\{\tilde{\mathbf{h}}\} \big\rangle \\
&\le \big\langle \mathbf{H}, \ \mathring{\mathcal{B}}_0 \circ \mathtt{mat}\{\tilde{\mathbf{h}}\} \big\rangle + \gamma c_1 2b_0 \big\langle \mathbf{H}, \ \mathtt{mat}\{\tilde{\mathbf{h}}\} \big\rangle \\
&= b_0 + \gamma c_1 2b_0 \big\langle \mathbf{H}, \ \tilde{\mathbf{H}} \big\rangle \\
&\le b_0 + \gamma c_1 2b_0 \mathtt{tr}(\mathbf{H})\mathtt{tr}(\tilde{\mathbf{H}}) \\
&= \big(1 + 2c_1\gamma\mathtt{tr}(\mathbf{H})\mathtt{tr}(\tilde{\mathbf{H}})\big)b_0,
\end{aligned}$$

which completes the induction. $\qquad\square$

**Lemma D.22** (A bound on the sum of the bias error). *Suppose that*

$$\gamma \le \frac{1}{2c_1 \mathtt{tr}(\mathbf{H})\mathtt{tr}(\tilde{\mathbf{H}})}.$$

*Suppose that*
$$\mathcal{B}_0 = (\mathbf{\Gamma}_0 - \mathbf{\Gamma}^*)^{\otimes 2}$$
*and that $\mathbf{\Gamma}_0$ commutes with $\mathbf{H}$. Then for every $n \geq 1$, we have*
$$\sum_{t=0}^{n-1} b_t \leq \frac{1}{\gamma} \big\langle \mathbf{I} - (\mathbf{I} - \gamma\mathbf{H}\tilde{\mathbf{H}})^{2n}, \ (\mathbf{\Gamma}_0 - \mathbf{\Gamma}^*)^2 \big\rangle.$$

*Proof.* By Lemma D.14, we have
$$\mathscr{G} \circ \mathring{\mathcal{B}}_{t-1} \circ \mathbf{I} = \mathring{\mathcal{B}}_{t-1} \circ \mathbf{I} - 2\gamma\mathbf{H}\mathring{\mathcal{B}}_{t-1} \circ \tilde{\mathbf{H}} + \gamma^2\mathbf{H}^2\mathring{\mathcal{B}}_{t-1} \circ (\tilde{\mathbf{H}})^2$$
$$\preceq \mathring{\mathcal{B}}_{t-1} \circ \mathbf{I} - 2\gamma\mathbf{H}\mathring{\mathcal{B}}_{t-1} \circ \tilde{\mathbf{H}} + \gamma^2\mathtt{tr}(\mathbf{H})\mathtt{tr}(\tilde{\mathbf{H}})\mathbf{H}\mathring{\mathcal{B}}_{t-1} \circ \tilde{\mathbf{H}}.$$
Using the above and (37), we have
$$\langle \mathbf{I}, \ \mathring{\mathcal{B}}_t \circ \mathbf{I} \rangle \leq \langle \mathbf{I}, \ \mathscr{G} \circ \mathring{\mathcal{B}}_{t-1} \circ \mathbf{I} \rangle + \gamma^2 c_1 b_{t-1} \langle \mathbf{I}, \ \mathcal{S}^{(1)} \circ \mathbf{I} \rangle \qquad \text{by (37)}$$
$$\leq \langle \mathbf{I}, \ \mathring{\mathcal{B}}_{t-1} \circ \mathbf{I} \rangle - 2\gamma \langle \mathbf{H}, \ \mathring{\mathcal{B}}_{t-1} \circ \tilde{\mathbf{H}} \rangle + \gamma^2\mathtt{tr}(\mathbf{H})\mathtt{tr}(\tilde{\mathbf{H}}) \langle \mathbf{H}, \ \mathring{\mathcal{B}}_{t-1} \circ \tilde{\mathbf{H}} \rangle$$
$$\qquad + \gamma^2 c_1 \mathtt{tr}(\mathbf{H})\mathtt{tr}(\tilde{\mathbf{H}}) b_{t-1}$$
$$= \langle \mathbf{I}, \ \mathring{\mathcal{B}}_{t-1} \circ \mathbf{I} \rangle - 2\gamma b_{t-1} + \gamma^2(1 + c_1)\mathtt{tr}(\mathbf{H})\mathtt{tr}(\tilde{\mathbf{H}}) b_{t-1}$$
$$\leq \langle \mathbf{I}, \ \mathring{\mathcal{B}}_{t-1} \circ \mathbf{I} \rangle - \gamma b_{t-1}. \qquad \text{since } \gamma \leq 1/(2c_1\mathtt{tr}(\mathbf{H})\mathtt{tr}(\tilde{\mathbf{H}}))$$
Performing a telescope sum, we have
$$\sum_{t=0}^{n-1} b_t \leq \frac{1}{\gamma} \Big( \langle \mathbf{I}, \ \mathring{\mathcal{B}}_0 \circ \mathbf{I} \rangle - \langle \mathbf{I}, \ \mathring{\mathcal{B}}_n \circ \mathbf{I} \rangle \Big).$$

We now derive a lower bound for $\mathring{\mathcal{B}}_n$. By Lemma D.12 and (24), we have
$$\mathcal{B}_t = \mathscr{S} \circ \mathcal{B}_{t-1} \qquad\qquad\qquad \text{by the definition in (24)}$$
$$\succeq \mathscr{G} \circ \mathcal{B}_{t-1}, \quad t \geq 1. \qquad\qquad \text{by Lemma D.12}$$
Performing diagonalization using Lemma D.13, we have
$$\mathring{\mathcal{B}}_t \succeq \mathscr{G} \circ \mathring{\mathcal{B}}_{t-1}, \quad t \geq 1.$$
Solving the recursion, we have
$$\mathring{\mathcal{B}}_n \succeq \mathscr{G}^n \circ \mathring{\mathcal{B}}_0$$
$$= \mathscr{G}^n \circ (\mathbf{\Gamma}_0 - \mathbf{\Gamma}^*)^{\otimes 2} \qquad\qquad \text{since both } \mathbf{\Gamma}_0 \text{ and } \mathbf{\Gamma}^* \text{ commute with } \mathbf{H}$$
$$= \Big( (\mathbf{I} - \gamma\mathbf{H}\tilde{\mathbf{H}})^n (\mathbf{\Gamma}_0 - \mathbf{\Gamma}^*) \Big)^{\otimes 2}. \qquad \text{by the definition of } \mathscr{G} \text{ in (23)}$$

Putting these together, we have
$$\sum_{t=0}^{n-1} b_t \leq \frac{1}{\gamma} \Big( \langle \mathbf{I}, \ \mathring{\mathcal{B}}_0 \circ \mathbf{I} \rangle - \langle \mathbf{I}, \ \mathring{\mathcal{B}}_n \circ \mathbf{I} \rangle \Big)$$
$$\leq \frac{1}{\gamma} \Big( \mathtt{tr}\big((\mathbf{\Gamma}_0 - \mathbf{\Gamma}^*)^2\big) - \mathtt{tr}\big((\mathbf{I} - \gamma\mathbf{H}\tilde{\mathbf{H}})^{2n}(\mathbf{\Gamma}_0 - \mathbf{\Gamma}^*)^2\big) \Big)$$
$$= \frac{1}{\gamma} \big\langle \mathbf{I} - (\mathbf{I} - \gamma\mathbf{H}\tilde{\mathbf{H}})^{2n}, \ (\mathbf{\Gamma}_0 - \mathbf{\Gamma}^*)^2 \big\rangle,$$
which completes the proof. $\qquad\qquad\qquad\qquad\qquad\qquad\qquad\qquad\qquad\qquad\qquad\qquad\qquad$ $\square$

**Lemma D.23** (A decreasing bound on bias error). *Suppose that*
$$\gamma \leq \frac{1}{6c_1\mathtt{tr}(\mathbf{H})\mathtt{tr}(\tilde{\mathbf{H}})}.$$

*Suppose that*
$$\mathcal{B}_0 = (\mathbf{\Gamma}_0 - \mathbf{\Gamma}^*)^{\otimes 2}$$
*and that $\mathbf{\Gamma}_0$ commutes with $\mathbf{H}$. Then for every $n \geq 0$, we have*
$$b_n \leq \frac{1}{\max\{n, 1\}\gamma} \langle \mathbf{I}, \ (\mathbf{\Gamma}_0 - \mathbf{\Gamma}^*)^2 \rangle.$$

*Proof.* We prove the claim by induction. For $n = 0$, we have

$$b_0 = \langle \mathbf{H}, \ (\mathbf{\Gamma}_0 - \mathbf{\Gamma}^*)\mathbf{H}(\mathbf{\Gamma}_0 - \mathbf{\Gamma}^*)^\top \rangle$$
$$\leq \mathtt{tr}(\mathbf{H})\mathtt{tr}(\tilde{\mathbf{H}})\langle \mathbf{I}, \ (\mathbf{\Gamma}_0 - \mathbf{\Gamma}^*)^2 \rangle$$
$$\leq \frac{1}{\gamma}\langle \mathbf{I}, \ (\mathbf{\Gamma}_0 - \mathbf{\Gamma}^*)^2 \rangle.$$

Now, suppose that

$$b_t \leq \frac{1}{\max\{t, 1\}\gamma}\langle \mathbf{I}, \ (\mathbf{\Gamma}_0 - \mathbf{\Gamma}^*)^2 \rangle, \quad t = 0, 1, \ldots, n - 1.$$

For $b_n$, considering an arbitrary non-negative vector $\mathbf{v} \in \mathbb{R}^d$, we have

$$\mathring{\mathcal{B}}_n \circ \mathtt{mat}\{\mathbf{v}\}$$
$$\preceq \mathscr{G}^n\left(\mathring{\mathcal{B}}_0\right) \circ \mathtt{mat}\{\mathbf{v}\} + \gamma^2 c_1 \sum_{t=0}^{n-1} b_t\left(\left(\mathcal{S}^{(0)} - \gamma\mathcal{S}^{(1)}\right)^{\bullet 2(n-1-t)} \bullet \mathcal{S}^{(1)}\right) \circ \mathtt{mat}\{\mathbf{v}\}$$
$$= \mathscr{G}^n\left(\mathring{\mathcal{B}}_0\right) \circ \mathtt{mat}\{\mathbf{v}\} + \gamma^2 c_1 \sum_{t=0}^{n-1} b_t\mathtt{mat}\left\{\left(\left(\mathbf{J} - \gamma\mathbf{h}\tilde{\mathbf{h}}^\top\right)^{\odot 2(n-1-t)} \odot \left(\mathbf{h}\tilde{\mathbf{h}}^\top\right)\right)\mathbf{v}\right\},$$

where the inequality is by (38) and the equality is by Lemma D.16. We will bound the second term in two parts, $\sum_{t=0}^{n/2-1}$ and $\sum_{t=n/2}^{n-1}$, separately. For the first half of the summation, we have

$$\sum_{t=0}^{n/2-1} b_t\mathtt{mat}\left\{\left(\left(\mathbf{J} - \gamma\mathbf{h}\tilde{\mathbf{h}}^\top\right)^{\odot 2(n-1-t)} \odot \left(\mathbf{h}\tilde{\mathbf{h}}^\top\right)\right)\mathbf{v}\right\}$$
$$\preceq \sum_{t=0}^{n/2-1} b_t\mathtt{mat}\left\{\left(\left(\mathbf{J} - \gamma\mathbf{h}\tilde{\mathbf{h}}^\top\right)^{\odot n} \odot \left(\mathbf{h}\tilde{\mathbf{h}}^\top\right)\right)\mathbf{v}\right\} \qquad \text{since } \mathbf{J} - \gamma\mathbf{h}\tilde{\mathbf{h}}^\top \leq \mathbf{J}, \text{ entrywise}$$
$$\preceq \sum_{t=0}^{n/2-1} b_t\mathtt{mat}\left\{\left(\frac{1}{n\gamma}\mathbf{J}\right)\mathbf{v}\right\} \qquad \text{since } (1 - x)^n \leq 1/(nx), \ 0 < x < 1$$
$$= \sum_{t=0}^{n/2-1} b_t\frac{1}{n\gamma}\mathtt{mat}\{\mathbf{v}\}$$
$$\preceq \frac{1}{\gamma}\langle \mathbf{I} - \left(\mathbf{I} - \gamma\mathbf{H}\tilde{\mathbf{H}}\right)^n, \ (\mathbf{\Gamma}_0 - \mathbf{\Gamma}^*)^2 \rangle\frac{1}{n\gamma}\mathtt{mat}\{\mathbf{v}\} \qquad \text{by Lemma D.22}$$
$$\preceq \frac{1}{\gamma}\langle \mathbf{I}, \ (\mathbf{\Gamma}_0 - \mathbf{\Gamma}^*)^2 \rangle\frac{1}{n\gamma}\mathtt{mat}\{\mathbf{v}\}.$$

For the second half of the summation, we have

$$\sum_{t=n/2}^{n-1} b_t\mathtt{mat}\left\{\left(\left(\mathbf{J} - \gamma\mathbf{h}\tilde{\mathbf{h}}^\top\right)^{\odot 2(n-1-t)} \odot \left(\mathbf{h}\tilde{\mathbf{h}}^\top\right)\right)\mathbf{v}\right\}$$
$$\preceq \frac{2}{n\gamma}\langle \mathbf{I}, \ (\mathbf{\Gamma}_0 - \mathbf{\Gamma}^*)^2 \rangle \sum_{t=n/2}^{n-1} \mathtt{mat}\left\{\left(\left(\mathbf{J} - \gamma\mathbf{h}\tilde{\mathbf{h}}^\top\right)^{\odot 2(n-1-t)} \odot \left(\mathbf{h}\tilde{\mathbf{h}}^\top\right)\right)\mathbf{v}\right\}$$
$$\qquad \text{by the induction hypothesis}$$
$$\preceq \frac{2}{n\gamma}\langle \mathbf{I}, \ (\mathbf{\Gamma}_0 - \mathbf{\Gamma}^*)^2 \rangle \sum_{t=n/2}^{n-1} \mathtt{mat}\left\{\left(\left(\mathbf{J} - \gamma\mathbf{h}\tilde{\mathbf{h}}^\top\right)^{\odot (n-1-t)} \odot \left(\mathbf{h}\tilde{\mathbf{h}}^\top\right)\right)\mathbf{v}\right\}$$
$$\qquad \text{since } 0 \leq \mathbf{J} - \gamma\mathbf{h}\tilde{\mathbf{h}}^\top \leq \mathbf{J}, \text{ entrywise}$$
$$= \frac{2}{n\gamma}\langle \mathbf{I}, \ (\mathbf{\Gamma}_0 - \mathbf{\Gamma}^*)^2 \rangle\frac{1}{\gamma}\mathtt{mat}\left\{\left(\mathbf{J} - \left(\mathbf{J} - \gamma\mathbf{h}\tilde{\mathbf{h}}^\top\right)^{\odot (n/2)}\right)\mathbf{v}\right\}$$

$$\preceq \frac{2}{n\gamma}\langle \mathbf{I},\ (\mathbf{\Gamma}_0 - \mathbf{\Gamma}^*)^2\rangle \frac{1}{\gamma}\mathtt{mat}\{\mathbf{v}\}. \qquad \text{since } \mathbf{J} - \left(\mathbf{J} - \gamma\mathbf{h}\tilde{\mathbf{h}}^\top\right)^{\odot(n/2)} \leq \mathbf{J}, \text{ entrywise}$$

Bringing these two bounds back, we have

$$\mathring{\mathcal{B}}_n \circ \mathtt{mat}\{\mathbf{v}\}$$

$$\preceq \mathscr{G}^n\big(\mathring{\mathcal{B}}_0\big) \circ \mathtt{mat}\{\mathbf{v}\} + \gamma^2 c_1\left(\frac{1}{\gamma}\langle \mathbf{I},\ (\mathbf{\Gamma}_0 - \mathbf{\Gamma}^*)^2\rangle \frac{1}{n\gamma}\mathtt{mat}\{\mathbf{v}\} + \frac{2}{n\gamma}\langle \mathbf{I},\ (\mathbf{\Gamma}_0 - \mathbf{\Gamma}^*)^2\rangle \frac{1}{\gamma}\mathtt{mat}\{\mathbf{v}\}\right)$$

$$= \mathscr{G}^n\big(\mathring{\mathcal{B}}_0\big) \circ \mathtt{mat}\{\mathbf{v}\} + 3\gamma c_1\mathtt{mat}\{\mathbf{v}\}\frac{1}{n\gamma}\langle \mathbf{I},\ (\mathbf{\Gamma}_0 - \mathbf{\Gamma}^*)^2\rangle$$

$$= \left(\big(\mathbf{I} - \gamma\mathbf{H}\tilde{\mathbf{H}}\big)^n(\mathbf{\Gamma}_0 - \mathbf{\Gamma}^*)\right)^{\otimes 2} \circ \mathtt{mat}\{\mathbf{v}\} + 3\gamma c_1\mathtt{mat}\{\mathbf{v}\}\frac{1}{n\gamma}\langle \mathbf{I},\ (\mathbf{\Gamma}_0 - \mathbf{\Gamma}^*)^2\rangle,$$

where the last equality is by the definition of $\mathscr{G}$ in (23). Based on the above, we have

$$b_n = \langle \mathbf{H},\ \mathring{\mathcal{B}}_n \circ \tilde{\mathbf{H}}\rangle$$

$$= \langle \mathbf{H},\ \mathring{\mathcal{B}}_n \circ \mathtt{mat}\{\tilde{\mathbf{h}}\}\rangle$$

$$\leq \left\langle \mathbf{H},\ \left(\big(\mathbf{I} - \gamma\mathbf{H}\tilde{\mathbf{H}}\big)^n(\mathbf{\Gamma}_0 - \mathbf{\Gamma}^*)\right)^{\otimes 2} \circ \mathtt{mat}\{\tilde{\mathbf{h}}\}\right\rangle + 3\gamma c_1\langle \mathbf{H},\ \mathtt{mat}\{\tilde{\mathbf{h}}\}\rangle \frac{1}{n\gamma}\langle \mathbf{I},\ (\mathbf{\Gamma}_0 - \mathbf{\Gamma}^*)^2\rangle$$

$$= \left\langle \big(\mathbf{I} - \gamma\mathbf{H}\tilde{\mathbf{H}}\big)^{2n}\mathbf{H}\tilde{\mathbf{H}},\ (\mathbf{\Gamma}_0 - \mathbf{\Gamma}^*)^2\right\rangle + 3\gamma c_1\langle \mathbf{H},\ \tilde{\mathbf{H}}\rangle \frac{1}{n\gamma}\langle \mathbf{I},\ (\mathbf{\Gamma}_0 - \mathbf{\Gamma}^*)^2\rangle$$

$$\qquad \text{since } \mathbf{\Gamma}_0 \text{ and } \mathbf{\Gamma}^* \text{ both commute with } \mathbf{H}$$

$$\leq \left\langle \frac{1}{2n\gamma}\mathbf{I},\ (\mathbf{\Gamma}_0 - \mathbf{\Gamma}^*)\right\rangle + 3\gamma c_1\mathtt{tr}(\mathbf{H})\mathtt{tr}(\tilde{\mathbf{H}})\rangle \frac{1}{n\gamma}\langle \mathbf{I},\ (\mathbf{\Gamma}_0 - \mathbf{\Gamma}^*)^2\rangle$$

$$\leq \frac{1}{n\gamma}\langle \mathbf{I},\ (\mathbf{\Gamma}_0 - \mathbf{\Gamma}^*)^2\rangle, \qquad \text{since } \gamma \leq 1/(6c_1\mathtt{tr}(\mathbf{H})\mathtt{tr}(\tilde{\mathbf{H}}))$$

which completes our induction. $\qquad\square$

### D.7.2 DECAYING-STEPSIZE CASE

We first show a crude bound on the bias iterate.

**Lemma D.24** (A crude bound). *Consider the bias iterate* (24). *Suppose that*

$$\gamma \leq \frac{1}{6c_1\mathtt{tr}(\mathbf{H})\mathtt{tr}(\tilde{\mathbf{H}})}.$$

*Suppose that*

$$\mathcal{B}_0 = \big(\mathbf{\Gamma}_0 - \mathbf{\Gamma}^*\big)^{\otimes 2}$$

*and that* $\mathbf{\Gamma}_0$ *commutes with* $\mathbf{H}$. *Then for every* $t \geq K$, *we have*

$$b_t \leq 4\left\langle \frac{1}{K\gamma_0}\mathbf{I}_{0:k} + \mathbf{H}_{k:\infty}\tilde{\mathbf{H}}_{k:\infty},\ (\mathbf{\Gamma}_0 - \mathbf{\Gamma}^*)^2\right\rangle.$$

*Proof.* Let

$$K = T/\log(T), \quad L = \log(T).$$

According to (7), in the first epoch, i.e., $t = 1, 2, \ldots, K$, the stepsize is constant, i.e., $\gamma_0$. Therefore, we can apply Lemmas D.21 and D.23 and obtain

$$b_K \leq \min\left\{\big(1 + 2c_1\gamma_0\mathtt{tr}(\mathbf{H})\mathtt{tr}(\tilde{\mathbf{H}})\big)b_0,\ \frac{1}{K\gamma_0}\langle \mathbf{I},\ (\mathbf{\Gamma}_0 - \mathbf{\Gamma}^*)^2\rangle\right\}$$

$$\leq 2\min\left\{\langle \mathbf{H},\ (\mathbf{\Gamma}_0 - \mathbf{\Gamma}^*)\tilde{\mathbf{H}}(\mathbf{\Gamma}_0 - \mathbf{\Gamma}^*)^\top\rangle,\ \frac{1}{K\gamma_0}\langle \mathbf{I},\ (\mathbf{\Gamma}_0 - \mathbf{\Gamma}^*)^2\rangle\right\}$$

$$\begin{aligned}
&= 2\min\left\{ \left\langle \mathbf{H}\tilde{\mathbf{H}},\ (\mathbf{\Gamma}_0 - \mathbf{\Gamma}^*)^2 \right\rangle,\ \frac{1}{K\gamma_0}\left\langle \mathbf{I},\ (\mathbf{\Gamma}_0 - \mathbf{\Gamma}^*)^2 \right\rangle \right\} \\
&\leq 2\left\langle \min\left\{ \mathbf{H}\tilde{\mathbf{H}},\ \frac{1}{K\gamma_0}\mathbf{I} \right\},\ (\mathbf{\Gamma}_0 - \mathbf{\Gamma}^*)^2 \right\rangle \\
&\leq 2\left\langle \frac{1}{K\gamma_0}\mathbf{I}_{0:k} + \mathbf{H}_{k:\infty}\tilde{\mathbf{H}}_{k:\infty},\ (\mathbf{\Gamma}_0 - \mathbf{\Gamma}^*)^2 \right\rangle.
\end{aligned}$$

Next, recall that the stepsize schedule (7) is epoch-wise constant, therefore we can recursively apply Lemma D.21 for epoch $2, 3, \ldots, L$. Suppose $t \geq K$ belongs to the $L^*$-th epoch, then we have

$$\begin{aligned}
b_t &\leq \prod_{\ell=1}^{L^*}\left(1 + 2c_1\frac{\gamma_0}{2^\ell}\mathtt{tr}(\mathbf{H})\mathtt{tr}(\tilde{\mathbf{H}})\right)b_K \\
&\leq \left(1 + 2c_1\gamma_0\mathtt{tr}(\mathbf{H})\mathtt{tr}(\tilde{\mathbf{H}})\right)b_K \\
&\leq 2b_K.
\end{aligned}$$

We complete the proof by bringing the upper bound on $b_K$. $\qquad\square$

**Theorem D.25** (Sharp bias bound). *Consider the bias iterate* (24). *Suppose that*

$$\gamma \leq \frac{1}{6 \cdot 8 \cdot 3^7 \mathtt{tr}(\mathbf{H})\mathtt{tr}(\tilde{\mathbf{H}})}.$$

*Suppose that*

$$\mathcal{B}_0 = (\mathbf{\Gamma}_0 - \mathbf{\Gamma}^*)^{\otimes 2}$$

*and that $\mathbf{\Gamma}_0$ commutes with $\mathbf{H}$. We have*

$$\begin{aligned}
b_T &\leq \left\langle \mathbf{H}\tilde{\mathbf{H}},\ \left(\prod_{t=1}^{T}(\mathbf{I} - \gamma_t\mathbf{H}\tilde{\mathbf{H}})(\mathbf{\Gamma}_0 - \mathbf{\Gamma}^*)\right)^2 \right\rangle \\
&\quad + 8 \cdot 3^7 \cdot 40\left\langle \frac{1}{K\gamma_0}\mathbf{I}_{0:k} + \mathbf{H}_{k:\infty}\tilde{\mathbf{H}}_{k:\infty},\ (\mathbf{\Gamma}_0 - \mathbf{\Gamma}^*)^2 \right\rangle\frac{1}{K}\sum_{i,j}\min\left\{1,\ K^2\gamma_0^2\lambda_i^2\tilde{\lambda}_j^2\right\}.
\end{aligned}$$

*Proof.* From (31), we have

$$\mathring{\mathcal{B}}_t \preceq \mathscr{G}_t \circ \mathring{\mathcal{B}}_{t-1} + c_1\gamma_t^2 b_{t-1}\mathcal{S}^{(1)}.$$

Unrolling the recursion, we have

$$\begin{aligned}
\mathring{\mathcal{B}}_T &\preceq \left(\prod_{t=1}^{T}\mathscr{G}_t\right)\circ\mathring{\mathcal{B}}_0 + c_1\sum_{t=0}^{T-1}\gamma_t^2 b_t\left(\prod_{k=t+1}^{T}\mathscr{G}_k\right)\circ\mathcal{S}^{(1)} \\
&= \left(\prod_{t=1}^{T}\mathscr{G}_t\right)\circ\mathring{\mathcal{B}}_0 + c_1\sum_{t=0}^{T-1}\gamma_t^2 b_t\prod_{k=t+1}^{T}\left(\mathcal{S}^{(0)} - \gamma_k\mathcal{S}^{(1)}\right)^{\bullet 2}\bullet\mathcal{S}^{(1)}, \quad \text{by Lemma D.15}
\end{aligned}$$

which implies that

$$\begin{aligned}
b_T &= \left\langle \mathbf{H},\ \mathring{\mathcal{B}}_T \circ \tilde{\mathbf{H}} \right\rangle \\
&\leq \left\langle \mathbf{H},\ \left(\prod_{t=1}^{T}\mathscr{G}_t\right)\circ\mathring{\mathcal{B}}_0\circ\tilde{\mathbf{H}} \right\rangle + c_1\sum_{t=0}^{T-1}\gamma_t^2 b_t\left\langle \mathbf{H},\ \left(\prod_{k=t+1}^{T}\left(\mathcal{S}^{(0)} - \gamma_k\mathcal{S}^{(1)}\right)^{\bullet 2}\bullet\mathcal{S}^{(1)}\right)\circ\tilde{\mathbf{H}} \right\rangle \\
&= \left\langle \mathbf{H},\ \left(\prod_{t=1}^{T}\mathscr{G}_t\right)\circ\mathring{\mathcal{B}}_0\circ\tilde{\mathbf{H}} \right\rangle \\
&\quad + c_1\sum_{t=0}^{T-1}\gamma_t^2 b_t\left\langle \mathbf{H},\ \mathtt{mat}\left\{\left(\prod_{k=t+1}^{T}(\mathbf{J} - \gamma_k\mathbf{h}\tilde{\mathbf{h}}^\top)^{\odot 2}\odot(\mathbf{h}\tilde{\mathbf{h}}^\top)\right)\tilde{\mathbf{h}}\right\} \right\rangle \quad \text{by Lemma D.16}
\end{aligned}$$

$$= \left\langle \mathbf{H}, \left( \prod_{t=1}^{T} \mathscr{G}_t \right) \circ \mathring{\mathcal{B}}_0 \circ \tilde{\mathbf{H}} \right\rangle$$

$$+ c_1 \sum_{t=0}^{T-1} \gamma_t^2 b_t \mathbf{h}^\top \left( \prod_{k=t+1}^{T} (\mathbf{J} - \gamma_k \mathbf{h}\tilde{\mathbf{h}}^\top)^{\odot 2} \odot (\mathbf{h}\tilde{\mathbf{h}}^\top) \right) \tilde{\mathbf{h}}. \tag{39}$$

For the first term in (39), using the assumption that $\mathbf{\Gamma}_0$ commutes with $\mathbf{H}$ and the definition of $\mathscr{G}$ in (23), we have

$$\left\langle \mathbf{H}, \left( \prod_{t=1}^{T} \mathscr{G}_t \right) \circ \mathring{\mathcal{B}}_0 \circ \tilde{\mathbf{H}} \right\rangle = \left\langle \mathbf{H}, \left( \prod_{t=1}^{T} (\mathbf{I} - \gamma_t \mathbf{H}\tilde{\mathbf{H}})(\mathbf{\Gamma}_0 - \mathbf{\Gamma}^*) \right)^{\otimes 2} \circ \tilde{\mathbf{H}} \right\rangle$$

$$= \left\langle \mathbf{H}\tilde{\mathbf{H}}, \left( \prod_{t=1}^{T} (\mathbf{I} - \gamma_t \mathbf{H}\tilde{\mathbf{H}})(\mathbf{\Gamma}_0 - \mathbf{\Gamma}^*) \right)^2 \right\rangle. \tag{40}$$

For the second term, we will bound $\sum_{t=0}^{K-1}$ and $\sum_{t=K}^{T}$ separately. For the first part of the sum, we have

$$\sum_{t=0}^{K-1} \gamma_t^2 b_t \mathbf{h}^\top \left( \prod_{k=t+1}^{T} (\mathbf{J} - \gamma_k \mathbf{h}\tilde{\mathbf{h}}^\top)^{\odot 2} \odot (\mathbf{h}\tilde{\mathbf{h}}^\top) \right) \tilde{\mathbf{h}}$$

$$\leq \sum_{t=0}^{K-1} \gamma_t^2 b_t \mathbf{h}^\top \left( \prod_{k=K}^{2K-1} (\mathbf{J} - \gamma_k \mathbf{h}\tilde{\mathbf{h}}^\top)^{\odot 2} \odot (\mathbf{h}\tilde{\mathbf{h}}^\top) \right) \tilde{\mathbf{h}} \qquad \text{since } \mathbf{J} - \gamma_k \mathbf{h}\tilde{\mathbf{h}}^\top \leq \mathbf{J}, \text{ entrywise}$$

$$= \gamma_0^2 \left( \sum_{t=0}^{K-1} b_t \right) \mathbf{h}^\top \left( \left( \mathbf{J} - \frac{\gamma_0}{2} \mathbf{h}\tilde{\mathbf{h}}^\top \right)^{\odot 2K} \odot (\mathbf{h}\tilde{\mathbf{h}}^\top) \right) \tilde{\mathbf{h}}. \quad \text{stepsize is epoch-wise constant}$$

Next, notice that

$$\text{for } 0 < x < 1, \quad (1-x)^{2K} \begin{cases} = (1-x)^K (1-x)^K \leq \frac{1}{Kx} \frac{1}{Kx} = \frac{1}{K^2 x^2}; \\ \leq 1. \end{cases}$$

So we have

$$\left( \mathbf{J} - \frac{\gamma_0}{2} \mathbf{h}\tilde{\mathbf{h}}^\top \right)^{\odot 2K} \odot (\mathbf{h}\tilde{\mathbf{h}}^\top) \leq \min \left\{ \frac{4}{K^2 \gamma_0} (\mathbf{h}\tilde{\mathbf{h}}^\top)^{\odot(-2)}, \mathbf{J} \right\} \odot (\mathbf{h}\tilde{\mathbf{h}}^\top)$$

$$= \min \left\{ \frac{4}{K^2 \gamma_0} (\mathbf{h}\tilde{\mathbf{h}}^\top)^{\odot(-1)}, \mathbf{h}\tilde{\mathbf{h}}^\top \right\},$$

where "min" and "$\leq$" are entrywise. Then we have

$$\mathbf{h}^\top \left( \left( \mathbf{J} - \frac{\gamma_0}{2} \mathbf{h}\tilde{\mathbf{h}}^\top \right)^{\odot 2K} \odot (\mathbf{h}\tilde{\mathbf{h}}^\top) \right) \tilde{\mathbf{h}} \leq \mathbf{h}^\top \min \left\{ \frac{4}{K^2 \gamma_0} (\mathbf{h}\tilde{\mathbf{h}}^\top)^{\odot(-1)}, \mathbf{h}\tilde{\mathbf{h}}^\top \right\} \tilde{\mathbf{h}}$$

$$\leq \frac{4}{K^2 \gamma_0^2} \sum_{i,j} \min \left\{ 1, K^2 \gamma_0^2 \lambda_i^2 \tilde{\lambda}_j^2 \right\},$$

where the last inequality is by the same argument as in the proof of Theorem D.20. Bringing this back, we have

$$\sum_{t=0}^{K-1} \gamma_t^2 b_t \mathbf{h}^\top \left( \prod_{k=t+1}^{T} (\mathbf{J} - \gamma_k \mathbf{h}\tilde{\mathbf{h}}^\top)^{\odot 2} \odot (\mathbf{h}\tilde{\mathbf{h}}^\top) \right) \tilde{\mathbf{h}}$$

$$\leq \gamma_0^2 \left( \sum_{t=0}^{K-1} b_t \right) \mathbf{h}^\top \left( \left( \mathbf{J} - \frac{\gamma_0}{2} \mathbf{h}\tilde{\mathbf{h}}^\top \right)^{\odot 2K} \odot (\mathbf{h}\tilde{\mathbf{h}}^\top) \right) \tilde{\mathbf{h}}$$

$$\leq \left( \sum_{t=0}^{K-1} b_t \right) \frac{4}{K^2} \sum_{i,j} \min \left\{ 1, K^2 \gamma_0^2 \lambda_i^2 \tilde{\lambda}_j^2 \right\}$$

$$\leq \frac{1}{\gamma_0} \big\langle \mathbf{I} - \big(\mathbf{I} - \gamma_0 \mathbf{H}\tilde{\mathbf{H}}\big)^{2K}, \, (\mathbf{\Gamma}_0 - \mathbf{\Gamma}^*)^2 \big\rangle \frac{4}{K^2} \sum_{i,j} \min\big\{1, \, K^2 \gamma_0^2 \lambda_i^2 \tilde{\lambda}_j^2\big\} \qquad \text{by Lemma D.22}$$

$$\leq \frac{1}{\gamma_0} \big\langle \mathbf{I}_{0:k} + 2K\gamma_0 \mathbf{H}_{k:\infty}\tilde{\mathbf{H}}_{k:\infty}, \, (\mathbf{\Gamma}_0 - \mathbf{\Gamma}^*)^2 \big\rangle \frac{4}{K^2} \sum_{i,j} \min\big\{1, \, K^2 \gamma_0^2 \lambda_i^2 \tilde{\lambda}_j^2\big\}$$

$$\leq 8 \big\langle \frac{1}{K\gamma_0} \mathbf{I}_{0:k} + \mathbf{H}_{k:\infty}\tilde{\mathbf{H}}_{k:\infty}, \, (\mathbf{\Gamma}_0 - \mathbf{\Gamma}^*)^2 \big\rangle \frac{1}{K} \sum_{i,j} \min\big\{1, \, K^2 \gamma_0^2 \lambda_i^2 \tilde{\lambda}_j^2\big\}. \tag{41}$$

For the second part of the sum in (39), we have

$$\sum_{t=K}^{T} \gamma_t^2 b_t \mathbf{h}^\top \bigg( \prod_{k=t+1}^{T} \big(\mathbf{J} - \gamma_k \mathbf{h}\tilde{\mathbf{h}}^\top\big)^{\odot 2} \odot \big(\mathbf{h}\tilde{\mathbf{h}}^\top\big) \bigg) \tilde{\mathbf{h}}$$

$$\leq \sum_{t=K}^{T} \gamma_t^2 b_t \mathbf{h}^\top \bigg( \prod_{k=t+1}^{T} \big(\mathbf{J} - \gamma_k \mathbf{h}\tilde{\mathbf{h}}^\top\big) \odot \big(\mathbf{h}\tilde{\mathbf{h}}^\top\big) \bigg) \tilde{\mathbf{h}} \qquad \text{since } \mathbf{J} - \gamma_k \mathbf{h}\tilde{\mathbf{h}}^\top \leq \mathbf{J} \text{ entrywise}$$

$$\leq 4 \big\langle \frac{1}{K\gamma_0} \mathbf{I}_{0:k} + \mathbf{H}_{k:\infty}\tilde{\mathbf{H}}_{k:\infty}, \, (\mathbf{\Gamma}_0 - \mathbf{\Gamma}^*)^2 \big\rangle \sum_{t=K}^{T} \gamma_t^2 \mathbf{h}^\top \bigg( \prod_{k=t+1}^{T} \big(\mathbf{J} - \gamma_k \mathbf{h}\tilde{\mathbf{h}}^\top\big) \odot \big(\mathbf{h}\tilde{\mathbf{h}}^\top\big) \bigg) \tilde{\mathbf{h}},$$

where the last inequality is by Lemma D.24. Notice that the sum in the above display is equivalent to the sum we encountered when analyzing the variance error (see (35) in Lemma D.18), with the only difference being that, here, the sum starts from the second epoch. Therefore, by repeating the arguments made in Lemma D.17 and Theorem D.20 (replacing $\gamma_0$ with $\gamma_0/2$), we have

$$\sum_{t=K}^{T} \gamma_t^2 \mathbf{h}^\top \bigg( \prod_{k=t+1}^{T} \big(\mathbf{J} - \gamma_k \mathbf{h}\tilde{\mathbf{h}}^\top\big)^{\odot 2} \odot \big(\mathbf{h}\tilde{\mathbf{h}}^\top\big) \bigg) \tilde{\mathbf{h}} \leq \frac{8}{K} \sum_{i,j} \min\big\{1, \, K^2 \gamma_0^2 \lambda_i^2 \tilde{\lambda}_j^2\big\}.$$

Bringing this back, we have

$$\sum_{t=K}^{T} \gamma_t^2 b_t \mathbf{h}^\top \bigg( \prod_{k=t+1}^{T} \big(\mathbf{J} - \gamma_k \mathbf{h}\tilde{\mathbf{h}}^\top\big)^{\odot 2} \odot \big(\mathbf{h}\tilde{\mathbf{h}}^\top\big) \bigg) \tilde{\mathbf{h}}$$

$$\leq 4 \big\langle \frac{1}{K\gamma_0} \mathbf{I}_{0:k} + \mathbf{H}_{k:\infty}\tilde{\mathbf{H}}_{k:\infty}, \, (\mathbf{\Gamma}_0 - \mathbf{\Gamma}^*)^2 \big\rangle \frac{8}{K} \sum_{i,j} \min\big\{1, \, K^2 \gamma_0^2 \lambda_i^2 \tilde{\lambda}_j^2\big\}. \tag{42}$$

Finally, putting (40), (41), and (42) in (39), we have

$$b_T \leq \Big\langle \mathbf{H}\tilde{\mathbf{H}}, \, \Big( \prod_{t=1}^{T} \big(\mathbf{I} - \gamma_t \mathbf{H}\tilde{\mathbf{H}}\big)(\mathbf{\Gamma}_0 - \mathbf{\Gamma}^*) \Big)^2 \Big\rangle$$

$$+ 8 \cdot 3^7 \cdot 40 \big\langle \frac{1}{K\gamma_0} \mathbf{I}_{0:k} + \mathbf{H}_{k:\infty}\tilde{\mathbf{H}}_{k:\infty}, \, (\mathbf{\Gamma}_0 - \mathbf{\Gamma}^*)^2 \big\rangle \frac{1}{K} \sum_{i,j} \min\big\{1, \, K^2 \gamma_0^2 \lambda_i^2 \tilde{\lambda}_j^2\big\},$$

which completes the proof. $\qquad\square$

### D.8 PROOF OF THEOREM 4.1

*Proof of Theorem 4.1.* It follows from Theorems D.20 and D.25. $\qquad\square$

### D.9 PROOF OF COROLLARY 4.2

*Proof of Corollary 4.2.* Under the assumptions, we have

$$\frac{\mathtt{tr}(\mathbf{H}) + \sigma^2/\psi^2}{N} \asymp \frac{1}{N}.$$

So we have

$$\mathbf{\Gamma}_N^* := \bigg( \frac{\mathtt{tr}(\mathbf{H}) + \sigma^2/\psi^2}{N} \mathbf{I} + \frac{N+1}{N} \mathbf{H} \bigg)^{-1} \asymp \bigg( \frac{1}{N} \mathbf{I} + \mathbf{H} \bigg)^{-1},$$

and

$$\tilde{\lambda}_j = \psi^2 \lambda_j \left( \frac{\texttt{tr}(\mathbf{H}) + \sigma^2/\psi^2}{N} + \frac{N+1}{N} \lambda_j \right)$$

$$\eqsim \lambda_j \max \left\{ \frac{1}{N}, \; \lambda_j \right\}$$

$$\eqsim \begin{cases} \lambda_j^2, & j \le \ell^*; \\ \lambda_j \frac{1}{N}, & j > \ell^*, \end{cases}$$

where we define

$$\ell^* := \min \left\{ i \ge 0 : \lambda_i \ge \frac{1}{N} \right\}.$$

The excess risk (9) contains two terms. The first term can be bounded by

$$\texttt{Error}_1 := \left\langle \mathbf{H}\tilde{\mathbf{H}}_N, \; \left( \prod_{t=1}^{T} \left( \mathbf{I} - \gamma_t \mathbf{H}\tilde{\mathbf{H}}_N \right) \boldsymbol{\Gamma}_N^* \right)^2 \right\rangle$$

$$\le \texttt{tr}\left( e^{-2T_{\texttt{eff}}\gamma_0 \mathbf{H}\tilde{\mathbf{H}}_N} \mathbf{H}\tilde{\mathbf{H}}_N \left( \boldsymbol{\Gamma}_N^* \right)^2 \right)$$

$$\eqsim \sum_i e^{-2T_{\texttt{eff}}\gamma_0 \lambda_i \tilde{\lambda}_i} \lambda_i \tilde{\lambda}_i \left( \frac{1}{N} + \lambda_i \right)^{-2}$$

$$\eqsim \sum_{i \le \ell^*} e^{-2T_{\texttt{eff}}\gamma_0 \lambda_i^3} \lambda_i^3 \lambda_i^{-2} + \sum_{i > \ell^*} e^{-2T_{\texttt{eff}}\gamma_0 \lambda_i^2 \frac{1}{N}} \lambda_i^2 \frac{1}{N} \cdot N^2$$

$$\eqsim \sum_{i \le \ell^*} e^{-2T_{\texttt{eff}}\gamma_0 \lambda_i^3} \lambda_i + \sum_{i > \ell^*} e^{-2T_{\texttt{eff}}\gamma_0 \lambda_i^2 \frac{1}{N}} \lambda_i^2 N. \tag{43}$$

The second term is

$$\texttt{Error}_2 = \left( \psi^2 \texttt{tr}(\mathbf{H}) + \sigma^2 \right) \frac{D_{\texttt{eff}}}{T_{\texttt{eff}}} \eqsim \frac{D_{\texttt{eff}}}{T_{\texttt{eff}}}. \tag{44}$$

Define

$$\mathbb{K} := \left\{ (i,j) : \lambda_i \tilde{\lambda}_j \ge \frac{1}{T_{\texttt{eff}}\gamma_0} \right\}$$

$$= \left\{ (i,j) : j \le \ell^*, \lambda_i \lambda_j^2 \ge \frac{1}{T_{\texttt{eff}}\gamma_0} \right\} \bigcup \left\{ (i,j) : j > \ell^*, \lambda_i \lambda_j \ge \frac{N}{T_{\texttt{eff}}\gamma_0} \right\},$$

then

$$D_{\texttt{eff}} = \sum_{i,j} \min \left\{ 1, \; \left( T_{\texttt{eff}}\gamma_0 \lambda_i \tilde{\lambda}_j \right)^2 \right\}$$

$$= |\mathbb{K}| + (T_{\texttt{eff}}\gamma_0)^2 \sum_{(i,j) \notin \mathbb{K}} (\lambda_i \tilde{\lambda}_j)^2. \tag{45}$$

**The uniform spectrum.** Here, we assume that $\lambda_i = 1/s$ for $i \le s$ and $\lambda_i = 0$ for $i > s$, and

$$N \le s \le d.$$

So we have that $\tilde{\lambda}_j = 0$ for $j > s$ and

$$\text{for } j \le s, \quad \tilde{\lambda}_j \eqsim \lambda_j \max \left\{ \frac{1}{N}, \; \lambda_j \right\} \eqsim \frac{1}{sN}.$$

Therefore

$$\texttt{tr}(\tilde{\mathbf{H}}) = \frac{1}{N},$$

and

$$\gamma_0 \eqsim \frac{1}{\mathtt{tr}(\tilde{\mathbf{H}})} = N.$$

By (43) we have

$$\begin{aligned}
\mathtt{Error}_1 &\lesssim s e^{-2T_{\mathtt{eff}}\gamma_0 \frac{1}{s^2 N}} \frac{N}{s^2} \\
&= e^{-2T_{\mathtt{eff}}\frac{1}{s^2}} \frac{N}{s} \\
&\lesssim \begin{cases} \dfrac{N}{s}, & T_{\mathtt{eff}} \le s^2 \\ \dfrac{Ns}{T_{\mathtt{eff}}}, & T_{\mathtt{eff}} > s^2. \end{cases}
\end{aligned}$$

By (45), we have

$$\begin{aligned}
D_{\mathtt{eff}} &:= \sum_{i,j} \min\left\{1, \ \left(T_{\mathtt{eff}}\gamma_0 \lambda_i \tilde{\lambda}_j\right)^2\right\} \\
&= s^2 \min\left\{1, \ \left(T_{\mathtt{eff}}\gamma_0 \frac{1}{s^2 N}\right)^2\right\} \\
&= s^2 \min\left\{1, \ \left(T_{\mathtt{eff}} \frac{1}{s^2}\right)^2\right\} \\
&= \begin{cases} \dfrac{T_{\mathtt{eff}}^2}{s^2}, & T_{\mathtt{eff}} \le s^2; \\ s^2, & T_{\mathtt{eff}} > s^2. \end{cases}
\end{aligned}$$

So by (44), we have

$$\mathtt{Error}_2 \eqsim \frac{D_{\mathtt{eff}}}{T_{\mathtt{eff}}} \eqsim \begin{cases} \dfrac{T_{\mathtt{eff}}}{s^2}, & T_{\mathtt{eff}} \le s^2; \\ \dfrac{s^2}{T_{\mathtt{eff}}}, & T_{\mathtt{eff}} > s^2. \end{cases}$$

In sum, we have

$$\begin{aligned}
\mathbb{E}\Delta(\boldsymbol{\Gamma}_T) &= \mathtt{Error}_1 + \mathtt{Error}_2 \\
&\lesssim \begin{cases} \dfrac{T_{\mathtt{eff}}}{s^2} + \dfrac{N}{s}, & T_{\mathtt{eff}} \le s^2; \\ \dfrac{s^2}{T_{\mathtt{eff}}} + \dfrac{Ns}{T_{\mathtt{eff}}} \eqsim \dfrac{s^2}{T_{\mathtt{eff}}}, & T_{\mathtt{eff}} > s^2. \end{cases}
\end{aligned}$$

**The polynomial spectrum.** Here, we assume $\lambda_i = i^{-a}$ for $a > 1$. Then

$$\ell^* = N^{\frac{1}{a}},$$

and

$$\tilde{\lambda}_j \eqsim \begin{cases} j^{-2a}, & j \le N^{\frac{1}{a}}; \\ j^{-a}N^{-1}, & j > N^{\frac{1}{a}}. \end{cases}$$

Therefore

$$\mathtt{tr}(\tilde{\mathbf{H}}) = \sum_j \tilde{\lambda}_j \eqsim 1,$$

and

$$\gamma_0 \eqsim \frac{1}{\mathtt{tr}(\tilde{\mathbf{H}})} \eqsim 1.$$

By (43), we have

$$
\begin{aligned}
\mathtt{Error}_1 &\lesssim \sum_{i \le \ell^*} e^{-2T_{\mathtt{eff}}\gamma_0 \lambda_i^3} \lambda_i + \sum_{i > \ell^*} e^{-2T_{\mathtt{eff}}\gamma_0 \lambda_i^2 \frac{1}{N}} \lambda_i^2 N \\
&\eqsim \sum_{i \le N^{\frac{1}{a}}} e^{-2T_{\mathtt{eff}}\gamma_0 i^{-3a}} i^{-a} + \sum_{i > N^{\frac{1}{a}}} e^{-2T_{\mathtt{eff}}\gamma_0 i^{-2a} \frac{1}{N}} i^{-2a} N \\
&\eqsim \sum_{i \le N^{\frac{1}{a}}} e^{-2T_{\mathtt{eff}}\gamma_0 i^{-3a}} i^{-a} + \sum_{N^{\frac{1}{a}} < i \le (T_{\mathtt{eff}}\gamma_0/N)^{\frac{1}{2a}}} e^{-2T_{\mathtt{eff}}\gamma_0 i^{-2a} \frac{1}{N}} i^{-2a} N \\
&\qquad + \sum_{i > (T_{\mathtt{eff}}\gamma_0/N)^{\frac{1}{2a}}} e^{-2T_{\mathtt{eff}}\gamma_0 i^{-2a} \frac{1}{N}} i^{-2a} N \\
&\lesssim \sum_{i \le N^{\frac{1}{a}}} e^{-2T_{\mathtt{eff}}\gamma_0 N^{-3}} i^{-a} + \sum_{N^{\frac{1}{a}} < i \le (T_{\mathtt{eff}}\gamma_0/N)^{\frac{1}{2a}}} e^{-\frac{2T_{\mathtt{eff}}\gamma_0 i^{-2a}}{N}} \frac{2T_{\mathtt{eff}}\gamma_0 i^{-2a}}{N} \frac{N^2}{2T_{\mathtt{eff}}\gamma_0} \\
&\qquad + \sum_{i > (T_{\mathtt{eff}}\gamma_0/N)^{\frac{1}{2a}}} i^{-2a} N \\
&\lesssim e^{-2T_{\mathtt{eff}}\gamma_0 N^{-3}} + \frac{N^2}{T_{\mathtt{eff}}\gamma_0} \int_1^\infty (e^{-t} t) \mathrm{d}t + \left(\frac{T_{\mathtt{eff}}\gamma_0}{N}\right)^{\frac{1-2a}{2a}} N \\
&\eqsim e^{-2T_{\mathtt{eff}}\gamma_0 N^{-3}} + \frac{N^2}{T_{\mathtt{eff}}\gamma_0} + \left(\frac{N}{T_{\mathtt{eff}}\gamma_0}\right)^{1-\frac{1}{2a}} N \\
&\eqsim \left(\frac{N}{T_{\mathtt{eff}}}\right)^{1-\frac{1}{2a}} N,
\end{aligned}
$$

where the last inequality is because

$$
\gamma_0 \eqsim 1
$$

and the assumption

$$
N^3 = o(T_{\mathtt{eff}}).
$$

The first part in (45) is

$$
\begin{aligned}
|\mathbb{K}| &= \left| \left\{ (i,j) : j \le \ell^*, \lambda_i \lambda_j^2 \ge \frac{1}{T_{\mathtt{eff}}\gamma_0} \right\} \right| + \left| \left\{ (i,j) : j > \ell^*, \lambda_i \lambda_j \ge \frac{N}{T_{\mathtt{eff}}\gamma_0} \right\} \right| \\
&= \left| \left\{ (i,j) : j \le N^{\frac{1}{a}}, ij^2 \le (T_{\mathtt{eff}}\gamma_0)^{\frac{1}{a}} \right\} \right| + \left| \left\{ (i,j) : j > N^{\frac{1}{a}}, ij \le \left(\frac{T_{\mathtt{eff}}\gamma_0}{N}\right)^{\frac{1}{a}} \right\} \right| \\
&\eqsim \sum_{1 \le j \le N^{\frac{1}{a}}} \frac{(T_{\mathtt{eff}}\gamma_0)^{\frac{1}{a}}}{j^2} + \sum_{N^{\frac{1}{a}} < j \le (T_{\mathtt{eff}}\gamma_0/N)^{\frac{1}{a}}} \frac{(T_{\mathtt{eff}}\gamma_0/N)^{\frac{1}{a}}}{j} \\
&\eqsim (T_{\mathtt{eff}}\gamma_0)^{\frac{1}{a}} + \left(\frac{T_{\mathtt{eff}}\gamma_0}{N}\right)^{\frac{1}{a}} \log\left(\frac{T_{\mathtt{eff}}\gamma_0}{N}\right).
\end{aligned}
$$

The sum in the second part in (45) is

$$
\begin{aligned}
\sum_{(i,j) \notin \mathbb{K}} (\lambda_i \tilde{\lambda}_j)^2 &= \sum_{j \le N^{\frac{1}{a}}, ij^2 > (T_{\mathtt{eff}}\gamma_0)^{\frac{1}{a}}} (\lambda_i \tilde{\lambda}_j)^2 + \sum_{j > N^{\frac{1}{a}}, ij > (T_{\mathtt{eff}}\gamma_0/N)^{\frac{1}{a}}} (\lambda_i \tilde{\lambda}_j)^2 \\
&= \sum_{j \le N^{\frac{1}{a}}, ij^2 > (T_{\mathtt{eff}}\gamma_0)^{\frac{1}{a}}} (\lambda_i \lambda_j^2)^2 + \sum_{j > N^{\frac{1}{a}}, ij > (T_{\mathtt{eff}}\gamma_0/N)^{\frac{1}{a}}} (\lambda_i \lambda_j/N)^2 \\
&= \sum_{j \le N^{\frac{1}{a}}, ij^2 > (T_{\mathtt{eff}}\gamma_0)^{\frac{1}{a}}} i^{-2a} j^{-4a} + \sum_{j > N^{\frac{1}{a}}, ij > (T_{\mathtt{eff}}\gamma_0/N)^{\frac{1}{a}}} i^{-2a} j^{-2a} N^{-2} \\
&= \sum_{j \le N^{\frac{1}{a}}} j^{-4a} \sum_{i > (T_{\mathtt{eff}}\gamma_0)^{\frac{1}{a}}/j^2} i^{-2a} + \sum_{j > N^{\frac{1}{a}}} j^{-2a} N^{-2} \sum_{i \ge 1, i > (T_{\mathtt{eff}}\gamma_0/N)^{\frac{1}{a}}/j} i^{-2a}
\end{aligned}
$$

$$\asymp \sum_{j \le N^{\frac{1}{a}}} j^{-4a} \left( (T_{\texttt{eff}} \gamma_0)^{\frac{1}{a}} / j^2 \right)^{1-2a}$$

$$+ \sum_{j > N^{\frac{1}{a}}} j^{-2a} N^{-2} \max \left\{ 1, \left( (T_{\texttt{eff}} \gamma_0 / N)^{\frac{1}{a}} / j \right)^{1-2a} \right\}$$

$$\asymp \sum_{j \le N^{\frac{1}{a}}} j^{-2} (T_{\texttt{eff}} \gamma_0)^{\frac{1-2a}{a}} + \sum_{j > (T_{\texttt{eff}} \gamma_0 / N)^{\frac{1}{a}}} j^{-2a} N^{-2}$$

$$+ \sum_{N^{\frac{1}{a}} < j \le (T_{\texttt{eff}} \gamma_0 / N)^{\frac{1}{a}}} j^{-2a} N^{-2} \left( (T_{\texttt{eff}} \gamma_0 / N)^{\frac{1}{a}} / j \right)^{1-2a}$$

$$\asymp (T_{\texttt{eff}} \gamma_0)^{\frac{1-2a}{a}} + \left( \frac{T_{\texttt{eff}} \gamma_0}{N} \right)^{\frac{1-2a}{a}} N^{-2}$$

$$+ \sum_{N^{\frac{1}{a}} < j \le (T_{\texttt{eff}} \gamma_0 / N)^{\frac{1}{a}}} j^{-1} N^{-2} \left( \frac{T_{\texttt{eff}} \gamma_0}{N} \right)^{\frac{1-2a}{a}}$$

$$\asymp (T_{\texttt{eff}} \gamma_0)^{\frac{1-2a}{a}} + (T_{\texttt{eff}} \gamma_0)^{\frac{1-2a}{a}} N^{-\frac{1}{a}} + (T_{\texttt{eff}} \gamma_0)^{\frac{1-2a}{a}} N^{-\frac{1}{a}} \log \left( \frac{T_{\texttt{eff}} \gamma_0}{N} \right)$$

$$\asymp (T_{\texttt{eff}} \gamma_0)^{\frac{1-2a}{a}} + (T_{\texttt{eff}} \gamma_0)^{\frac{1-2a}{a}} N^{-\frac{1}{a}} \log \left( \frac{T_{\texttt{eff}} \gamma_0}{N} \right).$$

So the effective dimension (45) is

$$D_{\texttt{eff}} = |\mathbb{K}| + (T_{\texttt{eff}} \gamma_0)^2 \sum_{(i,j) \notin \mathbb{K}} (\lambda_i \tilde{\lambda}_j)^2$$

$$\asymp (T_{\texttt{eff}} \gamma_0)^{\frac{1}{a}} + \left( \frac{T_{\texttt{eff}} \gamma_0}{N} \right)^{\frac{1}{a}} \log \left( \frac{T_{\texttt{eff}} \gamma_0}{N} \right)$$

$$+ (T_{\texttt{eff}} \gamma_0)^2 \left( (T_{\texttt{eff}} \gamma_0)^{\frac{1-2a}{a}} + (T_{\texttt{eff}} \gamma_0)^{\frac{1-2a}{a}} N^{-\frac{1}{a}} \log \left( \frac{T_{\texttt{eff}} \gamma_0}{N} \right) \right)$$

$$\asymp (T_{\texttt{eff}} \gamma_0)^{\frac{1}{a}} + \left( \frac{T_{\texttt{eff}} \gamma_0}{N} \right)^{\frac{1}{a}} \log \left( \frac{T_{\texttt{eff}} \gamma_0}{N} \right).$$

Therefore (44) is

$$\texttt{Error}_2 \asymp \frac{D_{\texttt{eff}}}{T_{\texttt{eff}}}$$

$$\lesssim T_{\texttt{eff}}^{-1} \left( (T_{\texttt{eff}} \gamma_0)^{\frac{1}{a}} + \left( \frac{T_{\texttt{eff}} \gamma_0}{N} \right)^{\frac{1}{a}} \log \left( \frac{T_{\texttt{eff}} \gamma_0}{N} \right) \right)$$

$$\asymp T_{\texttt{eff}}^{\frac{1}{a} - 1} \left( 1 + N^{-\frac{1}{a}} \log(T_{\texttt{eff}}) \right),$$

where the last inequality is because

$$\gamma_0 \asymp 1$$

and the assumption

$$N^3 = o(T_{\texttt{eff}}).$$

Putting the two error terms together, we have

$$\mathbb{E} \Delta(\Gamma_T) \lesssim \texttt{Error}_1 + \texttt{Error}_2$$

$$\lesssim \left( \frac{N}{T_{\texttt{eff}}} \right)^{1 - \frac{1}{2a}} N + T_{\texttt{eff}}^{\frac{1}{a} - 1} \left( 1 + N^{-\frac{1}{a}} \log(T_{\texttt{eff}}) \right)$$

$$\asymp T_{\texttt{eff}}^{\frac{1}{a} - 1} \left( 1 + N^{-\frac{1}{a}} \log(T_{\texttt{eff}}) + T_{\texttt{eff}}^{-\frac{1}{2a}} N^{2 - \frac{1}{2a}} \right).$$

**The exponential spectrum.** Here, we assume $\lambda_i = 2^{-i}$. Then

$$\ell^* = \log(N),$$

and

$$\tilde{\lambda}_j \approx \begin{cases} 2^{-2j}, & j \le \log(N); \\ 2^{-j}N^{-1}, & j > \log(N). \end{cases}$$

Therefore

$$\mathtt{tr}(\tilde{\mathbf{H}}) = \sum_j \tilde{\lambda}_j = 1,$$

and

$$\gamma_0 \approx \frac{1}{\mathtt{tr}(\tilde{\mathbf{H}})} \approx 1.$$

By (43), we have

$$\begin{aligned}
\mathtt{Error}_1 &\lesssim \sum_{i \le \ell^*} e^{-2T_{\mathtt{eff}}\gamma_0\lambda_i^3}\lambda_i + \sum_{i > \ell^*} e^{-2T_{\mathtt{eff}}\gamma_0\lambda_i^2\frac{1}{N}}\lambda_i^2 N \\
&\approx \sum_{i \le \log(N)} e^{-2T_{\mathtt{eff}}\gamma_0 2^{-3i}}2^{-i} + \sum_{i > \log(N)} e^{-2T_{\mathtt{eff}}\gamma_0 2^{-2i}N^{-1}}2^{-2i}N \\
&\approx \sum_{i \le \log(N)} e^{-2T_{\mathtt{eff}}\gamma_0 2^{-3i}}2^{-i} \\
&\quad + \sum_{\log(N) < i \le \log(2T_{\mathtt{eff}}\gamma_0/N)/2} e^{-\frac{2T_{\mathtt{eff}}\gamma_0 2^{-2i}}{N}}\frac{2T_{\mathtt{eff}}\gamma_0 2^{-2i}}{N}\frac{N^2}{2T_{\mathtt{eff}}\gamma_0} \\
&\quad + \sum_{i > \log(2T_{\mathtt{eff}}\gamma_0/N)/2} e^{-2T_{\mathtt{eff}}\gamma_0 2^{-2i}N^{-1}}2^{-2i}N \\
&\lesssim \sum_{i \le \log(N)} e^{-2T_{\mathtt{eff}}\gamma_0 N^{-3}}2^{-i} + \frac{N^2}{2T_{\mathtt{eff}}\gamma_0}\int_1^\infty (e^{-t}t)\mathrm{d}t \\
&\quad + \sum_{i > \log(2T_{\mathtt{eff}}\gamma_0/N)/2} 2^{-2i}N \\
&\lesssim e^{-2T_{\mathtt{eff}}\gamma_0 N^{-3}} + \frac{N^2}{2T_{\mathtt{eff}}\gamma_0} + \frac{N}{2T_{\mathtt{eff}}\gamma_0}N \\
&\approx \frac{N^2}{T_{\mathtt{eff}}},
\end{aligned}$$

where the last inequality is because

$$\gamma_0 \approx 1$$

and the assumption

$$N^3 = o(T_{\mathtt{eff}}).$$

The first part in (45) is

$$\begin{aligned}
|\mathbb{K}| &= \left|\left\{(i,j): j \le \ell^*, \lambda_i\lambda_j^2 \ge \frac{1}{T_{\mathtt{eff}}\gamma_0}\right\}\right| + \left|\left\{(i,j): j > \ell^*, \lambda_i\lambda_j \ge \frac{N}{T_{\mathtt{eff}}\gamma_0}\right\}\right| \\
&= \left|\left\{(i,j): j \le \log(N), i + 2j \le \log(T_{\mathtt{eff}}\gamma_0)\right\}\right| \\
&\quad + \left|\left\{(i,j): j > \log(N), i + j \le \log\left(\frac{T_{\mathtt{eff}}\gamma_0}{N}\right)\right\}\right| \\
&\approx \sum_{1 \le j \le \log(N)} \left(\log(T_{\mathtt{eff}}\gamma_0) - 2j\right) + \sum_{\log(N) < j \le \log(T_{\mathtt{eff}}\gamma_0/N)} \left(\log(T_{\mathtt{eff}}\gamma_0/N) - j\right) \\
&\approx \log(T_{\mathtt{eff}}\gamma_0)\log(N) + \log^2(T_{\mathtt{eff}}\gamma_0/N).
\end{aligned}$$

The sum in the second part in (45) is

$$\sum_{(i,j)\notin\mathbb{K}}(\lambda_i\tilde{\lambda}_j)^2$$

$$= \sum_{j\leq\log(N),i+2j>\log(T_{\text{eff}}\gamma_0)}(\lambda_i\tilde{\lambda}_j)^2 + \sum_{j>\log(N),i+j>\log(T_{\text{eff}}\gamma_0/N)}(\lambda_i\tilde{\lambda}_j)^2$$

$$= \sum_{j\leq\log(N),i+2j>\log(T_{\text{eff}}\gamma_0)}2^{-2(i+2j)} + \sum_{j>\log(N),i+j>\log(T_{\text{eff}}\gamma_0/N)}2^{-2(i+j)}N^{-2}$$

$$= \sum_{j\leq\log(N)}2^{-4j}\sum_{i>\log(T_{\text{eff}}\gamma_0)-2j}2^{-2i} + \sum_{j>\log(N)}2^{-2j}N^{-2}\sum_{i\geq1,i>\log(T_{\text{eff}}\gamma_0/N)-j}2^{-2i}$$

$$\approx \sum_{j\leq\log(N)}2^{-4j}(T_{\text{eff}}\gamma_0)^{-2}2^{4j}$$

$$+ \sum_{\log(N)\leq j<\log(T_{\text{eff}}\gamma_0/N)}2^{-2j}N^{-2}\sum_{i\geq\log(T_{\text{eff}}\gamma_0/N)-j}2^{-2j}$$

$$+ \sum_{j\geq\log(T_{\text{eff}}\gamma_0/N)}2^{-2j}N^{-2}\sum_{i\geq1}2^{-2j}$$

$$\approx (T_{\text{eff}}\gamma_0)^{-2}\log(N) + \sum_{\log(N)\leq j<\log(T_{\text{eff}}\gamma_0/N)}2^{-2j}N^{-2}(T_{\text{eff}}\gamma_0/N)^{-2}2^{2j}$$

$$+ \sum_{j\geq\log(T_{\text{eff}}\gamma_0/N)}2^{-2j}N^{-2}$$

$$\approx (T_{\text{eff}}\gamma_0)^{-2}\log(N) + N^{-2}(T_{\text{eff}}\gamma_0/N)^{-2}\log(T_{\text{eff}}\gamma_0/N) + N^{-2}(T_{\text{eff}}\gamma_0/N)^{-2}$$

$$\approx (T_{\text{eff}}\gamma_0)^{-2}\big(\log(N) + \log(T_{\text{eff}}\gamma_0/N)\big).$$

So the effective dimension (45) is

$$D_{\text{eff}} = |\mathbb{K}| + (T_{\text{eff}}\gamma_0)^2\sum_{(i,j)\notin\mathbb{K}}(\lambda_i\tilde{\lambda}_j)^2$$

$$\approx \log(T_{\text{eff}}\gamma_0)\log(N) + \log^2(T_{\text{eff}}\gamma_0/N) + \big(\log(N) + \log(T_{\text{eff}}\gamma_0/N)\big)$$

$$\approx \log(T_{\text{eff}}\gamma_0)\log(N) + \log^2(T_{\text{eff}}\gamma_0/N).$$

Therefore (44) is

$$\texttt{Error}_2 \approx \frac{D_{\text{eff}}}{T_{\text{eff}}}$$

$$\lesssim T_{\text{eff}}^{-1}\big(\log(T_{\text{eff}}\gamma_0)\log(N) + \log^2(T_{\text{eff}}\gamma_0/N)\big)$$

$$\approx T_{\text{eff}}^{-1}\log^2(T_{\text{eff}}),$$

where the last inequality is because

$$\gamma_0 \approx 1$$

and the assumption

$$N^3 = o(T_{\text{eff}}).$$

Putting the two error terms together, we have

$$\mathbb{E}\Delta(\boldsymbol{\Gamma}_T) \lesssim \texttt{Error}_1 + \texttt{Error}_2$$

$$\lesssim \frac{N^2}{T_{\text{eff}}} + T_{\text{eff}}^{-1}\log^2(T_{\text{eff}})$$

$$\approx \frac{N^2 + \log^2(T_{\text{eff}})}{T_{\text{eff}}}.$$

We have completed the proof. $\qquad\square$

# E   A COMPARISON BETWEEN THE PRETRAINED ATTENTION MODEL AND OPTIMAL RIDGE REGRESSION

## E.1   PROOF OF PROPOSITION 5.1

*Proof of Proposition 5.1.* We start with (11). We have

$$\mathcal{L}(h; \mathbf{X}) = \mathbb{E}\big[\big(h(\mathbf{X}, \mathbf{y}, \mathbf{x}) - y\big)^2 \mid \mathbf{X}\big]$$
$$= \mathbb{E}\big[\big(h(\mathbf{X}, \mathbf{y}, \mathbf{x}) - \mathbb{E}[y|\mathbf{X}, \mathbf{y}, \mathbf{x}]\big)^2 \mid \mathbf{X}\big] + \mathbb{E}\big[\big(\mathbb{E}[y|\mathbf{X}, \mathbf{y}, \mathbf{x}] - y\big)^2 \mid \mathbf{X}\big],$$

where the second term is independent of $h$. Therefore, the minimizer of $\mathcal{L}$ must be

$$h(\mathbf{X}, \mathbf{y}, \mathbf{x}) = \mathbb{E}[y|\mathbf{X}, \mathbf{y}, \mathbf{x}].$$

Recall from Assumption 1 that

$$y \sim \mathcal{N}(\mathbf{x}^\top \boldsymbol{\beta}, \sigma^2),$$

so we have

$$h(\mathbf{X}, \mathbf{y}, \mathbf{x}) = \mathbb{E}[\mathbf{x}^\top \boldsymbol{\beta}|\mathbf{X}, \mathbf{y}, \mathbf{x}]$$
$$= \big\langle \mathbb{E}[\boldsymbol{\beta}|\mathbf{X}, \mathbf{y}], \ \mathbf{x}\big\rangle.$$

By Bayes' theorem, we have

$$\mathbb{P}(\boldsymbol{\beta}|\mathbf{X}, \mathbf{y}) = \frac{\mathbb{P}(\mathbf{y}|\mathbf{X}, \boldsymbol{\beta})\mathbb{P}(\boldsymbol{\beta})}{\int \mathbb{P}(\mathbf{y}|\mathbf{X}, \boldsymbol{\beta})\mathbb{P}(\boldsymbol{\beta}) \, \mathrm{d}\boldsymbol{\beta}}.$$

Recall from Assumption 1 that

$$\mathbf{y} \sim \mathcal{N}(\mathbf{X}\boldsymbol{\beta}, \sigma^2\mathbf{I}), \quad \boldsymbol{\beta} \sim \mathcal{N}(0, \psi^2\mathbf{I}),$$

so we know $\mathbb{P}(\boldsymbol{\beta}|\mathbf{X}, \mathbf{y})$ must be a Gaussian distribution and that

$$\mathbb{P}(\boldsymbol{\beta}|\mathbf{X}, \mathbf{y}) \propto \mathbb{P}(\mathbf{y}|\mathbf{X}, \boldsymbol{\beta})\mathbb{P}(\boldsymbol{\beta})$$
$$\propto \exp\bigg( - \frac{\|\mathbf{y} - \mathbf{X}\boldsymbol{\beta}\|_2^2}{2\sigma^2} \bigg) \exp\bigg( - \frac{\|\boldsymbol{\beta}\|_2^2}{2\psi^2} \bigg),$$

which implies that (because the mean of a Gaussian random variable maximizes its density)

$$\mathbb{E}[\boldsymbol{\beta}|\mathbf{X}, \mathbf{y}] = \arg\min_{\boldsymbol{\mu}} \frac{\|\mathbf{y} - \mathbf{X}\boldsymbol{\mu}\|_2^2}{2\sigma^2} + \frac{\|\boldsymbol{\mu}\|_2^2}{2\psi^2}$$
$$= \big(\mathbf{X}^\top\mathbf{X} + \sigma^2/\psi^2\mathbf{I}\big)^{-1}\mathbf{X}^\top\mathbf{y}.$$

Putting everything together, we obtain that

$$h(\mathbf{X}, \mathbf{y}, \mathbf{x}) = \big\langle \mathbb{E}[\boldsymbol{\beta}|\mathbf{X}, \mathbf{y}], \ \mathbf{x}\big\rangle$$
$$= \big\langle \big(\mathbf{X}^\top\mathbf{X} + \sigma^2/\psi^2\mathbf{I}\big)^{-1}\mathbf{X}^\top\mathbf{y}, \ \mathbf{x}\big\rangle,$$

which concludes the proof.   □

## E.2   PROOF OF COROLLARY 5.2

*Proof of Corollary 5.2.* Let $\boldsymbol{\beta}$ be the sampled task parameter and $\hat{\boldsymbol{\beta}}$ be the ridge estimator in (12), that is,

$$\hat{\boldsymbol{\beta}} := \big(\mathbf{X}^\top\mathbf{X} + \sigma^2/\psi^2\mathbf{I}\big)^{-1}\mathbf{X}^\top\mathbf{y}.$$

By Assumption 1, we have

$$\mathbf{y}_i \sim \mathcal{N}(\boldsymbol{\beta}^\top\mathbf{x}_i, \sigma^2), \quad \mathbf{x}_i \sim \mathcal{N}(0, \mathbf{H}), \quad \boldsymbol{\beta} \sim \mathcal{N}(0, \psi^2\mathbf{I}),$$

which allows us to apply the upper and lower bound for ridge regression in Tsigler & Bartlett (2023), then we have that, with probability at least $1 - e^{-\Omega(M)}$ over the randomness of $\mathbf{X}$, it holds that

$$\mathbb{E}_{\text{sign}}\|\hat{\boldsymbol{\beta}} - \boldsymbol{\beta}\|_{\mathbf{H}}^2 \approx \bigg( \frac{\sigma^2/\psi^2 + \sum_{i>k^*} \lambda_i}{M} \bigg)^2 \|\boldsymbol{\beta}\|_{\mathbf{H}_{0:k^*}^{-1}}^2 + \|\boldsymbol{\beta}\|_{\mathbf{H}_{k^*:\infty}}^2$$

$$+ \frac{\sigma^2}{M}\left(k^* + \left(\frac{M}{\sigma^2/\psi^2 + \sum_{i>k}\lambda_i}\right)^2 \sum_{i>k^*}\lambda_i^2\right),$$

where $\mathbb{E}_{\text{sign}}$ refers to taking expectation over the sign flipping randomness of $\boldsymbol{\beta}$ and

$$k^* := \min\left\{k : \lambda_k \geq c\frac{\sigma^2/\psi^2 + \sum_{i>k}\lambda_i}{M}\right\},$$

where $c > 1$ is an absolute constant. Now, taking the expectation over the Gaussian prior of $\boldsymbol{\beta}$, we have

$$\mathcal{L}(h; \mathbf{X}) - \sigma^2 = \mathbb{E}_{\boldsymbol{\beta}\sim\mathcal{N}(0,\psi^2\mathbf{I})}\|\hat{\boldsymbol{\beta}} - \boldsymbol{\beta}\|_{\mathbf{H}}^2$$
$$\eqsim \left(\frac{\sigma^2/\psi^2 + \sum_{i>k^*}\lambda_i}{M}\right)^2 \psi^2 \sum_{i\leq k^*}\frac{1}{\lambda_i} + \psi^2 \sum_{i>k^*}\lambda_i^2$$
$$+ \frac{\sigma^2}{M}\left(k^* + \left(\frac{M}{\sigma^2/\psi^2 + \sum_{i>k}\lambda_i}\right)^2 \sum_{i>k^*}\lambda_i^2\right).$$

Denote

$$\tilde{\lambda} := c\frac{\sigma^2/\psi^2 + \sum_{i>k}\lambda_i}{M} \eqsim \frac{\sigma^2/\psi^2}{M},$$

then we have

$$k^* := \min\{k : \lambda_k \geq \tilde{\lambda}\},$$

so we have

$$\mathcal{L}(h; \mathbf{X}) - \sigma^2 \eqsim \psi^2\tilde{\lambda}^2 \sum_{i\leq k^*}\frac{1}{\lambda_i} + \psi^2 \sum_{i>k^*}\lambda_i^2 + \frac{\sigma^2}{M}\left(k^* + \frac{1}{\tilde{\lambda}^2}\sum_{i>k^*}\lambda_i^2\right)$$
$$\eqsim \psi^2 \sum_i \min\left\{\frac{\tilde{\lambda}^2}{\lambda_i}, \lambda_i\right\} + \frac{\sigma^2}{M}\sum_i \min\left\{1, \frac{\lambda_i^2}{\tilde{\lambda}^2}\right\}$$
$$\eqsim \psi^2 \sum_i \min\left\{\frac{\tilde{\lambda}^2}{\lambda_i}, \lambda_i\right\} + \psi^2\tilde{\lambda} \sum_i \min\left\{1, \frac{\lambda_i^2}{\tilde{\lambda}^2}\right\}$$
$$\eqsim \psi^2 \sum_i \left(\min\left\{\frac{\tilde{\lambda}^2}{\lambda_i}, \lambda_i\right\} + \min\left\{\tilde{\lambda}, \frac{\lambda_i^2}{\tilde{\lambda}}\right\}\right)$$
$$\eqsim \psi^2 \sum_i \min\{\tilde{\lambda}, \lambda_i\}.$$

This completes the proof. $\qquad\qquad\square$

### E.3 PROOF OF THEOREM 5.3

*Proof of Theorem 5.3.* Consider the attention estimator (10) and its induced average risk (11), we have

$$\mathbb{E}\mathcal{L}(f; \mathbf{X}) = \mathbb{E}\left(\left\langle \mathbf{x}, \boldsymbol{\Gamma}_N^* \frac{1}{M}\mathbf{X}^\top\mathbf{y}\right\rangle - y\right)^2$$
$$= \mathcal{R}_M(\boldsymbol{\Gamma}_N^*).$$

Therefore, we can apply Theorem 3.1 and obtain

$$\mathbb{E}\mathcal{L}(\hat{\boldsymbol{\beta}}_N; \mathbf{X}) = \mathcal{R}_M(\boldsymbol{\Gamma}_N^*)$$
$$= \left\langle \mathbf{H}, (\boldsymbol{\Gamma}_N^* - \boldsymbol{\Gamma}_M^*)\tilde{\mathbf{H}}_M(\boldsymbol{\Gamma}_N^* - \boldsymbol{\Gamma}_M^*)^\top\right\rangle + \min\mathcal{R}_M(\cdot)$$
$$= \left\langle \mathbf{H}\tilde{\mathbf{H}}_M, (\boldsymbol{\Gamma}_N^* - \boldsymbol{\Gamma}_M^*)^2\right\rangle + \min\mathcal{R}_M(\cdot).$$

For the second term, we have

$$\min\mathcal{R}_M(\cdot) - \sigma^2$$

$$= \psi^2 \mathrm{tr}\Big(\big((\mathrm{tr}(\mathbf{H}) + \sigma^2/\psi^2)\mathbf{H}^{-1} + (M+1)\mathbf{I}\big)^{-1}\big((\mathrm{tr}(\mathbf{H}) + \sigma^2/\psi^2)\mathbf{I} + \mathbf{H}\big)\Big)$$

$$\approx \psi^2 \mathrm{tr}\Big(\big((\mathrm{tr}(\mathbf{H}) + \sigma^2/\psi^2)\mathbf{H}^{-1} + (M+1)\mathbf{I}\big)^{-1}\big(2(\mathrm{tr}(\mathbf{H}) + \sigma^2/\psi^2)\mathbf{I}\big)\Big)$$

$$\approx 2\big(\psi^2\mathrm{tr}(\mathbf{H}) + \sigma^2\big)\mathrm{tr}\Big(\big((\mathrm{tr}(\mathbf{H}) + \sigma^2/\psi^2)\mathbf{H}^{-1} + M\mathbf{I}\big)^{-1}\Big)$$

$$= 2\psi^2\tilde{\lambda}_M \sum_i \frac{1}{\tilde{\lambda}_M/\lambda_i + 1}$$

$$\approx 2\psi^2 \sum_i \min\{\tilde{\lambda}_M,\ \lambda_i\},$$

where we define

$$\tilde{\lambda}_M := \frac{\mathrm{tr}(\mathbf{H}) + \sigma^2/\psi^2}{M} \approx \frac{\sigma^2/\psi^2}{M}.$$

For the first term, note that

$$\boldsymbol{\Gamma}_N^* - \boldsymbol{\Gamma}_M^*$$

$$= \left(\frac{\mathrm{tr}(\mathbf{H}) + \sigma^2/\psi^2}{N}\mathbf{I} + \frac{N+1}{N}\mathbf{H}\right)^{-1} - \left(\frac{\mathrm{tr}(\mathbf{H}) + \sigma^2/\psi^2}{M}\mathbf{I} + \frac{M+1}{M}\mathbf{H}\right)^{-1}$$

$$= \left(\frac{1}{M} - \frac{1}{N}\right)\big(\mathrm{tr}(\mathbf{H}) + \sigma^2/\psi^2\big)$$

$$\qquad \left(\frac{\mathrm{tr}(\mathbf{H}) + \sigma^2/\psi^2}{N}\mathbf{I} + \frac{N+1}{N}\mathbf{H}\right)^{-1}\left(\frac{\mathrm{tr}(\mathbf{H}) + \sigma^2/\psi^2}{M}\mathbf{I} + \frac{M+1}{M}\mathbf{H}\right)^{-1}$$

$$= \big(\tilde{\lambda}_M - \tilde{\lambda}_N\big)\left(\tilde{\lambda}_N\mathbf{I} + \frac{N+1}{N}\mathbf{H}\right)^{-1}\left(\tilde{\lambda}_M\mathbf{I} + \frac{M+1}{M}\mathbf{H}\right)^{-1}.$$

So the first term can be bounded by

$$\big\langle \mathbf{H}\tilde{\mathbf{H}}_M,\ \big(\boldsymbol{\Gamma}_N^* - \boldsymbol{\Gamma}_M^*\big)^2\big\rangle$$

$$= \psi^2\left\langle \mathbf{H}^2\left(\frac{\mathrm{tr}(\mathbf{H}) + \sigma^2/\psi^2}{M}\mathbf{I} + \frac{M+1}{M}\mathbf{H}\right),\ \big(\boldsymbol{\Gamma}_N^* - \boldsymbol{\Gamma}_M^*\big)^2\right\rangle$$

$$= \psi^2\big(\tilde{\lambda}_M - \tilde{\lambda}_N\big)^2\left\langle \mathbf{H}^2\left(\tilde{\lambda}_M\mathbf{I} + \frac{M+1}{M}\mathbf{H}\right),\ \left(\tilde{\lambda}_N\mathbf{I} + \frac{N+1}{N}\mathbf{H}\right)^{-2}\left(\tilde{\lambda}_M\mathbf{I} + \frac{M+1}{M}\mathbf{H}\right)^{-2}\right\rangle$$

$$\approx \psi^2\big(\tilde{\lambda}_M - \tilde{\lambda}_N\big)^2\mathrm{tr}\Big(\mathbf{H}^2\big(\tilde{\lambda}_N\mathbf{I} + \mathbf{H}\big)^{-2}\big(\tilde{\lambda}_M\mathbf{I} + \mathbf{H}\big)^{-1}\Big)$$

$$\approx \psi^2\big(\tilde{\lambda}_M - \tilde{\lambda}_N\big)^2\sum_i \lambda_i^2\min\left\{\frac{1}{\tilde{\lambda}_N^2},\ \frac{1}{\lambda_i^2}\right\}\min\left\{\frac{1}{\tilde{\lambda}_M},\ \frac{1}{\lambda_i}\right\}$$

$$\approx \psi^2\big(\tilde{\lambda}_M - \tilde{\lambda}_N\big)^2\sum_i \min\left\{\frac{\lambda_i}{\tilde{\lambda}_N^2},\ \frac{1}{\lambda_i}\right\}\min\left\{\frac{\lambda_i}{\tilde{\lambda}_M},\ 1\right\}.$$

Putting these two bounds together completes the proof. $\qquad\square$

### E.4 PROOF OF COROLLARY 5.4

*Proof of Corollary 5.4.* Under the assumptions we have

$$\mu_M \approx \frac{1}{M}.$$

We first compute ridge regression based on Corollary 5.2.

**The uniform case.** When $\lambda_i = 1/s$ for $i \le s$ and $\lambda_i = 0$ for $i > s$, we have

$$\mathcal{L}(h; \mathbf{X}) - \sigma^2 \asymp \psi^2 \sum_i \min\left\{\mu_M, \lambda_i\right\}$$

$$\asymp \sum_{i=1}^{s} \min\left\{\frac{1}{M}, \frac{1}{s}\right\}$$

$$\asymp \min\left\{1, \frac{s}{M}\right\}.$$

**The polynomial case.** When $\lambda_i = i^{-a}$ for $a > 1$, we have

$$\mathcal{L}(h; \mathbf{X}) - \sigma^2 \asymp \psi^2 \sum_i \min\left\{\mu_M, \lambda_i\right\}$$

$$\asymp \sum_i \min\left\{\frac{1}{M}, i^{-a}\right\}$$

$$\asymp M^{\frac{1}{a}-1}.$$

**The exponential case.** When $\lambda_i = 2^{-i}$, we have

$$\mathcal{L}(h; \mathbf{X}) - \sigma^2 \asymp \psi^2 \sum_i \min\left\{\mu_M, \lambda_i\right\}$$

$$\asymp \sum_i \min\left\{\frac{1}{M}, 2^{-i}\right\}$$

$$\asymp \frac{\log(M)}{M}.$$

We next compute the average risk of the attention model based on Theorem 5.3. Notice that

$$(\mu_M - \mu_N)^2 \asymp \left(\frac{1}{M} - \frac{1}{N}\right)^2 \asymp \frac{1}{M^2}, \quad \text{if } M < N/c \text{ for some constant } c > 1.$$

**The uniform case.** When $\lambda_i = 1/s$ for $i \le s$ and $\lambda_i = 0$ for $i > s$, we have

$$\mathbb{E}\mathcal{L}(f; \mathbf{X}) - \sigma^2 \asymp \psi^2 \sum_i \min\left\{\mu_M, \lambda_i\right\}$$

$$+ \psi^2 (\mu_M - \mu_N)^2 \sum_i \min\left\{\frac{\lambda_i}{\mu_N^2}, \frac{1}{\lambda_i}\right\} \min\left\{\frac{\lambda_i}{\mu_M}, 1\right\}$$

$$\asymp \min\left\{1, \frac{s}{M}\right\} + \frac{1}{M^2} \sum_{i=1}^{s} \min\left\{\frac{1}{s}N^2, s\right\} \min\left\{\frac{1}{s}M, 1\right\}$$

$$\asymp \min\left\{1, \frac{s}{M}\right\} + \sum_{i=1}^{s} \min\left\{\frac{1}{s}\frac{N^2}{M}, s\frac{1}{M}\right\} \min\left\{\frac{1}{s}, \frac{1}{M}\right\}$$

$$\asymp \min\left\{1, \frac{s}{M}\right\} + \begin{cases} \dfrac{s^2}{M^2}, & s \le M < N/c; \\[2mm] \dfrac{s}{M}, & M < s \le N/c; \\[2mm] \dfrac{N^2}{sM}, & M < N/c < s \end{cases}$$

$$\asymp \begin{cases} \dfrac{s}{M}, & s \le M < N/c; \\[2mm] \dfrac{s}{M}, & M < s \le N/c; \\[2mm] 1 + \dfrac{N^2}{sM}, & M < N/c < s. \end{cases}$$

So when $s < M$, or $s > N^2/M$, we have

$$\mathbb{E}\mathcal{L}(f; \mathbf{X}) - \sigma^2 \asymp \min\left\{\frac{s}{M}, 1\right\}.$$

**The polynomial case.** When $\lambda_i = i^{-a}$ for $a > 1$, we have

$$\mathbb{E}\mathcal{L}(f; \mathbf{X}) - \sigma^2 \asymp \psi^2 \sum_i \min\left\{\mu_M, \lambda_i\right\}$$

$$+ \psi^2 (\mu_M - \mu_N)^2 \sum_i \min\left\{\frac{\lambda_i}{\mu_N^2}, \frac{1}{\lambda_i}\right\} \min\left\{\frac{\lambda_i}{\mu_M}, 1\right\}$$

$$\asymp M^{\frac{1}{a}-1} + \frac{1}{M^2} \sum_i \min\left\{i^{-a}N^2, i^a\right\} \min\left\{i^{-a}M, 1\right\}$$

$$\asymp M^{\frac{1}{a}-1} + \sum_i \min\left\{i^{-a}\frac{N^2}{M}, i^a\frac{1}{M}\right\} \min\left\{i^{-a}, \frac{1}{M}\right\}$$

$$\asymp M^{\frac{1}{a}-1} + \sum_{i \leq M^{\frac{1}{a}}} i^a \frac{1}{M}\frac{1}{M} + \sum_{M^{\frac{1}{a}} < i \leq N^{\frac{1}{a}}} i^a \frac{1}{M}i^{-a} + \sum_{i > N^{\frac{1}{a}}} i^{-a}\frac{N^2}{M}i^{-a}$$

$$\asymp M^{\frac{1}{a}-1} + M^{\frac{1}{a}-1} + N^{\frac{1}{a}}M^{-1} + N^{\frac{1}{a}}M^{-1}$$

$$\asymp N^{\frac{1}{a}}M^{-1}.$$

**The exponential case.** When $\lambda_i = 2^{-i}$, we have

$$\mathbb{E}\mathcal{L}(f; \mathbf{X}) - \sigma^2 \asymp \psi^2 \sum_i \min\left\{\mu_M, \lambda_i\right\}$$

$$+ \psi^2 (\mu_M - \mu_N)^2 \sum_i \min\left\{\frac{\lambda_i}{\mu_N^2}, \frac{1}{\lambda_i}\right\} \min\left\{\frac{\lambda_i}{\mu_M}, 1\right\}$$

$$\asymp \frac{\log(M)}{M} + \frac{1}{M^2} \sum_i \min\left\{2^{-i}N^2, 2^i\right\} \min\left\{2^{-i}M, 1\right\}$$

$$\asymp \frac{\log(M)}{M} + \sum_i \min\left\{2^{-i}\frac{N^2}{M}, 2^i\frac{1}{M}\right\} \min\left\{2^{-i}, \frac{1}{M}\right\}$$

$$\asymp \frac{\log(M)}{M} + \sum_{i \leq \log(M)} 2^i\frac{1}{M}\frac{1}{M} + \sum_{\log(M) < i \leq \log(N)} 2^i\frac{1}{M}2^{-i}$$

$$+ \sum_{i > \log(N)} 2^{-i}\frac{N^2}{M}2^{-i}$$

$$\asymp \frac{\log(M)}{M} + \frac{1}{M} + \frac{\log(N)}{M} + \frac{1}{M}$$

$$\asymp \frac{\log(N)}{M}.$$

We have completed our calculation. $\qquad\square$

