# OpenReview forum: "How Many Pretraining Tasks Are Needed for In-Context Learning of Linear Regression?"
_ICLR.cc/2024/Conference — ICLR 2024 spotlight_

### Official Review · Reviewer_dEpy · 2023-10-23

**Soundness:** 4 excellent
**Presentation:** 4 excellent
**Contribution:** 3 good
**Rating:** 6
**Confidence:** 3

**Summary:**

Using a modified version of a single-layer linear attention model, the authors derive a dimension-independent complexity bound which suggests that efficient pretraining is possible even with a large number of model parameters for effective in-context learning. In the process, the authors demonstrate novel techniques for analyzing higher-order tensors that may be independently applicable.

**Strengths:**

- The authors provide a timely contribution to an important phenomenon of in-content learning.
- I am positive that this work provides a good theoretical checkpoint for future work to build on top of the provided analysis.
- The text is fairly well-presented and the research community will appreciate it as such.
- I also specially highlight that the authors have made proper effort to delineate the assumptions behind the theoretical results.

**Weaknesses:**

- The obvious shortcomings of the theoretical results come from the nature of assumptions that deviate from standard practice - the choice of linear attention and a restricting the structure of Q/K/V matrices. Also see Question 1.
- While the community is focused on Transformers, I am wondering if the analysis also holds for a different linear parametrization of the function $f$. In the broader context, if similar results hold for another parametrization, then attention would not appear that unique of a function. As an example, imagine a different parametrization that also leads to a dimension-free bound. In the broader context, such a result would then not distinguish what attention brings to the table, when in practice we have ample evidence that other parametrizations don't carry as flexible and generalizable inductive biases.
- Regarding the distributional assumptions of the fixed sized dataset in Assumption 1, it would be great to have experiments where the model is misspecified. Please correct me if I missed, but I don't think I see such an experiment in Appendix A. Misspecification is really important, since for all practical purposes, our models are misspecified, i.e. the data does not really come from the distribution we assume to be. If Transformers can still achieved a good decay of the empirical risk as the number of pretraining tasks increase, it would certainly be a unique characteristic. Also see Question 2.
- I would strongly recommend highlighting the experiments and moving them up to the main text. Perhaps Section 6 could be compressed a little to accommodate.

**Questions:**

1. Could the authors confirm if the restrictions are significantly restrictive? It would appear significant at face value. I certainly don't discount the fact that the work aims to provides theoretical grounding for similar observations in literature.
2. In Theorem 4.1, how is the inner product between matrices defined?
3. It would appear that the theorems do not necessarily say anything about misspecification, except that the excess risk is controlled via terms in Equation 8. In a sense, the Gaussian assumptions are used for derivations so anything in the Gaussian family would qualify for the bound. Is that the correct assessment?

---

> ### Author Response · Authors · 2023-11-20
> **Response to Reviewer dEpy**
>
> Thank you for your positive feedback!
>
> ---
>
> **Q1**. The obvious shortcomings of the theoretical results come from the nature of assumptions that deviate from standard practice - the choice of linear attention and a restricting the structure of Q/K/V matrices… Could the authors confirm if the restrictions are significantly restrictive?... I certainly don't discount the fact that the work aims to provide theoretical grounding for similar observations in literature.
>
> **A1**. We acknowledge that our setup deviates from practice by considering linear attention and linear reparameterization. We think the statistical bounds of SGD-trained $\\Gamma$ and SGD-trained standard softmax attention model might not be identical. One challenge is that our loss function is convex, but the loss for standard softmax attention is non-convex. So there will be an extra layer of complexity from non-convex optimization in the latter setup. With that being said, our results still provide meaningful insights into the statistical task complexity of ICL. Without a full understanding of the statistical behavior of SGD in the simplified setup, it seems difficult that one can achieve this in more complicated cases.
>
> ---
>
> **Q2**. “While the community is focused on Transformers, I am wondering if the analysis also holds for a different linear parametrization of the function $f$. In the broader context, if similar results hold for another parametrization, then attention would not appear that unique of a function… In the broader context, such a result would then not distinguish what attention brings to the table, when in practice we have ample evidence that other parametrizations don't carry as flexible and generalizable inductive biases.”
>
> **A2**. We would like to highlight that attention provides a natural way to parameterize gradient descent (GD), which is a key inductive bias for the model to achieve Bayes-optimal ICL. In equation (1), we perform a linear reparameterization to simplify our analysis, but our simplification still allows equation (1) to realize a GD procedure (in terms of the mean square loss). Other linear parameterization of $f$, while might be easy to pretrain, may not correspond to a meaningful GD procedure, so it might not be able to achieve Bayes-optimal ICL. Therefore, the inductive bias carried by the attention mechanism is not easily replaceable according to our theory in Section 5.
>
> ---
>
> **Q3**. ... it would be great to have experiments where the model is misspecified.… It would appear that the theorems do not necessarily say anything about misspecification…. the Gaussian assumptions are used for derivations so anything in the Gaussian family would qualify for the bound. Is that the correct assessment?
>
> **A3**. This is a great point. You are correct that neither our current theory nor our current experiments cover misspecified linear regression. Our theory can be extended to deal with non-Gaussian distributions with proper (up to 8-th) moment conditions. However, extra efforts would be required to extend our theory to deal with the correlations between noise and covariates.
>
> According to your suggestion, we conduct experiments for our attention model in misspecified settings. Please refer to Figure 4 in Appendix A in the revised paper. When we replace the Gaussian noise with an independent and uniformly distributed noise, the attention model can still perform good ICL. However, when the expected response is generated from a nonlinear function (e.g., square and sigmoid functions), the performance of ICL will decrease.
>
> ---
>
> **Q4**. I would strongly recommend highlighting the experiments and moving them up to the main text. Perhaps Section 6 could be compressed a little to accommodate.
>
> **A4**. We will move our numerical results to the main body provided we are granted additional pages in the final version.
>
>
> ---
>
> **Q5**. In Theorem 4.1, how is the inner product between matrices defined?
>
> **A5**. For two matrices $A$ and $B$, we define $\\langle A, B \\rangle = \\text{tr}(A\^\\top B)$. We have clarified this (in blue text) before Theorem 3.1 in the revised paper.

---

> > ### Comment · Reviewer_dEpy · 2023-11-20
> > **Response to authors**
> >
> > Thank you for all your comments.

---

### Official Review · Reviewer_m56U · 2023-10-31

**Soundness:** 3 good
**Presentation:** 3 good
**Contribution:** 3 good
**Rating:** 8
**Confidence:** 3

**Summary:**

This paper introduces a statistical task complexity bound for a one-layer linear attention model in solving fixed-length linear regression problems. The authors investigate the required number of independent sequences or tasks necessary for pretraining the model. They conclude that only a small number of sequences are needed compared to the model parameters, and this is independent of the input dimension. Additionally, the paper theoretically demonstrates a performance drop when the number of in-context examples during evaluation differs from the training phase.

**Strengths:**

1. The paper is well-organized, with clear contributions and promising results in the statistical analysis of in-context learning for linear regression problems using a one-layer linear attention model.
2. The theoretical results provided in the paper address the sample complexity of learning linear regression problems, highlighting that it is approximately $O(1/T)$. This indicates that the number of tasks required for learning the model is independent of both the model size and the input dimension ($d$).
3. The paper underscores the significance of context length during testing, proving that optimal prediction is attainable only when the number of context examples matches that of the pre-training phase.

**Weaknesses:**

1. While the theoretical analysis presented is commendable and contributes to our understanding, the paper could be greatly enhanced by including more empirical results. Given that implementing linear regression over a transformer model is feasible and has been achieved in prior work, specific implementations that could improve this paper include:

    - Demonstrating through examples that training models on linear tasks of varying dimensions yields similar performance, supporting the paper's claim that task complexity is independent of the task dimension $d$.
    - Conducting experiments to show that a pre-trained model performs poorly when there is a significant discrepancy between the in-context length during testing and training.
2. The paper’s assertion that task complexity is independent of $d$ is counterintuitive. It is typically expected that the number of tasks required would be on the order of $O(d^2/T)$ since the task distribution or covariance, encompassing at least $d^2$ parameters, needs to be retrieved. The authors are encouraged to provide additional clarification on this aspect of independence. Furthermore, the experiments presented seem to contradict the paper’s statement, as $T$ appears to be exponentially larger than $d^2$, which doesn’t align with the claim that "$T$ could be much smaller than $d^2$".

3. Despite Equation (9) not explicitly showing dependence on $d$, it seems that $d$ is implicitly involved in the matrix operations, which should be addressed for clarity.

**Questions:**

1. Is it possible to ensure that the first term of Equation (9) will consistently reach zero? Given the exponentially decreasing learning rate $\gamma_t$, this relationship does not seem immediately apparent.
2. In Theorem 4.1, the learning rate $\gamma_0$ is upper bounded. Could the authors elaborate on the reasoning behind this? Is it connected to the implicit convergence rate of gradient descent in linear problems? Additionally, what are the considerations and trade-offs in selecting an appropriate initial learning rate $\gamma_0$?

---

> ### Author Response · Authors · 2023-11-20
> **Response to Reviewer m56U**
>
> Thank you for your positive feedback!
>
> ---
>
> **Q1-1**. Demonstrating through examples that training models on linear tasks of varying dimensions yields similar performance, supporting the paper's claim that task complexity is independent of the task dimension $d$....
>
> **A1-1**. We have provided Figure 2 in Appendix A in the revised paper, which plots curves for risk vs. dimension. The results are consistent with our theory and suggest that the risk does not explicitly depend on the ambient dimension.
>
> ---
>
> **Q1-2**. Conducting experiments to show that a pre-trained model performs poorly when there is a significant discrepancy between the in-context length during testing and training.
>
> **A1-2**. We have provided Figure 3 in Appendix A in the revised paper, plotting curves for risk vs. number of context examples. We observe that when $M$ and $N$ are different (up to $4$ times), the pretrained model exhibits relatively stable performance compared to optimal ridge regression. We suspect that only when $M$ and $N$ are drastically different, the pre-trained model will deviate from optimal ridge regression. We will run additional experiments to verify this.
>
> ---
>
>
> **Q2**. The paper’s assertion that task complexity is independent of $d$ is counterintuitive. It is typically expected that the number of tasks required would be on the order of $O(d\^2 / T)$ since the task distribution or covariance, encompassing at least $d\^2$ parameters, needs to be retrieved. …Furthermore, the experiments presented seem to contradict the paper’s statement, as $T$ appears to be exponentially larger than $d\^2$, which doesn’t align with the claim that "$T$ could be much smaller than $d\^2$
>
> **A2**. We would like to clarify that our effective dimension $D\_{\\text{eff}}$ does not explicitly depend on $d$ but it still depends on the eigenvalues of $H$. Though we always have $D\_{\\text{eff}} \\le d\^2$, the equality can be attained in the worst case (e.g., $H=I$ and $T$ is large). Therefore, our bound in the worst case would be on the order of $\\tilde{O}(d\^2 / T)$, which is consistent with your intuition.
>
> For Figure 1(a) in the initial version, we would like to point out that $d\^2 = 100\^2 \approx 2\^{13.28}$, so the sample complexity is not exponential in the number of parameters and does not contradict our theory.
>
> ---
>
> **Q3**. Despite Equation (9) not explicitly showing dependence on $d$, it seems that $d$ is implicitly involved in the matrix operations, which should be addressed for clarity.
>
> **A3**. Please refer to our **A2** in the above.
>
> ---
>
> **Q4**. Is it possible to ensure that the first term of Equation (9) will consistently reach zero? Given the exponentially decreasing learning rate $\\gamma\_t$, this relationship does not seem immediately apparent.
>
> **A4**. Yes. Note that the learning rate is piecewise constant and is only decreased by a constant factor after every $\\log(T)$ steps. So the constant initial learning rate $\gamma_0$ is used in the first $T/\\log(T)$ steps. As a consequence, the first term in equation (9) tends to zero consistently as $T$ tends to infinity.
>
> ---
>
> **Q5**. In Theorem 4.1, the learning rate $\\gamma\_0$ is upper bounded. Could the authors elaborate on the reasoning behind this? Is it connected to the implicit convergence rate of gradient descent in linear problems? Additionally, what are the considerations and trade-offs in selecting an appropriate initial learning rate $\\gamma\_0$?
>
> **A5**. If the initial learning rate $\\gamma\_0$ is too large, the SGD could diverge. The upper bound on $\\gamma\_0$ is a technical condition to ensure the convergence of SGD.
>
> For a larger initial learning rate, the first term (i.e., bias error) in equation (9) will be smaller but the second term (i.e., variance error) in equation (9) will be larger. So an optimal initial learning rate requires a balance between the bias and variance errors. Prior knowledge such as the norm of $\\Gamma\^*\_N$ and the scale of $\\psi\^2 tr(H) + \\sigma\^2$ could be useful in choosing the initial learning rate.

---

> > ### Comment · Reviewer_m56U · 2023-11-22
> >
> > Thank you to the authors for their response. After reviewing the revised paper and the additional experiments provided, I have decided to increase my evaluation score. However, the paper would benefit from a more thorough discussion on the implicit relationship between data dimension and sample complexity, as well as the choice of learning rate.

---

> > > ### Author Response · Authors · 2023-11-22
> > > **Thank you**
> > >
> > > Thank you for increasing your score. We will be sure to add our rebuttal discussions on the implicit relationship between data dimension and sample complexity and the choice of learning rate to the final version.

---

### Official Review · Reviewer_HbBb · 2023-11-01

**Soundness:** 3 good
**Presentation:** 3 good
**Contribution:** 3 good
**Rating:** 8
**Confidence:** 3

**Summary:**

The paper dives into the statistical understanding of pre-training of a single linear layer attention on in-context examples generated with Gaussian data and linear regression with a Gaussian prior.  The important theoretical contributions can be summarized as follows.

(a) The authors first show the optimal solution and its excess risk in terms of the pre-training sequence lengths and the covariance matrix underlying the Gaussian data.

(b) The authors then show the behavior of the excess risk with training steps via SGD, where each step uses a randomly sampled sequence from the underlying distribution.

(c) The authors further show that when evaluation sequence lengths match to the training sequence lengths, the model matches the Bayes optimal solution. However, discrepancies between the two can lead to sub-optimalty.

Overall, the paper takes an important step toward statistical understanding of the dependence of pre-training data and in-context abilities of attention models.

**Strengths:**

The main strength of the paper lies in its clinical approach to relate the existing literature on ridge regression with Gaussian prior to the statistical understanding of linear attention pre-training. Furthermore, the authors introduce novel techniques like operator polynomials to solve order-8 tensors that show up in the risk analysis of SGD training, which might be of independent interest to the community.

The advantages of the theoretical framework can be summarized as follows. First, a single linear attention model reaches the optimal linear regression solution with SGD. The framework gives statistical convergence bounds with dependence on training sequence lengths and the data covariance matrix. Secondly, one can pinpoint the gaps between evaluation and training based on the discrepancies in data properties e.g. sequence length. Overall, the paper will be an important addition to the theory community.

**Weaknesses:**

Overall, the paper doesn't have many pitfalls and issues as is.

I have a question about the theoretical setup. Linear attentions used in practice have a query $Q$, key $K$, and value $V$ matrix. However, the authors use structural modifications in $Q$, $K$, and $V$ to represent the formulation with a single matrix $\Gamma$ and compute SGD convergence of $\Gamma$. Without the modification, I believe the optimal solution can be shown to be a $3$-matrix factorization of the optimal solution $\Gamma^*$ given in theorem 1. But what will be the statistical bounds of training these $3$ matrices (or any pair among the $3$ matrices)? Can the authors discuss whether it is answerable from their theoretical framework and if not, the difficulties one might face to solve?

Furthermore, how will the theory change when instead of using training sequences of a single length, we use randomly sampled training sequences with varying lengths? How will the excess risk in theorem 5.3 change then?

The authors also conduct a few experiments in the appendix on a real-world transformer to verify their theoretical claims. It would be interesting to empirically check the training time convergence with different singular value behaviors (as they pick in corollary 4.2) and observe differences in convergence and ICL performance throughout training.

Finally, I haven't looked deeply into the proof, since it is extremely long to read through. But at a glance, the paper seems to be utilizing similar proof techniques that prior works have used for ridge regression with Gaussian prior. Hence, I still recommend strong acceptance.

**Questions:**

Please see my questions in the previous section.

---

> ### Author Response · Authors · 2023-11-20
> **Response to Reviewer HbBb**
>
> Thank you for your strong support!
>
> ---
>
> **Q1**. ... Without the modification, I believe the optimal solution can be shown to be a 3-matrix factorization of the optimal solution $\\Gamma\^*$ given in theorem 1. But what will be the statistical bounds of training these 3 matrices (or any pair among the 3 matrices)? Can the authors discuss whether it is answerable from their theoretical framework and if not, the difficulties one might face to solve?
>
> **A1**. You are correct that the optimal solution can be shown to be a 3-matrix factorization, which can be seen from Appendix B. We think the statistical bounds of SGD-trained $\\Gamma$ and SGD-trained 3 matrices might not be identical. One challenge is that the loss is convex for $\Gamma$ but is non-convex for $Q, K, V$. So there will be an extra layer of complexity from non-convex optimization when SGD is run for the 3 matrices. Our techniques for analyzing SGD in $\Gamma$ might not be enough to deal with this non-convex optimization issue. With that being said, we believe our results still provide meaningful insights into the statistical task complexity of ICL.
>
> ---
>
> **Q2**. … how will the theory change when instead of using training sequences of a single length, we use randomly sampled training sequences with varying lengths? How will the excess risk in theorem 5.3 change then?
>
> A2. This is a good question. A model pretrained with a varying context length cannot match the Bayes optimal method for every $M \le N$ (in the setup of Theorem 5.3). This is because to achieve Bayes optimality with $M$ context examples, the model in equation (1) needs to have a parameter $\Gamma$ as a function of $M$. However, the pretrained model in equation (1) has a fixed model parameter (even when the context length is varying during pretraining) that cannot adapt to a varying context length $M$, therefore such a model cannot be Bayes optimal uniformly for $M$.
>
> ---
>
> **Q3**. … It would be interesting to empirically check the training time convergence with different singular value behaviors (as they pick in corollary 4.2) and observe differences in convergence and ICL performance throughout training.
>
> A3. Thank you for your suggestion. We have empirically checked the pretraining convergence under two $H$’s of different spectrums. Please refer to Figure 1 in Appendix A in the revised paper. We can see that under polynomial eigen-decay ($i\^{-4}$) and exponential eigen-decay ($2\^{-i}$), ICL exhibits slightly different convergence behaviors though the trend is the same, which are aligned with our theory.

---

### Official Review · Reviewer_ozh2 · 2023-11-05

**Soundness:** 3 good
**Presentation:** 2 fair
**Contribution:** 3 good
**Rating:** 5
**Confidence:** 4

**Summary:**

The paper investigates a single-layer linear attention model, which is equivalent to a linear model with parameters derived from a one-matrix-step gradient descent from the origin. The authors make Gaussian data generating assumptions and consider the population ICL risk, identifying the optimal step size. They demonstrate that applying gradient descent to the step size parameterization can lead to an excess risk characterized by an exponential decay plus a 1/T-like decay. The authors also compare this with the Bayes optimal estimator, analyzing their respective risks.

**Strengths:**

The paper's originality lies in analyzing the single-layer linear attention model with its one-step gradient descent parameterization. The authors' identification of the optimal step size under Gaussian data generating assumptions is a valuable contribution. The quality of the paper is evident in the rigorous mathematical analysis.

**Weaknesses:**

The paper's clarity could be improved.

- The assumption of the pretraining algorithm in equation (6) seems arguable, as the equivalence of the function classes does not necessarily imply identical behavior under different parameterizations in gradient descent.
- The paper is heavily reliant on mathematical notation and could benefit from more intuitive explanations to aid understanding.
- The choice of step size in Theorem 4.1 is not adequately justified, and the assumption that the initialization Γ0 commutes with H is not clearly motivated.
- The assumption that H can be diagonalized without loss of generality (WLOG) is also not sufficiently explained.
- The paper could be enhanced by including some numerical results in the main body.
- The terminology used is occasionally confusing, such as the use of "number of contexts".

**Questions:**

- Could you provide more justification for the pretraining algorithm assumed in equation (6)?
- Could you explain the choice of step size in Theorem 4.1?
- Why is it reasonable to assume that the initialization Γ0 commutes with H in Theorem 4.1?
- Could you provide more insight into why H can be assumed to be diagonal WLOG?
- Would it be possible to include some numerical results in the main body to support the theoretical findings?
- Could you clarify the term "number of contexts"? It reads like the number of in-context samples, but I guess you mean the number of sequences/datasets.

---

> ### Author Response · Authors · 2023-11-20
> **Response to Reviewer ozh2**
>
> Thank you for your constructive feedback.
>
> ---
>
> **Q1**. The assumption of the pretraining algorithm in equation (6) seems arguable, as the equivalence of the function classes does not necessarily imply identical behavior under different parameterizations in gradient descent… Could you provide more justification for the pretraining algorithm assumed in equation (6)?
>
> **A1**. Equation (6) corresponds to the one-pass SGD update for parameter $\\Gamma$, which is a natural algorithm. Considering SGD in $\\Gamma$ space is meaningful as it provides insights into the statistical hardness of learning the ICL model. We agree with you that the behavior of SGD in $\\Gamma$ parameter space is not identical to that of SGD in $Q, K, V$ parameter space. For example, the loss is convex for $\\Gamma$ but is non-convex for $Q, K, V$. Nevertheless, considering SGD in $\\Gamma$ space simplifies the challenge from non-convex optimization.
>
> Extending our results to more complicated $Q, K, V$ parameterization is an important direction. However, even for the simplified $\\Gamma$ parameterization, there is no such kind of result before our work. Without a comprehensive understanding of the statistical behavior of SGD in the simpler $\\Gamma$ space, achieving this in the more complex $Q, K, V$ space appears challenging. As our work is the first one on statistical pretraining task complexity for ICL, we believe this is an important step in this direction and the contribution of our work is significant.
>
> ---
>
> **Q2**. Could you explain the choice of step size in Theorem 4.1?
>
> **A2**. Equation (7) characterizes a decreasing, piecewise-constant stepsize scheduler. Specifically, the stepsize is initialized from $\\gamma\_0$ and is decreased by a constant factor (we choose 2 in the paper) for every $T/\\log(T)$ steps, where $T$ is the total number of iterates. This is a commonly used stepsize scheduler for SGD in deep learning.
>
> ---
>
> **Q3**. Why is it reasonable to assume that the initialization $\\Gamma\_0$ commutes with H in Theorem 4.1?
>
> **A3**. Our assumption is reasonable as it holds when setting $\\Gamma\_0$ to zero or a scalar matrix, that is, $\\Gamma\_0 = c I$ where $c$ is a constant, without prior information. Additionally, it allows setting $\\Gamma\_0$ to any matrix when $H$ is a scalar matrix.
>
> ---
>
> **Q4**. Could you provide more insight into why $H$ can be assumed to be diagonal WLOG?
>
> **A4**. We assume that $H$ is diagonal to simplify our discussion in “Key idea 1: diagonalization” (and related places in the appendix). This is made without loss of generality. If $H$ is not diagonal, let $H = V \\Lambda V\^\\top$ be the eigen-decomposition of $H$, then we can replace “diagonal matrices” with “matrices of form $V D V\^\\top$ where $D$ is a diagonal matrix”, and our arguments all go through. Taking equation (30) in Appendix D.4 as an example, we re-define a “diagonalization of an operator” by
> $$
> \\mathring {\\mathcal{O}} (A) = V \\text{diag} ( {\\mathcal{O}} ( \\text{diag}(V\^\\top A V)  ) ) V\^\\top.
> $$
> In this way, the current proof adapts to the eigenspace of $H$.
>
> ---
>
> **Q5**. Would it be possible to include some numerical results in the main body to support the theoretical findings?
>
> **A5**. We will move our numerical results (see Appendix A) to the main body provided we are granted additional pages in the final version.
>
> ---
>
> **Q6**. Could you clarify the term "number of contexts"? It reads like the number of in-context samples, but I guess you mean the number of sequences/datasets.
>
> **A6**. The term “number of contexts” indeed refers to the “number of in-context samples”.

---

> ### Author Response · Authors · 2023-11-22
> **Follow up**
>
> Dear Reviewer ozh2, we hope all of your concerns have been addressed. Please let us know if there is anything else that requires our clarification!

---

### Meta-Review · Area_Chair_D8oi · 2023-12-05

**Metareview:**

This paper studies the sample complexity of pretraining a transformer to do in-context linear regression, focusing on the case of a single-layer linear attention model. By reformulating into a convex quadratic problem in terms of a single matrix (reparametrizing from the QKV matrices), the paper analyzes the sample complexity of pretraining on finitely many instances using SGD, and also compares the resulting optimal one-layer model to the Bayes optimal ridge predictor. The sample complexity improves over existing work which depends directly on parameter count.

The reviewers generally find this paper to be a solid theoretical work on in-context learning in transformers, and could be an important addition to the theory community. The technical tools for analyzing higher-order tensors could also be of further interest.

I additionally encourage the authors to emphasize the following points at significant places (some of them already suggested by the reviewers):
* The sample complexity (9) still scales with $\tilde{O}(d^2/T)$ in the standard case of $H=I_d$. The improvement are in low-rank or decaying spectrum cases;
* The optimal one-layer model $\Gamma_N^*$ is not the same as the optimal ridge predictor (which is not in the one-layer model class) *at finite $N$*. This may be obvious from the fact that you compare them later, but it may be good to emphasize this early on. Further, this is not to be confused with the fact that they are both consistent as $N\to\infty$ (I think the paragraph after Theorem 3.1 claiming the "optimality" of the best one-layer model at $N\to\infty$ could be misleading on this end, and could be revised);
* The fact that the algorithmic result (SGD) is directly available, depends strongly on the convex reformulation (as well as the one-layer linear attention model). A discussion on whether a statistical risk analysis is available using original (non-convex) QKV formulation without algorithmic result may be helpful.

**Justification For Why Not Higher Score:**

The convex reformulation as well as the one-layer linear attention model used in the analysis are quite specific (though understandably, in order to obtain the fine-grained results), and may not resemble the behavior of transformers in reality.

**Justification For Why Not Lower Score:**

The theoretical results are solid and could be of interest to both the theory and the understanding transformers community.

---

### Decision · Program_Chairs · 2024-01-16

Accept (spotlight)